# SPhyR: Spatial-Physical Reasoning Benchmark on Material Distribution

## Abstract

We introduce a novel dataset designed to benchmark the physical and spatial reasoning capabilities of Large Language Models (LLM) based on topology optimization, a method for computing optimal material distributions within a design space under prescribed loads and supports. In this dataset, LLMs are provided with conditions such as 2D boundary, applied forces and supports, and must reason about the resulting optimal material distribution. The dataset includes a variety of tasks, ranging from filling in masked regions within partial structures to predicting complete material distributions. Solving these tasks requires understanding the flow of forces and the required material distribution under given constraints, without access to simulation tools or explicit physical models, challenging models to reason about structural stability and spatial organization. Our dataset targets the evaluation of spatial and physical reasoning abilities in 2D settings, offering a complementary perspective to traditional language and logic benchmarks [1].

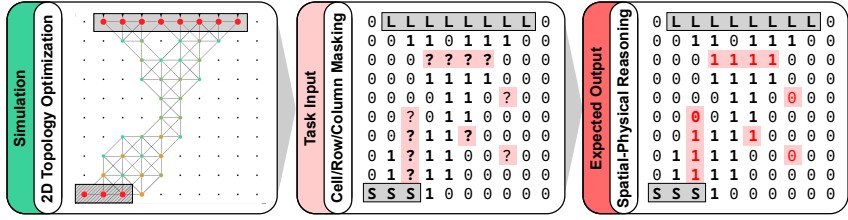

Figure 1: Topology Optimization is used to calculate material distribution. Masking individual cells, rows, columns or the complete distribution space offer challenging spatial physical reasoning tasks.

## 1 Introduction

Large language models (LLMs) have achieved strong performance on linguistic and logical tasks, but their ability to reason about physical systems and spatial structures remains underexplored Zhang et al. (2025). Existing benchmarks primarily probe either visual perception or text-based commonsense knowledge, but few explicitly test reasoning grounded in physical constraints.

For example, visual question-answering benchmarks such as CLEVR focus on object attributes and spatial relations in synthetic scenes Johnson et al. (2016), while intuitive physics datasets like IntPhys and Physion evaluate models' ability to predict or assess the plausibility of physical events in videos Riochet et al. (2020); Bear et al. (2022). Interactive environments such as PHYRE Bakhtin et al. (2019) and stability-focused datasets like ShapeStacks Groth et al. (2018) further probe causal reasoning and contact mechanics, whereas text-based datasets such as PIQA Bisk et al. (2019) and PhysReason Zhang et al. (2025) target physical commonsense and multi-step problem solving in language form.

Existing benchmarks have advanced our understanding of physical reasoning in LLMs, but they largely focus on object dynamics, intuitive physics, or qualitative predictions. They do not evaluate

---

[1]Huggingface Dataset: `anonymized`
Data Generation and Model Evaluation Code: `https://anonymous.4open.science/r/SPhyR-587C`

whether models can reason about how forces should be supported and transmitted through a structure, a capability fundamental to engineering and design. This gap leaves untested a crucial class of reasoning that requires integrating spatial layout with structural principles such as load paths, stiffness, and stability. Beyond physical reasoning, recent work like ARC-AGI-2 Chollet et al. (2025) has introduced grid-based tasks for testing abstract reasoning and generalization. While unrelated to physics, this work highlights the value of structured 2D representations for isolating reasoning capabilities. We build on this intuition but shift the focus from symbolic transformations to spatially grounded physical reasoning.

To address this gap, we introduce **SPhyR**, a new benchmark for evaluating spatial and physical reasoning in LLMs. SPhyR formulates topology optimization-inspired tasks in a grid-based format, where models must infer how to distribute material to support specified forces and constraints. By testing whether models can reason about load paths, stability, and structural connectivity from descriptions alone, SPhyR bridges the gap between language-based reasoning and physically grounded design tasks. We benchmark state-of-the-art LLMs on SPhyR and reveal fundamental limitations in their ability to integrate spatial and physical reasoning.

## 2 RELATED WORK

**Benchmarks for Physical and Spatial Reasoning**   A wide range of benchmarks probe models' understanding of physical and spatial reasoning (Table 1). CLEVR Johnson et al. (2016) evaluates visual reasoning about objects and spatial relations in synthetic scenes, while CLEVRER Yi et al. (2020) extends this to temporal and causal reasoning in videos. IntPhys Riochet et al. (2020) and Physion Bear et al. (2022) test whether models can predict or assess the plausibility of physical events, while ShapeStacks Groth et al. (2018) targets block stability prediction. In interactive settings, PHYRE Bakhtin et al. (2019) challenges agents to solve 2D physics puzzles by reasoning about actions and causal effects. Language-based datasets such as PIQA Bisk et al. (2019) and PhysReason Zhang et al. (2025) shift the focus from perception to textual physical reasoning, evaluating knowledge of everyday object interactions and multi-step physics problem solving, respectively.

While these benchmarks advance physical reasoning evaluation, they largely focus on event prediction or commonsense reasoning. None require models to determine optimal material arrangements under explicit load and support constraints - a capability crucial for real-world engineering reasoning.

| Benchmark | Format | Physical Reasoning | Spatial Reasoning | Notes |
|---|---|---|---|---|
| CLEVR (2017) | Visual QA | ✗ | ✓ | Scene reasoning |
| CLEVRER (2020) | Video QA | ✓ | ✓ | Causal events |
| IntPhys (2018) | Video plausibility | ✓ | ✓ | Violation detection |
| Physion (2021) | Video prediction | ✓ | ✓ | Object behavior prediction |
| ShapeStacks (2016) | Image classification | ✓ | ✓ | Block stability |
| PHYRE (2019) | 2D physics puzzles | ✓ | ✓ | Action planning |
| PIQA (2020) | Text QA | ✓ | ✗ | Physical commonsense |
| PhysReason (2023) | Text QA | ✓ | ✗ | Multi-step physics |
| **SPhyR (ours)** | Structured prediction | ✓ | ✓ | Material distribution |

Table 1: Comparison of existing benchmarks evaluating physical and spatial reasoning. Our proposed dataset (**SPhyR**) focuses specifically on material distribution reasoning under boundary conditions, combining spatial and physical understanding in structured tasks.

**Topology Optimization as a Benchmark**   Topology optimization (TO) Bendsøe & Sigmund (2004) is a well-established method for computing optimal material layouts in a domain under specified forces and supports. Prior work on Machine Learning (ML) in this space has focused on accelerating solvers or generating high-quality designs Banga et al. (2018); Rawat & Shen (2019). Our work repurposes topology optimization as a reasoning benchmark rather than a design tool. By framing it as a grid-based prediction problem, SPhyR tests whether LLMs can infer material distributions solely from boundary conditions and physical constraints - without access to solvers or simulation engines. This setup complements existing physical reasoning benchmarks by embedding spatial and physical structure into tasks that require more than pattern recognition.

**Machine Learning for Topology Optimization**   Prior machine learning work for Topology Optimization (TO) has focused on developing fast, high-fidelity solvers that can predict optimized material layouts with orders-of-magnitude speedup over conventional methods Banga et al. (2018); Rawat & Shen (2019); Zhang et al. (2020). These domain-specific approaches rely on embedding explicit structural knowledge, such as physics-informed loss functions or compliance constraints, into the model architecture and training process. In contrast, SPhyR evaluates general-purpose LLMs in a zero-shot setting, probing whether emergent, implicit physical knowledge acquired during broad training can substitute for explicitly learned physics.

**Structured Reasoning Beyond Physics**   Finally, our work connects to broader research on structured reasoning in grid-based environments. ARC-AGI-2 Chollet et al. (2025) tests abstract reasoning and generalization in symbolic, non-physical tasks. While ARC-AGI-2 and SPhyR share a structured representation, SPhyR introduces grounded physical constraints, bridging the gap between abstract symbolic reasoning and the physically grounded reasoning required for real-world design.

## 3   PROBLEM SETUP

**Topology Optimization Task**   Topology optimization determines an optimal material distribution within a domain under prescribed forces and supports. All dataset samples are generated using Millipede's density-based SIMP formulation, solving a minimum-compliance problem with a fixed volume fraction (Appendix B for solver parameters). This yields well-defined, single-objective solutions that capture characteristic load paths and material connectivity.

In this work, we repurpose these topology optimization instances as reasoning tasks for LLMs. Instead of performing numerical optimization, models must predict plausible material distributions from forces, supports, and boundaries alone, requiring them to infer principles of load transfer, stability, and efficient material use, approximating the behavior of minimum-compliance topology optimization without access to simulation tools.

**Input and Output Specification**   Each task instance in our benchmark is defined by a set of boundary conditions and a corresponding material distribution. The **inputs** provided to the model are: **2D boundaries**: A discretized 2D grid representing the spatial extent of the structure, **fixed supports**: Locations within the boundaries that act as load bearing supports and **applied forces**: Locations within the boundaries specifying external loads. The **output** expected from the model is a partial or complete **material distribution** over the domain grid, indicating where material should be placed to form a material optimized, that is minimum material distributed, but stable structure under the given boundary conditions. All inputs and outputs are represented in structured formats suitable for LLMs, through textual descriptions and serialized grids. No direct access to simulation results or numerical solvers is provided.

**Reasoning Challenges**   The tasks in our benchmark require a combination of physical and spatial reasoning that poses significant challenges for current large language models. First, models must infer how forces propagate through the structure, deciding where material is necessary to maintain stability and support loads. This involves understanding force paths, support connectivity, and load transfer-concepts that are rarely encountered in typical LLM training data. Second, models must reason spatially about the layout of material across a 2D grid. Predicting plausible completions requires local coherence (e.g., avoiding isolated material islands) as well as global structural organization (e.g., maintaining continuous load paths from forces to supports). Moreover, models must solve these tasks without explicit simulation tools or numerical methods. Instead, they must generalize from the provided boundary conditions and partial observations, synthesizing structures that satisfy implicit physical constraints. These reasoning demands span from local (individual cells or lines) to global (complete structures), creating a rich and graded challenge space for evaluating LLM capabilities beyond language-based tasks.

**Task Variations**   We define several task variations according to the nature and extent of the masked regions in the material distribution, and categorize them into two difficulty levels: easy and hard. Easy is distribution based on binary values such as material or no material, while hard is based on a continuous value range, 0 to 1. **N-Random Cell(s)**: Predict the material state of N randomly masked cell(s), where N is one of 1, 5 or 10. **N-Random Row(s)**: Predict the material state of N randomly

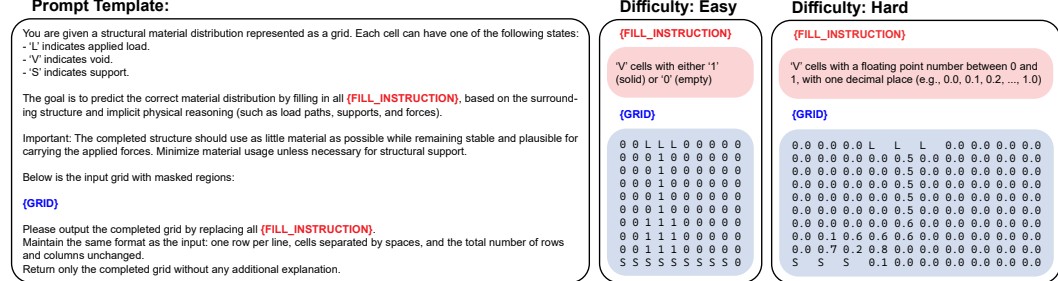

Figure 2: Prompt template used across all tasks and difficulty levels, showing instructions and grid format as served to models for evaluation.

masked row(s), where N is one of 1 or 3. **N-Random Columns(s)**: Predict the material state of N randomly masked columns(s), where N is one of 1 or 3. **Full Structure Prediction**: Predict the complete material distribution based only on boundary conditions. These variations allow us to systematically probe local and global reasoning abilities, from single-cell predictions to complex structural synthesis (Appendix C for samples).

## 4 DATASET DESCRIPTION

The SPhyR dataset was generated by solving 2D topology optimization problems, creating material distributions under various boundary conditions using the density-based solver Millipede Michalatos & Kaijima (2024). We constructed a set of 2D samples by systematically varying the positions of applied forces and supports, focusing on load-support scenarios typical of structural building design (load from the top, support on the bottom, ranging from 3 to 6 cells in width). Each material distribution was optimized for stiffness and efficiency using 10 solver iteration steps. The inherent variability in these boundary conditions ensures that tasks require generalization beyond memorization of fixed patterns (Appendix B, for detailed solver parameters).

**Dataset Statistics** The dataset consists of $10 \times 10$ structural optimization grids, balancing computational tractability with sufficient spatial complexity. In total, the dataset contains **1296** samples for all task variations and difficulties. These samples are organized into task-specific subsets, including cell completion, row/column completion, and full structure prediction, across both easy and hard difficulty levels. The full list of eight subject types (e.g., 1 Random Cell, 3 Random Row) and their descriptions is provided in Table C. Each sample includes structured representations of the boundary conditions and the corresponding ground truth material distribution.

**Input and Output Formats** Each sample in the dataset is represented as a structured input-output pair designed for compatibility with large language models. Samples are grouped into task-specific subjects, enabling targeted evaluation of different reasoning challenges.

The input consists of a natural language prompt that describes the task and defines the structural grid format. Within this grid, different symbols indicate key physical roles: **L** marks an applied load, **S** a support, and **V** (void) a masked cell whose material state must be predicted. Regions with known material values, whether binary or continuous, depending on the task difficulty (easy/hard), are explicitly included in the grid. The prompt provides clear instructions emphasizing structural plausibility and material efficiency, along with a grid where each row appears on a separate line and cell values are space-separated.

The expected output is a completed version of the same grid, where all V cells are replaced by predicted values (1 or 0) while preserving the original structure and formatting. No explanation or commentary is included in the output-only the raw grid content.

Each subject is labeled with a difficulty level. In easy variants, the ground truth material distribution is binary, focusing on high-level structural placement and discrete spatial reasoning. In hard variants, the underlying distributions are continuous or involve more complex structural dependencies, requiring finer-grained predictions and deeper reasoning about stress propagation and global support (Figure 2, for prompt template and Appendix J, E.1, for detailed model prompt and completion samples).

## 5 EVALUATION SETUP

### 5.1 EVALUATION METRICS

We evaluate model performance using three complementary families of metrics for a holistic assessment of both symbolic accuracy and structural realism: (1) **Reconstruction Accuracy Metrics**, quantifying cell-wise agreement with the ground truth, including measures of fidelity, penalization, and difficulty weighting; (2) **Topological Validity Metrics**, assessing global structural soundness through load-support connectivity and grid validity; and (3) **Physics-Approximating Metrics**, estimating the structural efficiency via gravity-aligned load-transfer paths. This comprehensive suite ensures robustness against simple pattern-matching success.

**Reconstruction Accuracy Metrics** We assess reconstruction fidelity using several grid-level measures based on cellwise agreement between the predicted grid $\hat{G}$ and the ground-truth grid $G^*$ (Appendix **??**, for prompt, completion and calculation scenarios).

- **Exact Match ↑ (EM)**: Binary indicator of perfect reconstruction:

$$\text{EM} = \begin{cases} 1, & \text{if } \hat{G} = G^*, \\ 0, & \text{otherwise.} \end{cases}$$

- **Difference Ratio (DiffRatio)**: Fraction of incorrect cells normalized by total ground-truth mass:

$$\text{DiffRatio} = 1 - \frac{D(\hat{G}, G^*)}{\sum_{i,j} g_{ij}^*},$$

where $D(A, B)$ counts cellwise mismatches. *Higher is better* (1 indicates perfect match).
- **Relative Difference Ratio (RelDiffRatio)**: A softer variant that measures numeric deviation:

$$\text{RelDiffRatio} = 1 - \frac{D_{\text{rel}}(\hat{G}, G^*)}{\sum_{i,j} g_{ij}^*},$$

where $D_{\text{rel}}$ accumulates $|a_{ij} - b_{ij}|$ for numeric cells and counts categorical mismatches $(L, S, V)$ as 1. *Higher is better.*
- **Penalized Difference Ratio (PenDiffRatio)**: Penalty-weighted version increasing the cost of modifying or introducing new load, support, or void cells:

$$\text{PenDiffRatio} = 1 - \frac{D_{\text{pen}}(\hat{G}, G^*)}{\sum_{i,j} g_{ij}^*},$$

where $D_{\text{pen}}$ multiplies $L$, $S$, or $V$ cell errors by a penalty (typically $3\times$). *Higher is better.*
- **Difficulty-Weighted Difference Ratios**: Optional variants that multiply each cell's contribution by its local difficulty weight (see DWCS below). These versions emphasize correctness in structurally ambiguous or high-difficulty regions. *Higher is better.*

**Topological Validity Metrics** Beyond pixelwise accuracy, we evaluate the structural and connectivity properties of the reconstructed topology (Appendix D.3, for prompt, completion and calculation scenarios):

- **Grid Validity (ValidGrid)**: Boolean check ensuring $\hat{G}$ matches $G^*$ in shape and uses only admissible values $(L, S, \text{or } [0, 1])$. *True is desired.*
- **Load–Support Connectivity (LSConn)**: True if any load cell $(L)$ connects to any support $(S)$ through contiguous solid cells $(> 0, L, \text{or } S)$:

$$\text{LSConn} = \begin{cases} 1, & \exists \text{ load–support path through solids,} \\ 0, & \text{otherwise.} \end{cases}$$

*True is desired.*
- **Directional Load–Support Connectivity (DirLSConn)**: Same as LSConn, but restricted to force paths aligned with the gravity vector **g** inferred from dataset rotation metadata. *True is desired.*

- **Isolated Cluster Count** ($N_{\text{islands}}$): Number of solid-cell clusters disconnected from any load or support, found via 4-connectivity. *Lower is better.*
- **Difficulty Score (DWCS)**: Average difficulty weight for originally masked cells:

$$\text{DWCS} = \frac{1}{|\mathcal{V}|} \sum_{(i,j)\in\mathcal{V}} w_{ij}, \quad w_{ij} \in \{1,2,3\}.$$

Higher DWCS implies the reconstruction region is more complex or ambiguous; it reflects task difficulty rather than model quality. *Higher indicates harder samples.*

**Physics-Approximating Metrics**  To estimate the physical plausibility of predicted topologies, we approximate directional load-support efficiency using a force-path traversal cost. We calculate the average minimum directional cost for each load to reach a support, computed via a gravity-aligned Dijkstra traversal with angular and depth penalties. Unsupported loads receive a large but finite penalty (Appendix D.1 for EPCEff calculation details, and D.3, for prompt, completion and calculation scenarios).

- **Force Path Cost Average Efficiency Ratio (FPCEff)**: Relative efficiency of predicted vs. ground-truth structures:

$$\text{FPCEff} = \text{clip}_{[0,1]}\left(\frac{C^*_{\text{avg}}}{\hat{C}_{\text{avg}}}\right),$$

where $C^*_{\text{avg}}$ and $\hat{C}_{\text{avg}}$ are average load–support path costs in $G^*$ and $\hat{G}$ respectively. *Higher is better.*

| Category | Metric Name | Type / Range | Desired Trend |
|---|---|---|---|
| **Reconstruction** | Exact Match (EM) | Boolean {0,1} | True |
| | Difference Ratio (DiffRatio) | Float [0,1] | Higher is better |
| | Penalized Difference Ratio (PenDiffRatio) | Float [0,1] | Higher is better |
| | Relative Difference Ratio (RelDiffRatio) | Float [0,1] | Higher is better |
| | Difficulty-Weighted Diff. Ratio | Float [0,1] | Higher is better |
| | Difficulty-Weighted Rel. Diff. Ratio | Float [0,1] | Higher is better |
| **Topology** | Valid Grid (ValidGrid) | Boolean {0,1} | True |
| | Load–Support Connectivity (LSConn) | Boolean {0,1} | True |
| | Directional L–S Connectivity (DirLSConn) | Boolean {0,1} | True |
| | Isolated Clusters ($N_{\text{islands}}$) | Integer $\geq 0$ | Lower is better |
| | Difficulty Score (DWCS) | Float [1,3] avg. | Higher is harder |
| **Physics-Approx.** | Force Path Cost Efficiency (FPCEff) | Float [0,1] | Higher is better |

Table 2: Summary of all evaluation metrics by category, with their types, typical ranges, and optimization direction.

## 5.2 EXPERIMENTS

To establish baseline performance, we evaluate a broad set of contemporary language models in a zero-shot setting. From OpenAI, we include GPT-3.5 Brown et al. (2020), GPT-4.1 OpenAI et al. (2024a), and GPT-4o OpenAI et al. (2024b), representing successive generations with improved reasoning and multimodal capabilities. From Anthropic, we test Claude 3.7 Sonnet anthropic (2025a) and Claude Opus 4 anthropic (2025b), the strongest in the Claude family. From Google DeepMind, we include Gemini 1.5 Pro Team et al. (2024) and Gemini 2.5 Pro Comanici et al. (2025), designed for complex multimodal reasoning. We also assess DeepSeek-R1 DeepSeek-AI et al. (2025), an open-source model for scientific and engineering tasks, and Perplexity Sonar Team (2025a) and Sonar Reasoning Team (2025b), tuned for information-seeking and multi-step reasoning. Models are prompted (Appendix J) with structured descriptions of boundary conditions, forces, and supports, without simulation tools or external knowledge. A random subset of **100** examples spanning all task variations, difficulties and all models are evaluated under identical conditions via publicly available APIs (Table 3). Performance is measured using the metrics defined in Section 5.1.

# 6 RESULTS AND ANALYSIS

We present quantitative results in Table 3 and analyze failure modes qualitatively. Detailed results on few-shot prompting, rotation, and physics-enhanced and -neutral prompt design are discussed in the subsequent sections and further expanded in the Appendix.

Table 3: **Zero-Shot Performance on SPhyR 2D Tasks (Easy vs. Hard).** Top-performing LLMs (Claude, Gemini) maintain high Load-Support Connectivity, demonstrating core topological understanding. However, performance degrades sharply on Hard tasks, with negative Difference Ratios (red) confirming inefficient material hallucination and structural over-designing across all models. (↑ indicates better, ↓ indicates worse).

| Task | Metric | Easy | | | | | Hard | | | | |
|---|---|---|---|---|---|---|---|---|---|---|---|
| | | GPT 4.1 | Claude Opus 4 | Gemini 2.5 Pro | DeepSeek-R1 | Perplexity Sonar | GPT 4.1 | Claude Opus 4 | Gemini 2.5 Pro | DeepSeek-R1 | Perplexity Sonar |
| 1 Random Cell | Exact Match ↑ | 26 | **82** | 81 | 58 | 52 | 13 | **77** | 37 | 37 | 13 |
| | Difference Ratio (%) ↑ | 95.47 | **99.05** | 99.03 | 97.37 | 93.28 | 88.14 | 95.45 | **96.70** | 91.44 | 80.07 |
| | Relative Difference Ratio (%) ↑ | 95.47 | **99.05** | 99.03 | 97.37 | 93.28 | 96.05 | 96.72 | **97.88** | 96.51 | 88.98 |
| | Penalized Difference Ratio (%) ↑ | 94.82 | **98.40** | 98.37 | 96.71 | 92.03 | 92.44 | 93.11 | **94.31** | 92.90 | 85.12 |
| | Average Difficulty Score | **1.99** | **1.99** | **1.99** | **1.99** | **1.99** | **1.96** | **1.96** | 1.86 | **1.96** | **1.96** |
| | Difficulty Weighted Difference Ratio (%) ↑ | 63.31 | **65.51** | 65.47 | 64.54 | 61.92 | 58.05 | **60.97** | 58.87 | 59.41 | 52.28 |
| | Difficulty Weighted Relative Difference Ratio (%) ↑ | 63.31 | **65.51** | 65.47 | 64.54 | 61.92 | 62.49 | 62.25 | 60.01 | **62.63** | 58.26 |
| | Valid Output Grid ↑ | 100.00 | 100.00 | 100.00 | 100.00 | 96.00 | 100.00 | 99.00 | 100.00 | 100.00 | 94.00 |
| | Load-Support Connectivity (%) ↑ | 100.00 | 100.00 | 100.00 | 100.00 | 95.00 | 99.00 | 98.00 | 100.00 | 99.00 | 93.00 |
| | Load-Support Directional Connectivity (%) ↑ | 100.00 | 100.00 | 100.00 | 100.00 | 95.00 | 99.00 | 98.00 | 100.00 | 99.00 | 93.00 |
| | Average Isolated Clusters Count ↓ | 0.32 | 0.00 | 0.00 | 0.15 | 0.16 | **0.44** | 0.00 | 0.00 | 0.26 | 0.37 |
| | Force Path Cost Average Efficiency Ratio (%) ↑ | 99.94 | 99.94 | 99.94 | 99.86 | 94.84 | 98.93 | 97.91 | **99.92** | 98.93 | 92.93 |
| 5 Random Cells | Exact Match ↑ | 1 | **45** | 39 | 39 | 10 | 0 | **38** | 37 | 15 | 3 |
| | Difference Ratio (%) ↑ | 75.87 | **95.76** | 95.08 | 88.16 | 75.79 | 35.23 | **87.27** | 83.19 | 65.68 | 38.29 |
| | Relative Difference Ratio (%) ↑ | 75.87 | **95.76** | 95.08 | 88.16 | 75.79 | 75.68 | 89.98 | **91.58** | 86.53 | 70.37 |
| | Penalized Difference Ratio (%) ↑ | 69.37 | **89.26** | 88.59 | 81.66 | 68.84 | 61.72 | **78.42** | 75.62 | 71.73 | 57.71 |
| | Average Difficulty Score | **1.89** | **1.89** | **1.89** | **1.89** | **1.89** | **1.97** | **1.97** | 1.96 | **1.97** | **1.97** |
| | Difficulty Weighted Difference Ratio (%) ↑ | 48.77 | **59.89** | 59.48 | 55.56 | 47.57 | 23.41 | **56.17** | 52.60 | 41.52 | 24.87 |
| | Difficulty Weighted Relative Difference Ratio (%) ↑ | 48.77 | **59.89** | 59.48 | 55.56 | 47.57 | 49.57 | **59.65** | 57.87 | 56.19 | 45.99 |
| | Valid Output Grid ↑ | 100.00 | 100.00 | 100.00 | 100.00 | 87.00 | 99.00 | 100.00 | 99.00 | 100.00 | 85.00 |
| | Load-Support Connectivity (%) ↑ | 100.00 | 100.00 | 100.00 | 99.00 | 80.00 | 99.00 | 100.00 | 98.00 | 99.00 | 80.00 |
| | Load-Support Directional Connectivity (%) ↑ | 100.00 | 100.00 | 100.00 | 99.00 | 80.00 | 99.00 | 100.00 | 98.00 | 99.00 | 80.00 |
| | Average Isolated Clusters Count ↓ | 1.64 | 0.00 | 0.00 | 0.44 | 0.37 | **1.88** | 0.00 | 0.03 | 0.56 | 1.22 |
| | Force Path Cost Average Efficiency Ratio (%) ↑ | 99.72 | 99.70 | **99.75** | 97.50 | 78.42 | 98.55 | **99.49** | 97.84 | 98.25 | 79.48 |
| 10 Random Cells | Exact Match ↑ | 0 | **15** | 13 | 2 | 1 | 0 | **14** | 3 | 3 | 0 |
| | Difference Ratio (%) ↑ | 59.82 | **89.08** | 88.88 | 78.78 | 72.68 | -26.62 | **69.70** | 67.83 | 36.60 | -1.98 |
| | Relative Difference Ratio (%) ↑ | 59.82 | **89.08** | 88.88 | 78.78 | 72.68 | 51.37 | 79.91 | **82.80** | 74.22 | 42.43 |
| | Penalized Difference Ratio (%) ↑ | 47.02 | **76.41** | 76.21 | 65.95 | 59.85 | 20.54 | 49.33 | **57.50** | 43.59 | 23.37 |
| | Average Difficulty Score | **1.97** | **1.97** | **1.97** | **1.97** | **1.97** | **2.01** | **2.01** | 1.94 | **2.01** | **2.01** |
| | Difficulty Weighted Difference Ratio (%) ↑ | 40.24 | **58.23** | 58.06 | 51.72 | 47.81 | -17.38 | **45.17** | 41.99 | 22.14 | -0.67 |
| | Difficulty Weighted Relative Difference Ratio (%) ↑ | 40.24 | **58.23** | 58.06 | 51.72 | 47.81 | 34.45 | **52.88** | 52.66 | 48.86 | 29.03 |
| | Valid Output Grid ↑ | 99.00 | 100.00 | 100.00 | 100.00 | 92.00 | 99.00 | 99.00 | 100.00 | 99.00 | 64.00 |
| | Load-Support Connectivity (%) ↑ | 99.00 | 100.00 | 100.00 | 90.00 | 76.00 | 99.00 | 93.00 | 100.00 | 85.00 | 58.00 |
| | Load-Support Directional Connectivity (%) ↑ | 99.00 | 100.00 | 100.00 | 90.00 | 76.00 | 99.00 | 93.00 | 100.00 | 85.00 | 58.00 |
| | Average Isolated Clusters Count ↓ | 2.27 | 0.00 | 0.00 | 0.46 | 0.40 | 2.82 | 0.01 | 0.02 | 0.69 | 1.20 |
| | Force Path Cost Average Efficiency Ratio (%) ↑ | 98.28 | 99.06 | **99.31** | 88.72 | 73.62 | 98.13 | 91.98 | **99.34** | 83.93 | 56.25 |
| 1 Random Row | Exact Match ↑ | 20 | **52** | 44 | 14 | 21 | 2 | **49** | 46 | 34 | 7 |
| | Difference Ratio (%) ↑ | 84.50 | **94.92** | 93.90 | 73.09 | 77.87 | 18.44 | **80.55** | 71.69 | 73.13 | 27.07 |
| | Relative Difference Ratio (%) ↑ | 84.50 | **94.92** | 93.90 | 73.09 | 77.87 | 73.91 | **94.39** | 93.86 | 94.13 | 74.82 |
| | Penalized Difference Ratio (%) ↑ | 84.50 | **94.92** | 93.90 | 73.09 | 77.65 | 73.91 | **94.39** | 93.86 | 94.13 | 74.82 |
| | Average Difficulty Score | **1.94** | **1.94** | **1.94** | **1.94** | **1.94** | 1.92 | 1.92 | **1.99** | 1.92 | 1.92 |
| | Difficulty Weighted Difference Ratio (%) ↑ | 54.72 | **61.04** | 60.55 | 47.90 | 50.65 | 11.00 | **49.95** | 45.14 | 45.37 | 17.05 |
| | Difficulty Weighted Relative Difference Ratio (%) ↑ | 54.72 | **61.04** | 60.55 | 47.90 | 50.65 | 47.38 | 60.02 | **61.91** | 59.94 | 47.99 |
| | Valid Output Grid ↑ | 97.00 | 100.00 | 100.00 | 100.00 | 91.00 | 98.00 | 100.00 | 100.00 | 100.00 | 91.00 |
| | Load-Support Connectivity (%) ↑ | 97.00 | 100.00 | 100.00 | 97.00 | 73.00 | 97.00 | 99.00 | 100.00 | 86.00 | 85.00 |
| | Load-Support Directional Connectivity (%) ↑ | 97.00 | 100.00 | 100.00 | 97.00 | 73.00 | 97.00 | 99.00 | 100.00 | 86.00 | 85.00 |
| | Average Isolated Clusters Count ↓ | 0.02 | 0.00 | 0.00 | 0.00 | 0.07 | 0.01 | 0.00 | 0.00 | 0.00 | 0.05 |
| | Force Path Cost Average Efficiency Ratio (%) ↑ | 96.95 | 99.88 | **99.99** | 97.03 | 72.93 | 96.99 | 98.92 | **99.99** | 86.07 | 84.94 |
| 3 Random Rows | Exact Match ↑ | 6 | **35** | 29 | 20 | 24 | 0 | **21** | 12 | 17 | 1 |
| | Difference Ratio (%) ↑ | 59.31 | **88.64** | 84.09 | 62.64 | 74.23 | -162.36 | **32.01** | 18.75 | 16.98 | -106.35 |
| | Relative Difference Ratio (%) ↑ | 59.31 | **88.64** | 84.09 | 62.64 | 74.23 | 23.74 | **84.70** | 77.42 | 76.37 | 46.20 |
| | Penalized Difference Ratio (%) ↑ | 59.31 | **88.64** | 82.84 | 62.04 | 73.99 | 23.74 | **84.70** | 77.42 | 76.37 | 45.35 |
| | Average Difficulty Score | 1.89 | 1.89 | **1.92** | 1.89 | 1.89 | **1.99** | **1.99** | 1.99 | **1.99** | **1.99** |
| | Difficulty Weighted Difference Ratio (%) ↑ | 37.91 | **55.81** | 53.52 | 39.27 | 46.97 | -107.71 | **17.50** | 9.64 | 8.49 | -71.32 |
| | Difficulty Weighted Relative Difference Ratio (%) ↑ | 37.91 | **55.81** | 53.52 | 39.27 | 46.97 | 17.09 | **55.67** | 49.98 | 49.88 | 30.87 |
| | Valid Output Grid ↑ | 99.00 | 100.00 | 99.00 | 100.00 | 100.00 | 100.00 | 100.00 | 100.00 | 96.00 | 98.00 |
| | Load-Support Connectivity (%) ↑ | 95.00 | 100.00 | 98.00 | 69.00 | 74.00 | 100.00 | 97.00 | 100.00 | 61.00 | 96.00 |
| | Load-Support Directional Connectivity (%) ↑ | 95.00 | 100.00 | 98.00 | 69.00 | 74.00 | 100.00 | 97.00 | 100.00 | 61.00 | 96.00 |
| | Average Isolated Clusters Count ↓ | 0.28 | 0.00 | 0.01 | 0.27 | **0.28** | 0.03 | 0.00 | 0.00 | **0.38** | 0.01 |
| | Force Path Cost Average Efficiency Ratio (%) ↑ | 94.09 | **99.68** | 97.54 | 69.22 | 73.47 | 99.99 | 96.97 | 99.87 | 61.31 | 95.91 |
| 1 Random Column | Exact Match ↑ | 0 | **23** | 26 | 15 | 8 | 0 | 21 | **26** | 15 | 3 |
| | Difference Ratio (%) ↑ | 51.92 | **83.09** | 81.95 | 63.38 | 67.71 | -34.24 | 37.46 | **44.93** | 29.07 | -26.12 |
| | Relative Difference Ratio (%) ↑ | 51.92 | **83.09** | 81.95 | 63.38 | 67.71 | 29.24 | 62.68 | **72.53** | 60.88 | 36.26 |
| | Penalized Difference Ratio (%) ↑ | 42.23 | **73.51** | 70.55 | 53.69 | 55.52 | -3.14 | 31.65 | **46.17** | 27.89 | 4.24 |
| | Average Difficulty Score | **1.90** | **1.90** | 1.85 | **1.90** | **1.90** | **2.13** | **2.13** | 1.87 | **2.13** | **2.13** |
| | Difficulty Weighted Difference Ratio (%) ↑ | 36.02 | **50.85** | 48.21 | 40.90 | 44.12 | -23.21 | 14.68 | **15.91** | 10.73 | -17.01 |
| | Difficulty Weighted Relative Difference Ratio (%) ↑ | 36.02 | **50.85** | 48.21 | 40.90 | 44.12 | 22.54 | 37.62 | **40.27** | 38.36 | 26.35 |
| | Valid Output Grid ↑ | 100.00 | 100.00 | 98.00 | 98.00 | 97.00 | 97.00 | 97.00 | 100.00 | 97.00 | 91.00 |
| | Load-Support Connectivity (%) ↑ | 100.00 | 100.00 | 98.00 | 98.00 | 88.00 | 97.00 | 97.00 | 100.00 | 97.00 | 90.00 |
| | Load-Support Directional Connectivity (%) ↑ | 100.00 | 100.00 | 98.00 | 98.00 | 88.00 | 97.00 | 97.00 | 100.00 | 97.00 | 90.00 |
| | Average Isolated Clusters Count ↓ | 0.13 | 0.00 | 0.00 | 0.00 | 0.06 | 0.07 | 0.00 | 0.00 | 0.01 | 0.04 |
| | Force Path Cost Average Efficiency Ratio (%) ↑ | 99.55 | 94.90 | 97.30 | 97.61 | 73.45 | 96.31 | 84.17 | **99.48** | 93.47 | 88.37 |
| 3 Random Columns | Exact Match ↑ | 1 | **5** | 3 | 2 | 1 | 0 | **7** | 3 | 5 | 1 |
| | Difference Ratio (%) ↑ | 2.46 | **58.69** | 52.03 | 18.34 | 24.01 | -238.99 | **-17.63** | -56.28 | -22.76 | -142.17 |
| | Relative Difference Ratio (%) ↑ | 2.46 | **58.69** | 52.03 | 18.34 | 24.01 | -35.52 | **31.78** | 20.55 | 22.87 | -18.10 |
| | Penalized Difference Ratio (%) ↑ | -28.47 | **27.96** | 17.06 | -12.54 | -9.25 | -108.99 | **-43.98** | -60.21 | -49.47 | -109.09 |
| | Average Difficulty Score | 1.88 | 1.88 | **1.90** | 1.88 | 1.88 | 1.90 | 1.90 | **1.96** | 1.90 | 1.90 |
| | Difficulty Weighted Difference Ratio (%) ↑ | 1.69 | **34.52** | 31.14 | 12.74 | 15.59 | -153.59 | **-22.99** | -50.66 | -24.21 | -94.09 |
| | Difficulty Weighted Relative Difference Ratio (%) ↑ | 1.69 | **34.52** | 31.14 | 12.74 | 15.59 | -25.00 | **14.44** | 7.72 | 10.10 | -12.75 |
| | Valid Output Grid ↑ | 99.00 | 100.00 | 98.00 | 100.00 | 93.00 | 94.00 | 100.00 | 88.00 | 80.00 | |
| | Load-Support Connectivity (%) ↑ | 97.00 | 94.00 | 96.00 | 93.00 | 70.00 | 94.00 | 93.00 | **96.00** | 71.00 | 61.00 |
| | Load-Support Directional Connectivity (%) ↑ | 97.00 | 94.00 | 96.00 | 93.00 | 70.00 | 94.00 | 93.00 | **96.00** | 71.00 | 61.00 |
| | Average Isolated Clusters Count ↓ | 0.32 | 0.00 | 0.02 | 0.12 | 0.24 | 0.30 | 0.00 | 0.02 | 0.01 | 0.32 |
| | Force Path Cost Average Efficiency Ratio (%) ↑ | 94.43 | 88.28 | 92.30 | 86.51 | 54.44 | 92.95 | 77.05 | **94.28** | 54.47 | 58.37 |
| Full | Exact Match ↑ | 0 | 0 | 0 | 0 | 0 | 0 | 0 | 0 | 0 | 0 |
| | Difference Ratio (%) ↑ | -62.06 | **-35.78** | -25.03 | -126.02 | -49.16 | -816.91 | **-466.42** | -548.98 | -585.87 | -537.83 |
| | Relative Difference Ratio (%) ↑ | -62.06 | **-35.78** | -25.03 | -126.02 | -49.16 | -251.75 | -177.48 | -316.57 | -162.59 | **-144.70** |
| | Penalized Difference Ratio (%) ↑ | -62.06 | **-35.78** | -25.96 | -142.78 | -49.86 | -251.75 | -177.48 | -318.95 | -178.14 | -151.62 |
| | Average Difficulty Score | 1.93 | 1.92 | 1.91 | 1.93 | 1.93 | 1.95 | **1.96** | 1.95 | 1.95 | 1.95 |
| | Difficulty Weighted Difference Ratio (%) ↑ | -37.69 | -21.79 | **-14.61** | -79.62 | -29.78 | -530.22 | -310.26 | -360.89 | -382.38 | -348.11 |
| | Difficulty Weighted Relative Difference Ratio (%) ↑ | -37.69 | -21.79 | **-14.61** | -79.62 | -29.78 | -162.49 | -117.17 | -208.21 | -105.80 | **-92.41** |
| | Valid Output Grid ↑ | 100.00 | 100.00 | 99.00 | 100.00 | 81.00 | 100.00 | 100.00 | 99.00 | 100.00 | 76.00 |
| | Load-Support Connectivity (%) ↑ | 94.00 | 94.00 | 98.00 | 48.00 | 42.00 | 100.00 | 88.00 | 98.00 | 68.00 | 49.00 |
| | Load-Support Directional Connectivity (%) ↑ | 94.00 | 94.00 | 98.00 | 48.00 | 42.00 | 100.00 | 88.00 | 98.00 | 68.00 | 49.00 |
| | Average Isolated Clusters Count ↓ | 0.01 | 0.00 | 0.01 | 0.06 | **0.09** | 0.00 | 0.00 | 0.01 | 0.06 | **0.08** |
| | Force Path Cost Average Efficiency Ratio (%) ↑ | 88.87 | 90.33 | **96.85** | 48.31 | 37.61 | 99.86 | 87.83 | 97.83 | 68.32 | 48.89 |
| **Average** | Exact Match ↑ | 6.75 | **32.12** | 29.38 | 14.88 | 15.12 | 1.88 | **28.25** | 26.75 | 15.75 | 3.50 |
| | Difference Ratio (%) ↑ | 45.91 | **71.68** | 71.24 | 44.47 | 54.55 | -142.16 | **-10.20** | -27.77 | -36.97 | -83.63 |
| | Relative Difference Ratio (%) ↑ | 45.91 | **71.68** | 71.24 | 44.47 | 54.55 | 7.84 | **45.53** | 27.31 | 43.61 | 24.53 |
| | Penalized Difference Ratio (%) ↑ | 38.34 | **64.16** | 62.70 | 34.73 | 46.10 | -11.44 | **26.27** | 8.21 | 22.38 | 3.74 |
| | Average Difficulty Score | 1.92 | 1.92 | **1.92** | 1.92 | 1.92 | 1.98 | **1.98** | 1.94 | 1.98 | 1.98 |
| | Difficulty Weighted Difference Ratio (%) ↑ | 30.62 | **45.51** | 45.23 | 29.12 | 35.61 | -92.46 | **-11.10** | -23.42 | -27.37 | -54.63 |
| | Difficulty Weighted Relative Difference Ratio (%) ↑ | 30.62 | **45.51** | 45.23 | 29.12 | 35.61 | 5.75 | **28.17** | 15.28 | 27.52 | 16.67 |
| | Valid Output Grid ↑ | 99.25 | 100.00 | 99.25 | 100.00 | 92.12 | 98.38 | 99.62 | **99.75** | 97.50 | 84.88 |
| | Load-Support Connectivity (%) ↑ | 97.75 | 98.50 | 98.75 | 86.75 | 74.75 | 98.12 | 95.62 | **99.00** | 83.25 | 76.50 |
| | Load-Support Directional Connectivity (%) ↑ | 97.75 | 98.50 | **98.75** | 86.75 | 74.75 | 98.12 | 95.62 | **99.00** | 83.25 | 76.50 |
| | Average Isolated Clusters Count ↓ | 0.59 | 0.00 | 0.01 | 0.20 | 0.21 | **0.69** | 0.00 | 0.01 | 0.25 | 0.41 |
| | Force Path Cost Average Efficiency Ratio (%) ↑ | 96.48 | 96.47 | **97.87** | 85.59 | 69.85 | 97.71 | 91.79 | **98.57** | 80.59 | 75.64 |

## 6.1 QUANTITATIVE RESULTS

**General Performance Trends**   Table 3 presents model performance across all task variations using the metrics defined in Section 5.1. As expected, performance degrades as task complexity increases; "Easy" (binary) tasks consistently yield higher accuracy than "Hard" (continuous) variants. Top-performing models like Claude Opus 4 and Gemini 2.5 Pro achieve near-perfect Load-Support Connectivity (>98%) and Valid Grid scores on easy tasks, suggesting that while they fail to replicate the exact ground-truth geometry (low Exact Match), they successfully reason about global structural integrity and force propagation (Appendix F, for additional plots).

**The Hard Task Anomaly and Material Hallucination**   A critical observation in the hard (continuous) tasks is the prevalence of negative Difference Ratios across almost all models. Physically, this result implies significant over-designing: rather than converging on efficient load paths, models tend to "smear" material across the void, producing dense, non-optimal clusters. This hallucination of mass suggests that while models grasp the concept of filling space, they lack the physical intuition to minimize volume while maintaining stability, a core tenet of topology optimization.

**DeepSeek-R1 and the Limits of Chain-of-Thought**   Notably, DeepSeek-R1, a model optimized for reasoning, exhibits a strong performance drop between easy and hard tasks. While it maintains reasonable connectivity on binary tasks, its performance collapses on continuous distributions (Table 3). We hypothesize that the model's Chain-of-Thought (CoT) process struggles to ground floating-point grid representations into spatial intuition. Instead of visualizing the physical load path, the model likely attempts arithmetic or symbolic manipulation of the density values. This symbolic approach fails to capture the global topological constraints required for stability, resulting in outputs that are computationally "reasoned" but structurally incoherent.

**Rotation Experiments and Gravity Bias**   Among localized tasks, row completions consistently outperform column completions. Our rotation experiments ($k = 3, 270°$) reveal that this is not merely a formatting artifact but a "gravity bias." When loads are applied horizontally (simulating a cantilever or rotated structure), models frequently fail to reorient their structural intuition, attempting to build "downward" relative to the grid rather than in the direction of the force vector $L$. This indicates that models rely heavily on memorized visual patterns of vertical buildings rather than reasoning about the directed vector of applied forces (Appendix G, for additional rotation experiment results).

**Few-Shot Experiments**   To investigate the in-context learning capabilities of the models, we performed few-shot experiments complementary to the zero-shot baseline. In this setting, we prepended $k = 1$ and $k = 3$ randomly selected input-output pairs from the dataset to the prompt before presenting the target test instance. The examples were drawn from the same task variation (e.g., 3 Random Row) and difficulty level (easy or hard) as the query. This approach evaluates whether models can improve their spatial reasoning and output formatting by observing valid load-path distributions, thereby allowing us to quantify the extent to which physical constraints can be inferred from examples versus explicit instructions (Appendix H, for additional few-shot experiment results).

**Physics-Enhanced vs. Physics-Neutral Prompts**   Counter-intuitively, our prompt ablation studies reveal that physics-enhanced prompts, those augmented with terminology like "stress," "load path," and "equilibrium", actually degraded performance on harder tasks compared to the base prompt. While the Physics-Neutral setting suffered in connectivity metrics, the failure of the Enhanced prompt suggests that models do not ground physical jargon to the visual grid. Instead, terms like "stress" likely act as semantic distractors, shifting the model's focus away from the necessary spatial pattern-matching and leading to worse topological validity (Appendix I, for details).

## 6.2 QUALITATIVE ANALYSIS OF FAILURE MODES

To complement the quantitative metrics, we visually inspected model predictions to identify recurring patterns of reasoning failure. We observed three distinct failure modes that explain the performance gaps reported in Table 3.

**The "Smearing" Effect in Continuous Tasks**    In hard (continuous) tasks, models frequently fail to commit to a defined structure. Instead of placing high-density material in critical load paths, they distribute low-density values $(0.1 - 0.3)$ broadly across the void (Appendix J.15). This "smearing" behavior results in the negative Difference Ratios observed quantitatively; the models appear to be minimizing risk by filling space rather than optimizing for stiffness, effectively hallucinating material where none is needed.

**Disconnected Islands and Local Bias**    A common error in lower-performing models (e.g., Perplexity Sonar, DeepSeek-R1 on hard tasks) is the generation of "floating islands", clusters of material completely disconnected from supports. This confirms: these models are operating primarily on local pattern consistency (placing a 1 next to another 1) rather than global constraint satisfaction. They fail to trace the load path $L \to S$ back to a fixed point, violating fundamental equilibrium principles.

**Gravity Bias in Rotated Scenarios**    Qualitative inspection of the rotated experiments $(270°)$ reveals a strong directional bias. Even when the load $L$ is applied horizontally from the left, models often attempt to build "downward" relative to the grid layout, ignoring the rotated force vector. This results in structures that "hang" into empty space or connect to non-existent supports at the bottom of the grid, providing strong evidence that the models are relying on visual memorization of vertical architectural forms rather than physical reasoning.

**Over-Constrained "Safety"**    Conversely, top-performing models like Gemini 2.5 Pro often "over-build," creating blocky, wall-like structures rather than truss-like efficient designs. While this strategy achieves high Load-Support Connectivity (resulting in high success rates), it fails the efficiency objective of topology optimization, treating the task as a "fill-the-gap" segmentation problem rather than a minimum-compliance optimization problem.

## 7    DISCUSSION

The quantitative and qualitative results highlight fundamental gaps between linguistic reasoning and physical-spatial understanding in Large Language Models.

**Lack of Grounded Physical Understanding**    The failure of physics-enhanced prompts and the struggle with hard tasks suggest that current LLMs do not possess a grounded model of physics. When a model reads "load path", it does not translate this into a constraint satisfaction problem on the grid; it treats it as a textual token associated with general engineering contexts. Consequently, models perform best when the task is framed as a visual pattern completion (base prompt) rather than a physics simulation problem.

**Visual Memorization vs. Force Reasoning**    The "gravity bias" observed in our rotation experiments confirms that models are solving SPhyR tasks primarily through visual memorization of architectural forms (e.g., columns support beams from below) rather than first-principles reasoning about force vectors. When the "floor" is moved to the "wall" (rotated setup), the model's heuristic fails, proving that it is not tracing the force $L$ to the support $S$, but rather completing a learned image schema.

**The Challenge of Continuous Optimization**    The "smearing" effect and negative Difference Ratios in continuous tasks highlight a specific deficiency in LLM spatial reasoning: the inability to perform gradient-like optimization. While models can predict discrete binary occupancy (material vs. void) based on connectivity rules, they cannot intuitively minimize compliance or volume in a continuous space. This remains a significant barrier for using LLMs in generative design and engineering applications where efficiency is paramount.

## 8    CONCLUSION

SPhyR reveals that while LLMs exhibit strong general reasoning, they fail to integrate spatial layout with grounded physical constraints. Observed failure modes (e.g., gravity bias, material smearing) confirm reliance on visual pattern matching over global force-directed reasoning, necessitating future work on geometric constraint satisfaction.

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

## A  APPENDIX

## B  TOPOLOGY OPTIMIZATION SOLVER PARAMETERS: GRASSHOPPER MILLIPEDE

**Solver parameters are:**

target density = 0.1
self-weight = 0
iterations = 10
smoothing = 0.1
penalization = 3.0
minimum density = 0.001
delete threshold = 0.5
compliant mechanism disabled

## C   VISUAL TASK VARIATIONS OVERVIEW

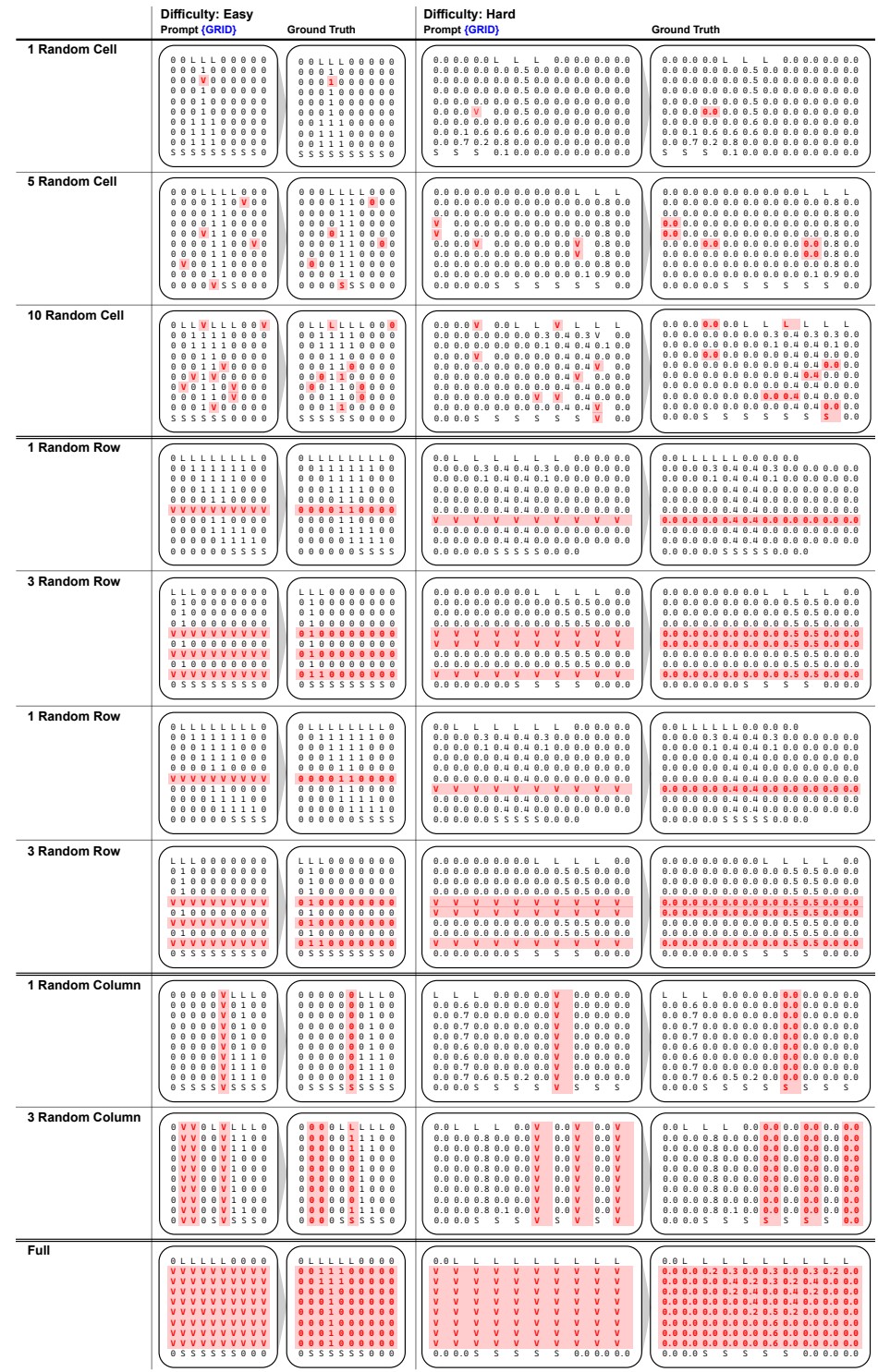

Figure 3: Overview of task variations: predicting material distributions for N random cells, rows, columns, or full structures for easy (binary) and hard (continous) difficulties.

# D  ADDITIONAL EVALUATION METRICS COMPUTATION AND PROMPT AND COMPLETION EXAMPLES

## D.1  FORCE-PATH COST COMPUTATION

To approximate the physical efficiency of load transmission through the predicted topology, we define a gravity-aligned cost metric that measures the minimum traversal effort for any load cell to reach a support cell through contiguous solid material.

Each grid cell $g_{ij}$ can take values in $\{L, S\} \cup [0, 1]$, where $L$ and $S$ denote applied load and support, respectively, and real-valued entries represent material density. We assume a fixed gravity direction $\mathbf{g} = (d_r, d_c) \in \{(1, 0), (0, 1), (-1, 0), (0, -1)\}$.

**Directional neighborhood.** We consider all 8-connected neighbors of $(i, j)$,

$$\mathcal{N}(i, j) = \{(i', j') \mid (i' - i, j' - j) \in \{(\pm 1, 0), (0, \pm 1), (\pm 1, \pm 1)\}\},$$

with direction vector $\mathbf{d} = (i' - i, j' - j)$. Each neighbor is assigned a traversal cost $w_{\mathbf{d}}$ based on its angular deviation from gravity:

$$w_{\mathbf{d}} = \begin{cases} 1.0, & \angle(\mathbf{g}, \mathbf{d}) < 15°, \\ 1.2, & 15° \leq \angle < 45°, \\ 1.5, & 45° \leq \angle < 100°, \\ 3.0, & \text{otherwise.} \end{cases}$$

Upward (against-gravity) moves are disallowed whenever $\mathbf{d} \cdot \mathbf{g} < -0.5$, ensuring that load flow occurs only downward or laterally.

**Shortest-path computation.** For each load cell $\ell = (i_\ell, j_\ell)$, we compute the minimal cost to any support $s \in S$ using Dijkstra's algorithm over the graph of solid nodes $\{(i, j) \mid g_{ij} > 0\}$. The cumulative path cost is defined as

$$C(\ell) = \min_{p \in P_{\ell \to S}} \sum_{((i,j),(i',j')) \in p} w_{(i'-i),(j'-j)} \left(1 + 0.05 \left|i' - i_\ell\right|\right),$$

where the multiplicative term $1 + 0.05 \left|i' - i_\ell\right|$ imposes a mild depth penalty to discourage long vertical travel from the load origin. If no valid support is reachable, a finite penalty $C_{\max}$ is assigned.

The mean force-path cost for a grid $G$ is

$$\overline{C}(G) = \frac{1}{N_L} \sum_{\ell \in L} C(\ell), \qquad C(\ell) = C_{\max} \text{ if unsupported.}$$

**Force-Path Cost Average Efficiency Ratio.** We define the final metric as

$$\text{FPCEff} = \text{clip}_{[0,1]} \left( \frac{\overline{C}(G^*)}{\overline{C}(\hat{G})} \right),$$

where $G^*$ and $\hat{G}$ denote the ground-truth and predicted grids, respectively. Higher values indicate that the predicted structure achieves comparable or better load–support transmission efficiency than the reference.

## D.2 RECONSTRUCTION METRIC TESTS

### D.2.1 EXACT MATCH EXAMPLES

---
**Exact Match Examples**

```
This test validates the get_exact_match function, which returns True
if the predicted grid Ĝ exactly matches the ground truth G*
cell-by-cell.
1.  Perfect match (True)
Ground truth:  [ 0 , L , 0 ],
               [ 0 , 1 , 0 ],
               [ 0 , S , 0 ]
Prediction:    [ 0 , L , 0 ],
               [ 0 , 1 , 0 ],
               [ 0 , S , 0 ]
Expected output:  True
2.  Slight difference (False)
Ground truth:  [ 0 , L , 0 ],
               [ 0 , 1 , 0 ],
               [ 0 , S , 0 ]
Prediction:    [ 1 , L , 0 ],
               [ 1 , 1 , 0 ],
               [ 0 , S , 0 ]
Expected output:  False
```
---

### D.2.2 DIFFERENCE RATIO EXAMPLES

---
**Difference Ratio Examples**

```
This test validates get_difference_ratio, which measures similarity
between Ĝ and G*.
A value of 1.0 means perfect reconstruction, while lower values
indicate greater deviation.
1.  Perfect match (1.000)
Ground truth:  [ 0 , L , 0 ],
               [ 0 , 1 , 0 ],
               [ 0 , S , 0 ]
Prediction:    [ 0 , L , 0 ],
               [ 0 , 1 , 0 ],
               [ 0 , S , 0 ]
Expected output:  1.000
2.  One altered column (0.000)
Ground truth:  [ 0 , L , 0 ],
               [ 0 , 1 , 0 ],
               [ 0 , S , 0 ]
Prediction:    [ 1 , L , 0 ],
               [ 1 , 1 , 0 ],
               [ 0 , S , 0 ]
Expected output:  0.000
3.  Half correct (0.500)
Ground truth:  [ 0 , L , 0 ],
               [ 1 , 1 , 0 ],
               [ 0 , S , 0 ]
Prediction:    [ 0 , L , 0 ],
               [ 0 , 1 , 0 ],
               [ 0 , S , 0 ]
Expected output:  0.500
```
---

### D.2.3 RELATIVE AND PENALIZED DIFFERENCE RATIO EXAMPLES

---

**Relative and Penalized Difference Ratio Examples**

```
These tests validate get_relative_difference_ratio and
get_penalized_difference_ratio, which account for numeric
cell differences and penalize fixed-cell deviations respectively.
1.  Perfect alignment (1.000)
Ground truth:  [ 0 , L , 0 ],
               [ 0 , 1 , 0 ],
               [ 0 , S , 0 ]
Prediction:    [ 0 , L , 0 ],
               [ 0 , 1 , 0 ],
               [ 0 , S , 0 ]
Expected output:  1.000
2.  Gradual deviation (0.333)
Ground truth:  [ 0 , L , 0 ],
               [ 1 , 1 , 1 ],
               [ 0 , S , 0 ]
Prediction:    [ 0 , L , 0 ],
               [ 0 , 1 , 0 ],
               [ 0 , S , 0 ]
Expected output:  0.333
3.  Continuous values (0.500)
Ground truth:  [ 0 , L , 0 ],
               [ 0.8 , 1 , 0.8 ],
               [ 0 , S , 0 ]
Prediction:    [ 0 , L , 0 ],
               [ 0.4 , 0.5 , 0.4 ],
               [ 0 , S , 0 ]
Expected output:  0.500
4.  Over-extrapolation (0.308)
Ground truth:  [ 0 , L , 0 ],
               [ 0.8 , 1 , 0.8 ],
               [ 0 , S , 0 ]
Prediction:    [ 0 , L , 0 ],
               [ 0.4 , 2.0 , 0.4 ],
               [ 0 , S , 0 ]
Expected output:  0.308
5.  Negative ratio (-1.000 or -2.000)
Ground truth:  [ 0 , L , 0 ],
               [ 0 , 1 , 0 ],
               [ 0 , S , 0 ]
Prediction:    [ 0 , 1 , 0 ],
               [ 1 , 1 , 1 ],
               [ 0 , S , 0 ]
Expected output:  -1.000 (unpenalized), -2.000 (penalized)
```

**Interpretation:**
The ratios decrease as predictions deviate numerically from the ground truth,
and penalized variants further reduce the score when fixed regions
(load or support)
are incorrectly modified.  Scores near or below 0 reflect large or structurally
meaningful errors.

---

## D.3 TOPOLOGY METRIC

### D.3.1 GRID VALIDITY EXAMPLES

---

**Grid Validity Examples**

This test validates the get_grid_shape_and_value_validity function,
which ensures that a generated grid has valid symbols and a consistent
rectangular shape.
A valid grid:
  • Uses only the symbols {0, 1, L, S};
  • Contains values within allowed numeric bounds;
  • Has equal row lengths (rectangular shape).
**1.  Valid grid**
Completion:     [ 0 , **L** , 0 ],
                [ 0 , **1** , 0 ],
                [ 0 , **S** , 0 ]
Expected:  True
(All symbols valid, shape consistent.)
**2.  Invalid character (X)**
Completion:     [ 0 , X , 0 ],
                [ 0 , **1** , 0 ],
                [ 0 , **S** , 0 ]
Expected:  False
(Unrecognized symbol X.)
**3.  Invalid character (P)**
Completion:     [ 0 , **L** , 0 ],
                [ 0 , **1** , 0 ],
                [ 0 , P , 0 ]
Expected:  False
(Unrecognized symbol P.)
**4.  Out-of-range value (-1)**
Completion:     [ 0 , **L** , 0 ],
                [ 0 , -1 , 0 ],
                [ 0 , **S** , 0 ]
Expected:  False
(Negative numeric value not allowed.)
**5.  Out-of-range value (2)**
Completion:     [ 0 , **L** , 0 ],
                [ 0 , 2 , 0 ],
                [ 0 , **S** , 0 ]
Expected:  False
(Value exceeds permitted range.)
**6.  Non-rectangular grid**
Completion:     [ 0 , **L** , 0 ],
                [ 0 , 1 ],
                [ 0 , **S** , 0 ]
Expected:  False
(Inconsistent row lengths.)
**Interpretation:**
This check ensures that downstream metrics operate on well-formed
grids only.
Any invalid symbol, numeric range violation, or non-rectangular
structure
results in a False validity flag.

---

### D.3.2    LOAD–SUPPORT CONNECTIVITY EXAMPLES

**Load–Support Connectivity Examples**

```
These tests validate is_load_supported and
is_load_supported_force_directional, which determine whether
loads (L) are connected to supports (S) through solid
cells (1, L, S). The directional variant allows
only gravity-aligned or lateral connections.
1.  Perfect vertical connection
Completion:    [ 0 , L , 0 ],
               [ 0 , 1 , 0 ],
               [ 0 , S , 0 ]
Expected:  True (both directional & non-directional)
2.  Diagonal bridge
Completion:    [ 0 , L , 0 ],
               [ 1 , 1 , 0 ],
               [ 0 , S , S ]
Expected:  True (connected diagonally)
3.  Horizontal load alignment
Completion:    [ 0 , L , L ],
               [ 0 , 1 , 0 ],
               [ 0 , S , 0 ]
Expected:  True (non-directional)
4.  Incomplete bridge
Completion:    [ 0 , L , L ],
               [ 1 , 0 , 0 ],
               [ 0 , S , 0 ]
Expected:  True (non-directional), False (directional)
5.  Disconnected load
Completion:    [ 0 , 0 , L ],
               [ 1 , 0 , 0 ],
               [ 0 , S , 0 ]
Expected:  False (no path)
6.  Complex multi-load structure
Completion:    [ 1 , 1 , 1 ,  0 , 1 , L ],
               [ 1 , 0 , 1 ,  0 , 1 , 0 ],
               [ 1 , 0 , 1 ,  1 , 1 , 0 ],
               [ 1 , 0 , 0 ,  0 , 0 , 0 ],
               [ S , 0 , 0 ,  0 , 0 , 0 ]
Expected:  True (non-directional), False (directional)
Interpretation:
The directional test approximates gravity-aligned force flow, while
the
non-directional variant checks only geometric reachability.
Disconnected or upward-only paths yield False.
```

### D.3.3 ISOLATED CLUSTER COUNT EXAMPLES

---

**Isolated Cluster Count Examples**

This test validates get_isolated_clusters_count, which counts
solid regions (1) disconnected from any load (L) or
support (S). A higher count indicates fragmented or non-functional
material regions.

**1.  Single isolated column (1)**
```
Completion:     [ L , 0 , 0 ,  0 , 0 , 0 ],
                [ 1 , 0 , 0 ,  0 , 0 , 0 ],
                [ 1 , 0 , 1 ,  0 , 0 , 0 ],
                [ 1 , 0 , 0 ,  0 , 0 , 0 ],
                [ 1 , 0 , 0 ,  0 , 0 , 0 ],
                [ S , 0 , 0 ,  0 , 0 , 0 ]
```
Expected:  1

**2.  Slightly connected cluster (1)**
```
Completion:     [ L , 0 , 0 ,  0 , 0 , 0 ],
                [ 1 , 0 , 0 ,  0 , 0 , 0 ],
                [ 1 , 0 , 1 ,  1 , 0 , 0 ],
                [ 1 , 0 , 0 ,  1 , 0 , 0 ],
                [ 1 , 0 , 0 ,  0 , 0 , 0 ],
                [ S , 0 , 0 ,  0 , 0 , 0 ]
```
Expected:  1

**3.  Two isolated clusters (2)**
```
Completion:     [ L , 0 , 0 ,  0 , 0 , 1 ],
                [ 1 , 0 , 0 ,  0 , 0 , 1 ],
                [ 1 , 0 , 1 ,  1 , 0 , 1 ],
                [ 1 , 0 , 0 ,  1 , 0 , 0 ],
                [ 1 , 0 , 0 ,  0 , 0 , 0 ],
                [ S , 0 , 0 ,  0 , 0 , 0 ]
```
Expected:  2

**4.  Multiple detached clusters (3)**
```
Completion:     [ L , 0 , 0 ,  1 , 0 , 0 ],
                [ 1 , 0 , 0 ,  0 , 0 , 0 ],
                [ 1 , 0 , 1 ,  1 , 0 , 0 ],
                [ 1 , 0 , 0 ,  1 , 0 , 1 ],
                [ 1 , 0 , 0 ,  0 , 0 , 0 ],
                [ S , 0 , 0 ,  0 , 0 , 0 ]
```
Expected:  3

**Interpretation:**
Isolated clusters represent solid "islands" that do not participate in
load-support transfer.  Lower counts indicate more integrated and
structurally valid predictions.

---

### D.3.4 DIFFICULTY SCORE (DWCS) EXAMPLES

---

**Difficulty Score (DWCS) Examples**

```
This test validates get_difficulty_score, which computes the
average difficulty of masked (V) cells in the input grid
based on their ground-truth neighborhood configuration in the GT grid.
The completion grid is used to confirm the reconstruction context.
Higher scores correspond to more complex masked regions.
```

**1. Simple vertical case (2.0)**

```
Input:          [ 0 , L , 0 ],
                [ 0 , V , 0 ],
                [ 0 , S , 0 ]
Ground truth:   [ 0 , L , 0 ],
                [ 0 , 1 , 0 ],
                [ 0 , S , 0 ]
Completion:     [ 0 , L , 0 ],
                [ 0 , 1 , 0 ],
                [ 0 , S , 0 ]
Expected output:  2.000
```

**2. Mixed neighborhood (3.0)**

```
Input:          [ 0 , L , 0 ],
                [ V , 1 , 0 ],
                [ 0 , S , 0 ]
Ground truth:   [ 0 , L , 0 ],
                [ 0 , 1 , 0 ],
                [ 0 , S , 0 ]
Completion:     [ 0 , L , 0 ],
                [ 1 , 1 , 0 ],
                [ 0 , S , 0 ]
Expected output:  3.000
```

**3. Large structure (1.0)**

```
Input:          [ 0 , L , 0 ],
                [ 0 , 1 , 0 ],
                [ 0 , 1 , V ],
                [ 0 , 1 , 0 ],
                [ 0 , S , 0 ]
Ground truth:   [ 0 , L , 0 ],
                [ 0 , 1 , 0 ],
                [ 0 , 1 , 0 ],
                [ 0 , 1 , 0 ],
                [ 0 , S , 0 ]
Completion:     [ 0 , L , 0 ],
                [ 0 , 1 , 0 ],
                [ 0 , 1 , 0 ],
                [ 0 , 1 , 0 ],
                [ 0 , S , 0 ]
Expected output:  1.000
```

**4. Dense structure with boundary void (3.0)**

```
Input:          [ 0 , L , L , L , 0 ],
                [ 0 , 1 , 1 , 1 , 0 ],
                [ 0 , 1 , V , 1 , 0 ],
                [ 0 , 1 , 1 , 1 , 0 ],
                [ 0 , S , S , S , 0 ]
Ground truth:   [ 0 , L , L , L , 0 ],
                [ 0 , 1 , 1 , 0 , 0 ],
                [ 0 , 1 , 0 , 1 , 0 ],
                [ 0 , 1 , 1 , 0 , 0 ],
                [ 0 , S , S , S , 0 ]
Completion:     [ 0 , L , L , L , 0 ],
                [ 0 , 1 , 1 , 0 , 0 ],
                [ 0 , 1 , 0 , 1 , 0 ],
                [ 0 , 1 , 1 , 0 , 0 ],
                [ 0 , S , S , S , 0 ]
```

```
Expected output:  3.000
Interpretation:
The score increases when masked cells (V) occur in ambiguous or mixed
regions, particularly around structural boundaries.  Uniform
neighborhoods yield
lower scores, reflecting easier reconstruction.
```

### D.4 PHYSICS APPROXIMATION METRIC

### D.4.1 FORCE PATH COST EXAMPLES

```
Force Path Cost Examples

This test validates the
get_total_force_path_cost_average_efficiency_ratio function, which
computes the Force Path Cost Average Efficiency Ratio (FPCEff).
Higher ratios indicate more efficient and physically plausible
load-support paths aligned with gravity.
Gravity direction:  (1, 0) downward
Test cases:
1.  Perfect vertical alignment
Ground truth:  [ 0 , L , 0 ],
               [ 0 , 1 , 0 ],
               [ 0 , S , 0 ]
Prediction:    [ 0 , L , 0 ],
               [ 0 , 1 , 0 ],
               [ 0 , S , 0 ]
Expected output:  1.000
2.  Slightly wider vertical column (still efficient)
Ground truth:  [ 0 , L , 0 ],
               [ 0 , 1 , 0 ],
               [ 0 , S , 0 ]
Prediction:    [ 0 , L , 0 ],
               [ 1 , 1 , 0 ],
               [ 0 , S , 0 ]
Expected output:  1.000
3.  Offset load-support connection (less efficient)
Ground truth:  [ 0 , L , 0 ],
               [ 0 , 1 , 0 ],
               [ S , 0 , 0 ]
Prediction:    [ 0 , L , 0 ],
               [ 1 , 1 , 0 ],
               [ S , 0 , 0 ]
Expected output:  0.8037
4.  Broken vertical link (similar inefficiency)
Ground truth:  [ 0 , L , 0 ],
               [ 0 , 1 , 0 ],
               [ S , 0 , 0 ]
Prediction:    [ 0 , L , 0 ],
               [ 1 , 0 , 0 ],
               [ S , 0 , 0 ]
Expected output:  0.8037
5.  Horizontally displaced load (least efficient)
Ground truth:  [ 0 , 0 , L ],
               [ 0 , 1 , 0 ],
               [ S , 0 , 0 ]
Prediction:    [ 0 , 1 , L ],
               [ 1 , 0 , 0 ],
               [ S , 0 , 0 ]
Expected output:  0.7724
Interpretation:
As load-support paths deviate from the gravity direction or become
discontinuous,
FPCEff decreases from 1.0 toward 0, reflecting reduced
physical plausibility of the structure.
```

# E TOPOLOGY OPTIMIZATION SAMPLE PLOTS

## E.1 2D SAMPLES

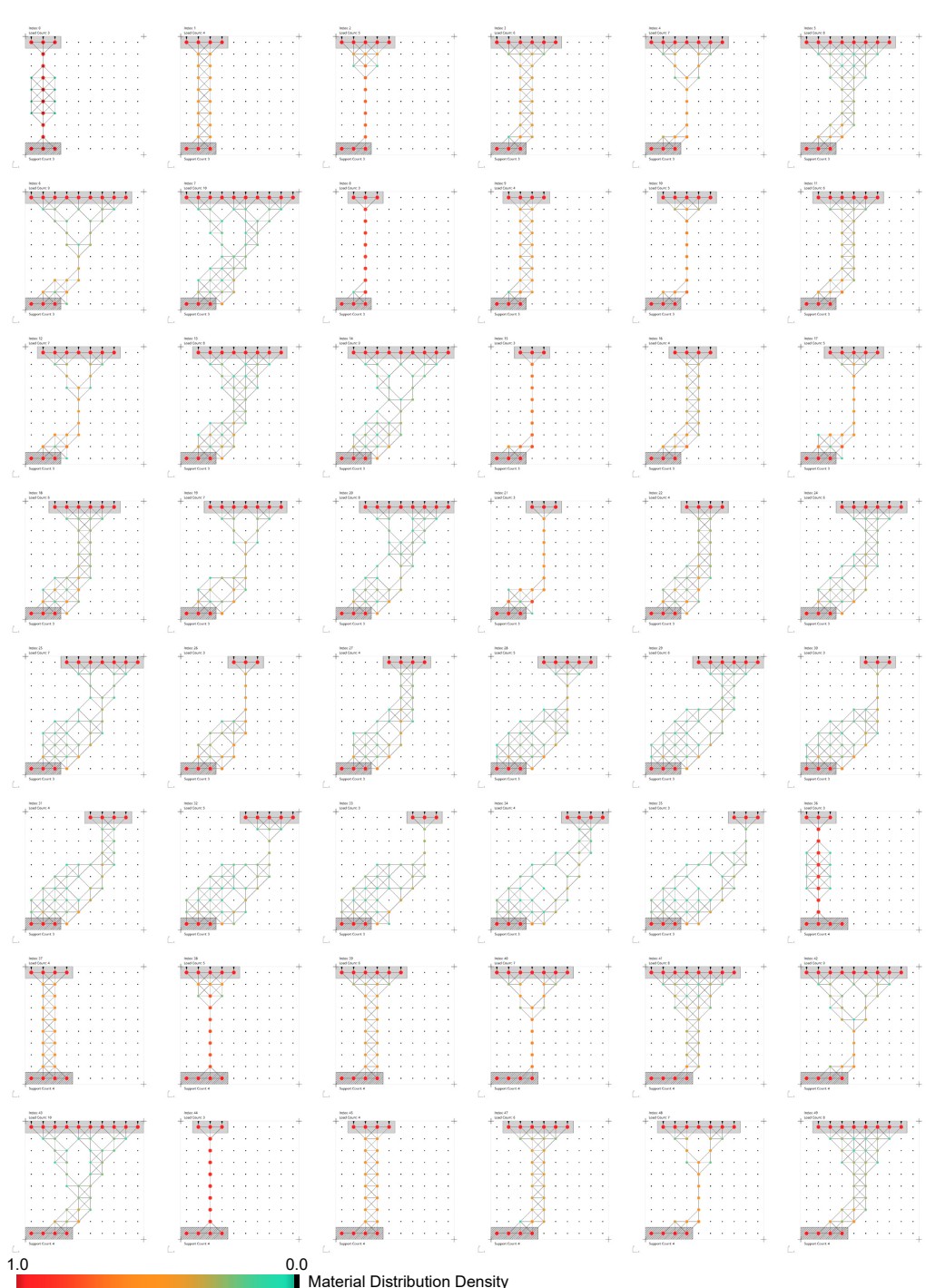

Figure 4: Example 2D topology optimization samples from the SPhyR dataset.

## E.2    3D SAMPLES

Figure 5: Example 3D topology optimization samples included for future benchmark extensions.

# F  ADDITIONAL MAIN RUN RESULTS

## F.1  RESULTS FOR ALL MODELS AND TASKS

Main Evaluation Results

**Reconstruction Accuracy Metrics 1/2**

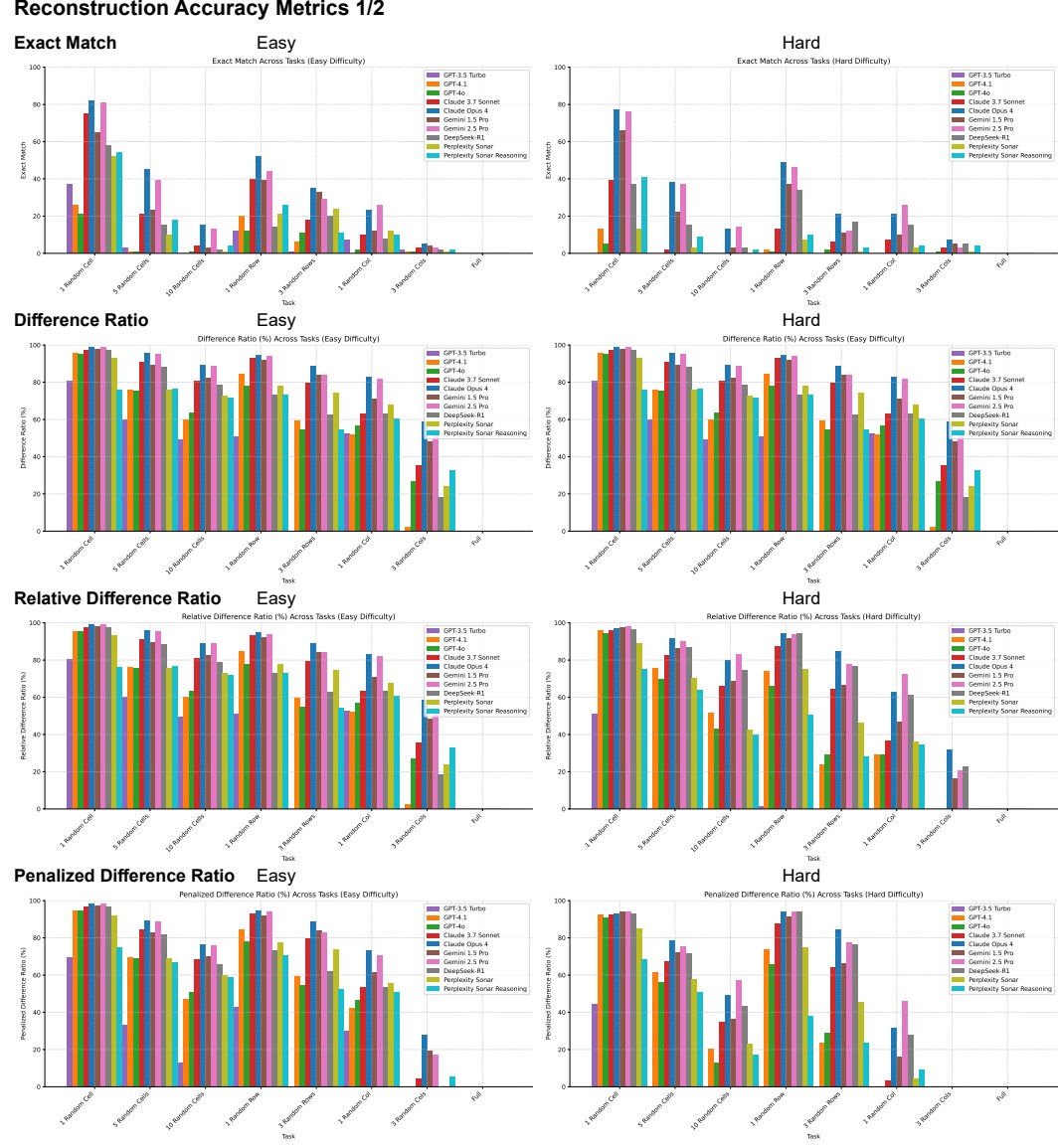

Figure 6: Main evaluation run results: Exact Match, Difference Ratio, Relative Difference Ratio and Penalized Difference Ratio for all models, across all tasks and difficulties.

Main Evaluation Results

**Reconstruction Accuracy Metrics 2/2**

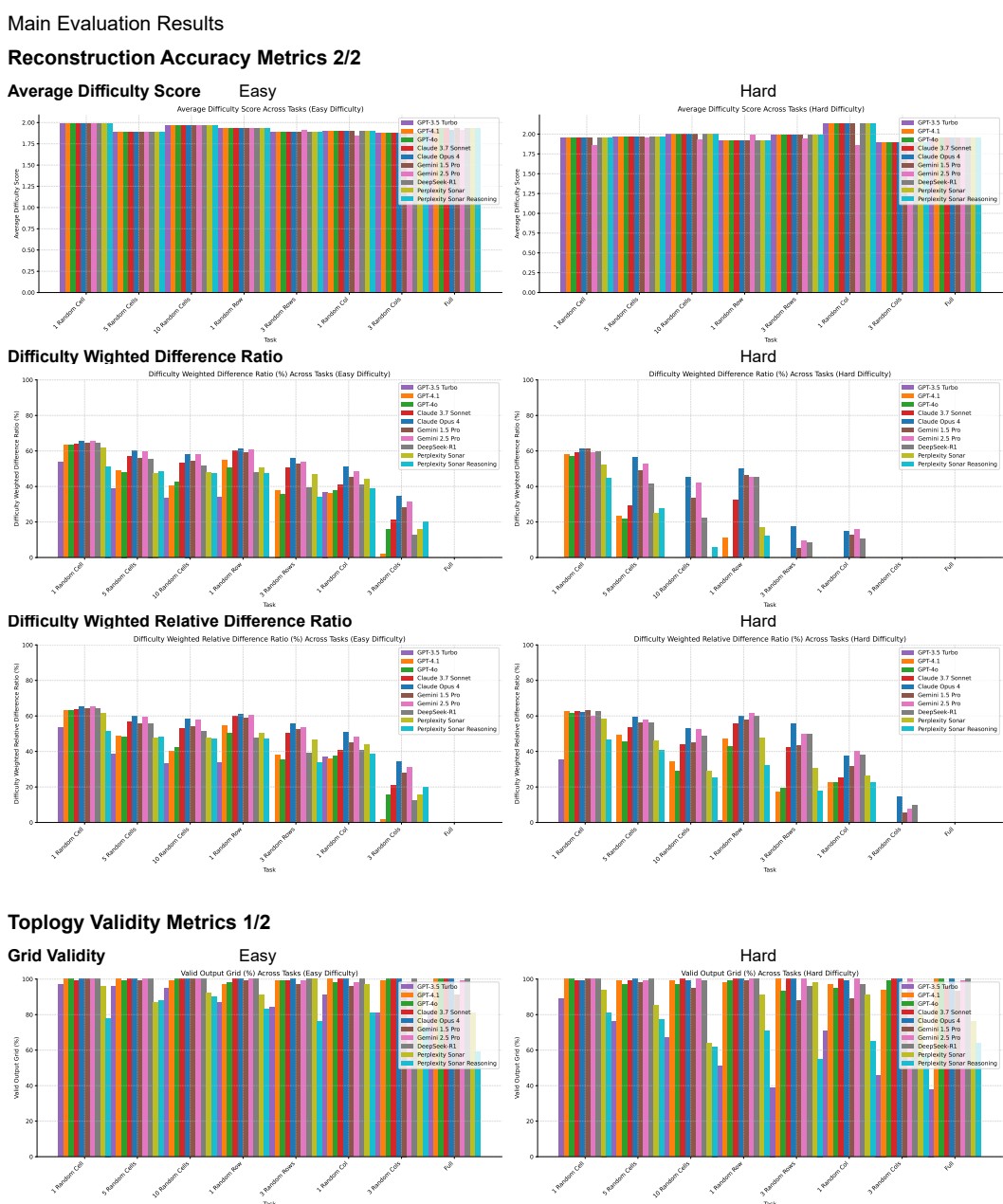

Figure 7: Main evaluation run results: Average Difficulty Score, Difficulty Weighted Difference Ratio, Difficulty Weighted Relative Difference Ratio and Grid Validity for all models, across all tasks and difficulties.

Main Evaluation Results

**Topology Validity Metrics 2/2**

**Load-Support Connectivity**   Easy                                      Hard

**Directional Load-Support Connectivity**                                 Hard

**Average Isolated Cluster Count**                                        Hard

**Physics-Approximating Metrics 1/1**

**Force Path Cost Average Efficiency Ratio**
Easy                                      Hard

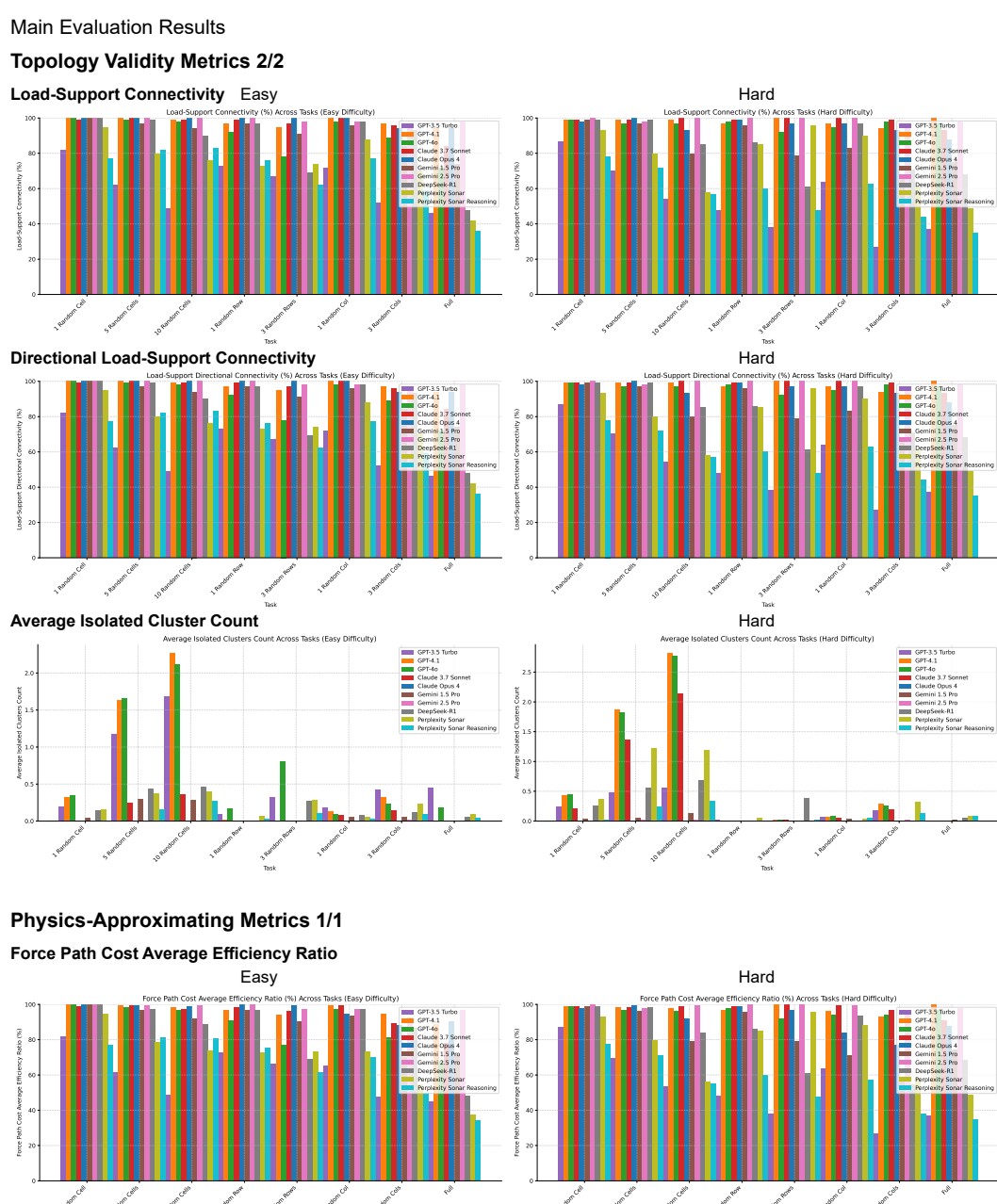

Figure 8: Main evaluation run results: Load-Support Connectivity, Directional Load-Support Connectivity, Average Isolated Cluster Count and Force Path Cost Average Efficiency Ratio for all models, across all tasks and difficulties.

# G    ADDITIONAL ROTATION EXPERIMENT RESULTS

## G.1    PROMPT AND COMPLETION - 1 RANDOM CELL EASY: ROTATED X 3

---

**Prompt and Completion - 1 Random Cell Easy: Rotated x 3**

**Prompt:**
You are given a structural material distribution represented as a grid.  Each cell can
have one of the following states:
- 'L' indicates applied load.
- 'V' indicates void.
- 'S' indicates support.

The goal is to predict the correct material distribution by filling in all 'V' cells
with either '1' (solid) or '0' (empty), based on the surrounding structure and implicit
physical reasoning (such as load paths, supports, and forces).

Important:  The completed structure should use as little material as possible while
remaining stable and plausible for carrying the applied forces.  Minimize material usage
unless necessary for structural support.

Below is the input grid with masked regions:

```
0 0 0 0 0 0 0 0 0 0
0 0 0 0 0 0 0 0 0 S
0 0 0 0 0 0 0 0 0 S
0 0 0 0 0 0 0 0 0 S
0 0 0 0 0 0 0 0 0 S
L 0 0 0 0 0 1 1 1 S
L 1 V 1 1 1 1 1 1 S
L 0 0 0 0 0 1 1 1 S
0 0 0 0 0 0 0 0 0 S
0 0 0 0 0 0 0 0 0 S
```

Please output the completed grid by replacing all 'V' cells with either '1' (solid) or '0'
(empty).
Maintain the same format as the input:  one row per line, cells separated by spaces, and
the total number of rows and columns unchanged.
Return only the completed grid without any additional explanation.

**Model Completion (differences in red):**
```
0 0 0 0 0 0 0 0 0 0
0 0 0 0 0 0 0 0 0 S
0 0 0 0 0 0 0 0 0 S
0 0 0 0 0 0 0 0 0 S
0 0 0 0 0 0 0 0 0 S
L 0 0 0 0 0 1 1 1 S
L 1 1 1 1 1 1 1 1 S
L 0 0 0 0 0 1 1 1 S
0 0 0 0 0 0 0 0 0 S
0 0 0 0 0 0 0 0 0 S
```

**Ground Truth:**
```
0 0 0 0 0 0 0 0 0 0
0 0 0 0 0 0 0 0 0 S
0 0 0 0 0 0 0 0 0 S
0 0 0 0 0 0 0 0 0 S
0 0 0 0 0 0 0 0 0 S
L 0 0 0 0 0 1 1 1 S
L 1 1 1 1 1 1 1 1 S
L 0 0 0 0 0 1 1 1 S
0 0 0 0 0 0 0 0 0 S
0 0 0 0 0 0 0 0 0 S
```

---

## G.2 RESULTS FOR MODEL SUB-SET AND TASK SUB-SET

Grid Rotation on All Models and Selected Tasks Evaluation Results: Easy

**Reconstruction Accuracy Metrics 1/1**

**Reconstruction Accuracy Metrics 1/2**

Figure 9: Grid rotation evaluation results: Exact Match, Difference Ratio, Relative Difference Ratio, Penalized Difference Ratio, Average Difficulty Score, Difficulty Weighted Difference Ratio, Difficulty Weighted Relative Difference Ratio and Grid Validity for GPT-4.1, Claude Opus 4, Gemini 2.5 Pro, DeepSeek-R1 and Perplexity, for 10 Random Cells, 3 Random Rows, 3 Random Cells and Full tasks and easy difficulty.

Grid Rotation on All Models and Selected Tasks Evaluation Results: Easy

**Topology Validity Metrics 2/2**

**Load-Support Connectivity** Easy    **Directional Load-Support Connectivity**

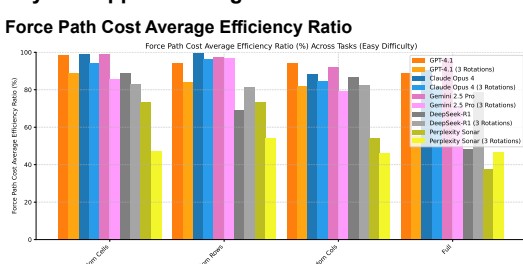

**Average Isolated Cluster Count**

**Physics-Approximating Metrics 1/1**

**Force Path Cost Average Efficiency Ratio**

Figure 10: Grid rotation evaluation results: Load-Support Connectivity, Directional Load-Support Connectivity, Average Isolated Cluster Count and Force Path Cost Average Efficiency Ratio for GPT-4.1, Claude Opus 4, Gemini 2.5 Pro, DeepSeek-R1 and Perplexity, for 10 Random Cells, 3 Random Rows, 3 Random Cells and Full tasks and easy difficulty.

Table 4: Grid rotation evaluation results for all metrics, for GPT-4.1, Claude Opus 4, Gemini 2.5 Pro, DeepSeek-R1 and Perplexity, for 10 Random Cells, 3 Random Rows, 3 Random Cells and Full tasks and easy difficulty.

| Task | Metric | GPT 4.1 | GPT 4.1 (3 Rotations) | Claude Opus 4 | Claude Opus 4 (3 Rotations) | Gemini 2.5 Pro | Gemini 2.5 Pro (3 Rotations) | DeepSeek-R1 | DeepSeek-R1 (3 Rotations) | Perplexity Sonar | Perplexity Sonar (3 Rotations) |
|---|---|---|---|---|---|---|---|---|---|---|---|
| **Difficulty: Easy** | | | | | | | | | | | |
| 10 Random Cells | Exact Match ↑ | 0 | 0 | 15 | **22** | 13 | 10 | 2 | 5 | 1 | 2 |
| | Difference Ratio (%) ↑ | 59.82 | 56.85 | 89.08 | **91.02** | 88.88 | 78.26 | 78.78 | 75.62 | 72.68 | 50.19 |
| | Relative Difference Ratio (%) ↑ | 59.82 | 56.85 | 89.08 | **91.02** | 88.88 | 78.26 | 78.78 | 75.62 | 72.68 | 50.19 |
| | Penalized Difference Ratio (%) ↑ | 47.02 | 44.95 | 76.41 | **80.78** | 76.21 | 66.98 | 65.95 | 64.27 | 59.85 | 41.15 |
| | Average Difficulty Score | **1.97** | 1.94 | **1.97** | 1.94 | **1.97** | 1.94 | **1.97** | 1.94 | **1.97** | 1.94 |
| | Difficulty Weighted Difference Ratio ↑ | 40.24 | 37.62 | 58.23 | **58.52** | 58.06 | 50.94 | 51.72 | 48.69 | 47.81 | 31.74 |
| | Difficulty Weighted Relative Difference Ratio (%) ↑ | 40.24 | 37.62 | 58.23 | **58.52** | 58.06 | 50.94 | 51.72 | 48.69 | 47.81 | 31.74 |
| | Valid Output Grid ↑ | 99.00 | 98.00 | **100.00** | **100.00** | **100.00** | 92.00 | **100.00** | 96.00 | 92.00 | 64.00 |
| | Load-Support Connectivity (%) ↑ | 99.00 | 98.00 | **100.00** | **100.00** | **100.00** | 92.00 | 90.00 | 92.00 | 76.00 | 47.00 |
| | Load-Support Directional Connectivity (%) ↑ | 99.00 | 83.00 | **100.00** | 86.00 | **100.00** | 78.00 | 90.00 | 72.00 | 76.00 | 32.00 |
| | Average Isolated Clusters Count ↓ | 2.27 | 2.24 | 0.00 | 0.00 | 0.00 | 0.07 | 0.46 | 0.43 | 0.40 | 0.29 |
| | Force Path Cost Average Efficiency Ratio (%) ↑ | 98.28 | 88.93 | 99.06 | 94.41 | **99.31** | 85.60 | 88.72 | 82.77 | 73.62 | 47.18 |
| 3 Random Rows | Exact Match ↑ | 6 | 1 | **35** | 17 | 29 | 3 | 20 | 9 | 24 | 0 |
| | Difference Ratio (%) ↑ | 59.31 | 23.28 | **88.64** | 82.19 | 84.09 | 65.52 | 62.64 | 40.38 | 74.23 | 26.11 |
| | Relative Difference Ratio (%) ↑ | 59.31 | 23.28 | **88.64** | 82.19 | 84.09 | 65.52 | 62.64 | 40.38 | 74.23 | 26.11 |
| | Penalized Difference Ratio (%) ↑ | 59.31 | 23.28 | **88.64** | 82.19 | 82.84 | 65.08 | 62.04 | 40.21 | 73.99 | 19.33 |
| | Average Difficulty Score | 1.89 | **1.96** | 1.89 | 1.89 | 1.92 | 1.89 | 1.89 | 1.89 | 1.89 | 1.89 |
| | Difficulty Weighted Difference Ratio ↑ | 37.91 | 16.47 | **55.81** | 51.60 | 53.52 | 42.02 | 39.27 | 26.05 | 46.97 | 17.17 |
| | Difficulty Weighted Relative Difference Ratio (%) ↑ | 37.91 | 16.47 | **55.81** | 51.60 | 53.52 | 42.02 | 39.27 | 26.05 | 46.97 | 17.17 |
| | Valid Output Grid ↑ | 99.00 | 88.00 | **100.00** | **100.00** | 99.00 | 98.00 | **100.00** | 96.00 | **100.00** | 83.00 |
| | Load-Support Connectivity (%) ↑ | 95.00 | 80.00 | **100.00** | 94.00 | 98.00 | 98.00 | 69.00 | 62.00 | 34.00 | 34.00 |
| | Load-Support Directional Connectivity (%) ↑ | 95.00 | 62.00 | **100.00** | 82.00 | 98.00 | 84.00 | 69.00 | 58.00 | 74.00 | 28.00 |
| | Average Isolated Clusters Count ↓ | 0.01 | 0.05 | 0.00 | 0.03 | 0.01 | 0.01 | 0.27 | 0.08 | 0.28 | **0.30** |
| | Force Path Cost Average Efficiency Ratio (%) ↑ | 94.09 | 83.83 | **99.68** | 96.39 | 97.54 | 96.68 | 69.22 | 81.37 | 73.47 | 54.16 |
| 3 Random Columns | Exact Match ↑ | 1 | 3 | **5** | 2 | 3 | 2 | 2 | 0 | 1 | 1 |
| | Difference Ratio (%) ↑ | 2.46 | 56.64 | 58.69 | **60.16** | 52.03 | 37.06 | 18.34 | 26.74 | 24.01 | 32.05 |
| | Relative Difference Ratio (%) ↑ | 2.46 | 56.64 | 58.69 | **60.16** | 52.03 | 37.06 | 18.34 | 26.74 | 24.01 | 32.05 |
| | Penalized Difference Ratio (%) ↑ | -28.47 | 25.76 | 27.96 | **30.59** | 17.06 | 8.67 | -12.54 | -4.77 | -9.25 | 15.25 |
| | Average Difficulty Score | 1.88 | 1.88 | 1.88 | **1.96** | 1.90 | **1.96** | 1.88 | **1.96** | 1.88 | **1.96** |
| | Difficulty Weighted Difference Ratio ↑ | 1.69 | 32.76 | 34.52 | **37.29** | 31.14 | 24.32 | 12.74 | 17.90 | 15.59 | 18.99 |
| | Difficulty Weighted Relative Difference Ratio (%) ↑ | 1.69 | 32.76 | 34.52 | **37.29** | 31.14 | 24.32 | 12.74 | 17.90 | 15.59 | 18.99 |
| | Valid Output Grid ↑ | 99.00 | 81.00 | **100.00** | **100.00** | 98.00 | 90.00 | **100.00** | 99.00 | 93.00 | 60.00 |
| | Load-Support Connectivity (%) ↑ | 97.00 | 81.00 | 94.00 | 84.00 | 96.00 | 89.00 | 93.00 | 88.00 | 70.00 | 36.00 |
| | Load-Support Directional Connectivity (%) ↑ | 97.00 | 69.00 | 94.00 | 68.00 | 96.00 | 68.00 | 93.00 | 69.00 | 70.00 | 31.00 |
| | Average Isolated Clusters Count ↓ | 0.32 | 0.17 | 0.00 | 0.01 | 0.02 | 0.05 | 0.12 | 0.10 | 0.24 | 0.13 |
| | Force Path Cost Average Efficiency Ratio (%) ↑ | 94.43 | 82.06 | 88.28 | 84.62 | 92.30 | 79.45 | 86.51 | 82.60 | 54.44 | 46.29 |
| Full | Exact Match ↑ | **0** | **0** | **0** | **0** | **0** | **0** | **0** | **0** | **0** | **0** |
| | Difference Ratio (%) ↑ | -62.06 | -32.96 | -35.78 | -32.88 | -25.03 | -32.85 | -126.02 | -84.98 | -49.16 | -34.84 |
| | Relative Difference Ratio (%) ↑ | -62.06 | -32.96 | -35.78 | -32.88 | -25.03 | -32.85 | -126.02 | -84.98 | -49.16 | -34.84 |
| | Penalized Difference Ratio (%) ↑ | -62.06 | -32.96 | -35.78 | -32.96 | -25.96 | -38.93 | -142.73 | -104.95 | -49.86 | -45.60 |
| | Average Difficulty Score | 1.93 | **1.95** | 1.92 | **1.95** | 1.91 | **1.95** | 1.93 | **1.95** | 1.93 | **1.95** |
| | Difficulty Weighted Difference Ratio ↑ | -37.69 | -20.14 | -21.79 | -20.08 | -14.61 | -19.98 | -79.62 | -54.42 | -29.78 | -21.20 |
| | Difficulty Weighted Relative Difference Ratio (%) ↑ | -37.69 | -20.14 | -21.79 | -20.08 | -14.61 | -19.98 | -79.62 | -54.42 | -29.78 | -21.20 |
| | Valid Output Grid ↑ | **100.00** | 89.00 | **100.00** | **100.00** | 99.00 | 81.00 | **100.00** | **100.00** | 81.00 | 84.00 |
| | Load-Support Connectivity (%) ↑ | 94.00 | 49.00 | 94.00 | 94.00 | **98.00** | 78.00 | 48.00 | 78.00 | 42.00 | 26.00 |
| | Load-Support Directional Connectivity (%) ↑ | 94.00 | 46.00 | 94.00 | 72.00 | **98.00** | 72.00 | 48.00 | 76.00 | 42.00 | 20.00 |
| | Average Isolated Clusters Count ↓ | 0.01 | 0.00 | 0.00 | 0.00 | 0.01 | 0.01 | 0.06 | 0.13 | 0.09 | **0.16** |
| | Force Path Cost Average Efficiency Ratio (%) ↑ | 88.87 | 55.53 | 90.33 | 81.69 | **96.85** | 70.68 | 48.31 | 79.00 | 37.61 | 46.55 |
| **Average** | Exact Match ↑ | 1 | 1 | **13** | 10 | 11 | 3 | 6 | 3 | 6 | 0 |
| | Difference Ratio (%) ↑ | 14.88 | 25.95 | **50.16** | 50.13 | 49.99 | 37.00 | 8.43 | 14.44 | 30.44 | 18.38 |
| | Relative Difference Ratio (%) ↑ | 14.88 | 25.95 | **50.16** | 50.13 | 49.99 | 37.00 | 8.43 | 14.44 | 30.44 | 18.38 |
| | Penalized Difference Ratio (%) ↑ | 3.95 | 15.26 | 39.31 | **40.15** | 37.54 | 25.45 | -6.82 | -1.31 | 18.68 | 7.53 |
| | Average Difficulty Score | 1.92 | **1.93** | 1.92 | **1.93** | 1.92 | **1.93** | 1.92 | **1.93** | 1.92 | **1.93** |
| | Difficulty Weighted Difference Ratio ↑ | 10.54 | 16.68 | 31.69 | 31.83 | **32.03** | 24.33 | 6.03 | 9.55 | 20.15 | 11.68 |
| | Difficulty Weighted Relative Difference Ratio (%) ↑ | 10.54 | 16.68 | 31.69 | 31.83 | **32.03** | 24.33 | 6.03 | 9.55 | 20.15 | 11.68 |
| | Valid Output Grid ↑ | 99.25 | 93.75 | **100.00** | **100.00** | 99.00 | 90.25 | **100.00** | 97.75 | 91.50 | 72.75 |
| | Load-Support Connectivity (%) ↑ | 96.25 | 77.00 | 97.00 | 93.00 | **98.00** | 89.25 | 75.00 | 80.00 | 65.50 | 35.75 |
| | Load-Support Directional Connectivity (%) ↑ | 96.25 | 65.00 | 97.00 | 77.00 | **98.00** | 75.50 | 75.00 | 68.75 | 65.50 | 27.75 |
| | Average Isolated Clusters Count ↓ | **0.65** | 0.61 | 0.00 | 0.01 | 0.01 | 0.04 | 0.23 | 0.18 | 0.25 | 0.22 |
| | Force Path Cost Average Efficiency Ratio (%) ↑ | 93.92 | 77.59 | 94.34 | 89.28 | **96.50** | 83.11 | 73.19 | 81.43 | 59.79 | 48.55 |

## G.3 RESULTS FOR CLAUDE OPUS 4 ON ALL TASKS

Grid Rotation on Claude Opus 4 Evaluation Results

**Reconstruction Accuracy Metrics 1/2**

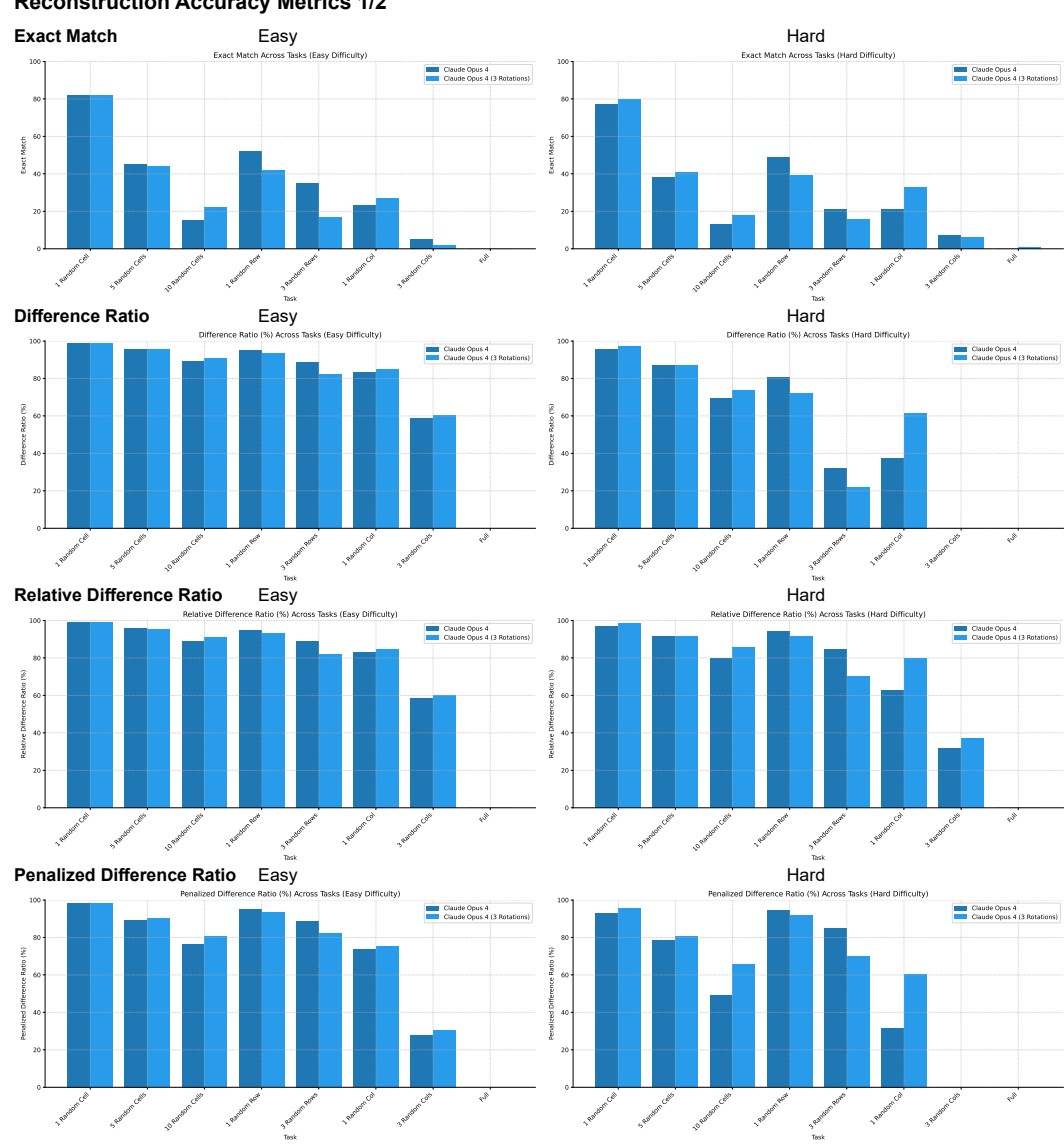

Figure 11: Grid rotation evaluation results: Exact Match, Difference Ratio, Relative Difference Ratio and Penalized Difference Ratio for Claude Opus 4, for all tasks, easy and hard difficulty.

Grid Rotation on Claude Opus 4 Evaluation Results

**Reconstruction Accuracy Metrics 2/2**

**Average Difficulty Score**     Easy                              Hard

**Difficulty Wighted Difference Ratio**                           Hard

**Difficulty Wighted Relative Difference Ratio**                  Hard

**Toplogy Validity Metrics 1/2**

**Grid Validity**              Easy                              Hard

Figure 12: Grid rotation evaluation results: Average Difficulty Score, Difficulty Weighted Difference Ratio, Difficulty Weighted Relative Difference Ratio and Grid Validity for Claude Opus 4, for all tasks, easy and hard difficulty.

Grid Rotation on Claude Opus 4 Evaluation Results

**Topology Validity Metrics 2/2**

**Load-Support Connectivity**   Easy                     Hard

**Directional Load-Support Connectivity**              Hard

**Average Isolated Cluster Count**                     Hard

**Physics-Approximating Metrics 1/1**

**Force Path Cost Average Efficiency Ratio**
Easy                                                   Hard

Figure 13: Grid rotation evaluation results: Load-Support Connectivity, Directional Load-Support Connectivity, Average Isolated Cluster Count and Force Path Cost Average Efficiency Ratio for Claude Opus 4, for all tasks, easy and hard difficulty.

Table 5: Grid rotation evaluation run results for all metrics for Claude Opus 4, for all tasks, easy and hard difficulty.

| Task | Metric | Easy | | Hard | |
|---|---|---|---|---|---|
| | | Claude Opus 4 | Claude Opus 4 (3 Rotations) | Claude Opus 4 | Claude Opus 4 (3 Rotations) |
| 1 Random Cell | Exact Match ↑ | **82** | **82** | 77 | **80** |
| | Difference Ratio (%) ↑ | **99.05** | 99.02 | 95.45 | **97.24** |
| | Relative Difference Ratio (%) ↑ | **99.05** | 99.02 | 96.72 | **98.32** |
| | Penalized Difference Ratio (%) ↑ | 98.40 | **98.47** | 93.11 | **95.60** |
| | Average Difficulty Score | **1.99** | **1.99** | **1.96** | 1.87 |
| | Difficulty Weighted Difference Ratio (%) ↑ | **65.51** | 65.40 | **60.97** | 59.80 |
| | Difficulty Weighted Relative Difference Ratio (%) ↑ | **65.51** | 65.40 | **62.25** | 60.88 |
| | Valid Output Grid ↑ | **100.00** | **100.00** | 99.00 | **100.00** |
| | Load-Support Connectivity (%) ↑ | **100.00** | **100.00** | 98.00 | **100.00** |
| | Load-Support Directional Connectivity (%) ↑ | **100.00** | 86.00 | **98.00** | 84.00 |
| | Average Isolated Clusters Count ↓ | **0.00** | **0.00** | **0.00** | **0.00** |
| | Force Path Cost Average Efficiency Ratio (%) ↑ | **99.94** | 99.72 | 97.91 | **100.00** |
| 5 Random Cells | Exact Match ↑ | **45** | 44 | 38 | **41** |
| | Difference Ratio (%) ↑ | **95.76** | 95.54 | **87.27** | 87.23 |
| | Relative Difference Ratio (%) ↑ | **95.76** | 95.54 | 91.58 | **91.71** |
| | Penalized Difference Ratio (%) ↑ | 89.26 | **90.06** | 78.42 | **80.79** |
| | Average Difficulty Score | **1.89** | **1.89** | 1.97 | **2.05** |
| | Difficulty Weighted Difference Ratio (%) ↑ | 59.89 | **59.91** | 56.17 | **58.59** |
| | Difficulty Weighted Relative Difference Ratio (%) ↑ | 59.89 | **59.91** | 59.65 | **62.07** |
| | Valid Output Grid ↑ | **100.00** | **100.00** | **100.00** | 99.00 |
| | Load-Support Connectivity (%) ↑ | **100.00** | **100.00** | **100.00** | 95.00 |
| | Load-Support Directional Connectivity (%) ↑ | **100.00** | 87.00 | **100.00** | 75.00 |
| | Average Isolated Clusters Count ↓ | **0.00** | **0.00** | **0.00** | **0.00** |
| | Force Path Cost Average Efficiency Ratio (%) ↑ | **99.70** | 98.20 | **99.49** | 95.37 |
| 10 Random Cells | Exact Match ↑ | 15 | **22** | 13 | **18** |
| | Difference Ratio (%) ↑ | 89.08 | **91.02** | 69.70 | **73.57** |
| | Relative Difference Ratio (%) ↑ | 89.08 | **91.02** | 79.91 | **85.93** |
| | Penalized Difference Ratio (%) ↑ | 76.41 | **80.78** | 49.33 | **65.62** |
| | Average Difficulty Score | **1.97** | 1.94 | **2.01** | 2.00 |
| | Difficulty Weighted Difference Ratio (%) ↑ | 58.23 | **58.52** | 45.17 | **47.31** |
| | Difficulty Weighted Relative Difference Ratio (%) ↑ | 58.23 | **58.52** | 52.88 | **56.59** |
| | Valid Output Grid ↑ | **100.00** | **100.00** | 99.00 | **100.00** |
| | Load-Support Connectivity (%) ↑ | **100.00** | **100.00** | 93.00 | **96.00** |
| | Load-Support Directional Connectivity (%) ↑ | **100.00** | 86.00 | **93.00** | 80.00 |
| | Average Isolated Clusters Count ↓ | **0.00** | **0.00** | 0.01 | **0.00** |
| | Force Path Cost Average Efficiency Ratio (%) ↑ | **99.06** | 94.41 | 91.98 | **95.00** |
| 1 Random Row | Exact Match ↑ | **52** | 42 | **49** | 39 |
| | Difference Ratio (%) ↑ | **94.92** | 93.30 | **80.55** | 71.92 |
| | Relative Difference Ratio (%) ↑ | **94.92** | 93.30 | **94.39** | 91.71 |
| | Penalized Difference Ratio (%) ↑ | **94.92** | 93.30 | **94.39** | 91.71 |
| | Average Difficulty Score | 1.94 | **1.98** | 1.92 | **1.97** |
| | Difficulty Weighted Difference Ratio (%) ↑ | 61.04 | **61.36** | **49.95** | 45.77 |
| | Difficulty Weighted Relative Difference Ratio (%) ↑ | 61.04 | **61.36** | 60.02 | **60.12** |
| | Valid Output Grid ↑ | **100.00** | **100.00** | **100.00** | **100.00** |
| | Load-Support Connectivity (%) ↑ | **100.00** | **100.00** | 99.00 | 98.00 |
| | Load-Support Directional Connectivity (%) ↑ | **100.00** | 82.00 | **99.00** | 82.00 |
| | Average Isolated Clusters Count ↓ | **0.00** | **0.00** | **0.00** | **0.00** |
| | Force Path Cost Average Efficiency Ratio (%) ↑ | 99.88 | **100.00** | **98.92** | 98.85 |
| 3 Random Rows | Exact Match ↑ | **35** | 17 | **21** | 16 |
| | Difference Ratio (%) ↑ | **88.64** | 82.19 | **32.01** | 21.98 |
| | Relative Difference Ratio (%) ↑ | **88.64** | 82.19 | **84.70** | 70.12 |
| | Penalized Difference Ratio (%) ↑ | **88.64** | 82.19 | **84.70** | 70.12 |
| | Average Difficulty Score | **1.89** | **1.89** | **1.99** | 1.96 |
| | Difficulty Weighted Difference Ratio (%) ↑ | **55.81** | 51.60 | **17.50** | 11.08 |
| | Difficulty Weighted Relative Difference Ratio (%) ↑ | **55.81** | 51.60 | **55.67** | 44.35 |
| | Valid Output Grid ↑ | **100.00** | **100.00** | **100.00** | **100.00** |
| | Load-Support Connectivity (%) ↑ | **100.00** | 94.00 | **97.00** | 88.00 |
| | Load-Support Directional Connectivity (%) ↑ | **100.00** | 82.00 | **97.00** | 78.00 |
| | Average Isolated Clusters Count ↓ | 0.00 | **0.03** | 0.00 | **0.04** |
| | Force Path Cost Average Efficiency Ratio (%) ↑ | **99.68** | 96.39 | **96.97** | 95.42 |
| 1 Random Column | Exact Match ↑ | 23 | **27** | 21 | **33** |
| | Difference Ratio (%) ↑ | 83.09 | **84.86** | 37.46 | **61.54** |
| | Relative Difference Ratio (%) ↑ | 83.09 | **84.86** | 62.68 | **79.72** |
| | Penalized Difference Ratio (%) ↑ | 73.51 | **75.48** | 31.65 | **60.26** |
| | Average Difficulty Score | 1.90 | **2.02** | **2.13** | 1.85 |
| | Difficulty Weighted Difference Ratio (%) ↑ | 50.85 | **54.04** | 14.68 | **27.37** |
| | Difficulty Weighted Relative Difference Ratio (%) ↑ | 50.85 | **54.04** | 37.62 | **44.46** |
| | Valid Output Grid ↑ | **100.00** | **100.00** | 99.00 | **100.00** |
| | Load-Support Connectivity (%) ↑ | **100.00** | 97.00 | **97.00** | 95.00 |
| | Load-Support Directional Connectivity (%) ↑ | **100.00** | 86.00 | **97.00** | 77.00 |
| | Average Isolated Clusters Count ↓ | **0.00** | 0.05 | 0.00 | **0.03** |
| | Force Path Cost Average Efficiency Ratio (%) ↑ | **94.90** | 92.78 | 84.17 | **93.51** |
| 3 Random Columns | Exact Match ↑ | **5** | 2 | **7** | 6 |
| | Difference Ratio (%) ↑ | 58.69 | **60.16** | -17.63 | -18.77 |
| | Relative Difference Ratio (%) ↑ | 58.69 | **60.16** | 31.78 | **36.93** |
| | Penalized Difference Ratio (%) ↑ | 27.96 | **30.59** | -43.98 | **-38.28** |
| | Average Difficulty Score | 1.88 | **1.96** | 1.90 | **2.01** |
| | Difficulty Weighted Difference Ratio (%) ↑ | 34.52 | **37.29** | -22.99 | **-21.91** |
| | Difficulty Weighted Relative Difference Ratio (%) ↑ | 34.52 | **37.29** | 14.44 | **20.53** |
| | Valid Output Grid ↑ | **100.00** | **100.00** | **100.00** | **100.00** |
| | Load-Support Connectivity (%) ↑ | **94.00** | 84.00 | **93.00** | 89.00 |
| | Load-Support Directional Connectivity (%) ↑ | **94.00** | 68.00 | **93.00** | 73.00 |
| | Average Isolated Clusters Count ↓ | **0.00** | 0.01 | **0.00** | 0.01 |
| | Force Path Cost Average Efficiency Ratio (%) ↑ | **88.28** | 84.62 | 77.05 | **82.98** |
| Full | Exact Match ↑ | **0** | **0** | 0 | **1** |
| | Difference Ratio (%) ↑ | -35.78 | **-32.88** | -466.42 | **-403.39** |
| | Relative Difference Ratio (%) ↑ | -35.78 | **-32.88** | -177.48 | **-136.14** |
| | Penalized Difference Ratio (%) ↑ | -35.78 | **-32.96** | -177.48 | **-138.43** |
| | Average Difficulty Score | 1.92 | **1.95** | **1.96** | 1.93 |
| | Difficulty Weighted Difference Ratio (%) ↑ | -21.79 | **-20.08** | -310.26 | **-266.48** |
| | Difficulty Weighted Relative Difference Ratio (%) ↑ | -21.79 | **-20.08** | -117.17 | **-90.28** |
| | Valid Output Grid ↑ | **100.00** | **100.00** | **100.00** | **100.00** |
| | Load-Support Connectivity (%) ↑ | **94.00** | **94.00** | **88.00** | 77.00 |
| | Load-Support Directional Connectivity (%) ↑ | **94.00** | 72.00 | **88.00** | 74.00 |
| | Average Isolated Clusters Count ↓ | **0.00** | **0.00** | **0.00** | **0.00** |
| | Force Path Cost Average Efficiency Ratio (%) ↑ | **90.33** | 81.69 | **87.83** | 86.43 |
| **Average** | Exact Match ↑ | **32.12** | 29.50 | 28.25 | **29.25** |
| | Difference Ratio (%) ↑ | **71.68** | 71.65 | -10.20 | **-1.08** |
| | Relative Difference Ratio (%) ↑ | **71.68** | 71.65 | 45.53 | **52.29** |
| | Penalized Difference Ratio (%) ↑ | 64.16 | **64.74** | 26.27 | **35.92** |
| | Average Difficulty Score | 1.92 | **1.95** | **1.98** | 1.95 |
| | Difficulty Weighted Difference Ratio (%) ↑ | 45.51 | **46.01** | -11.10 | **-4.81** |
| | Difficulty Weighted Relative Difference Ratio (%) ↑ | 45.51 | **46.01** | 28.17 | **32.34** |
| | Valid Output Grid ↑ | **100.00** | **100.00** | 99.62 | **99.88** |
| | Load-Support Connectivity (%) ↑ | **98.50** | 96.12 | **95.62** | 92.25 |
| | Load-Support Directional Connectivity (%) ↑ | **98.50** | 81.12 | **95.62** | 77.88 |
| | Average Isolated Clusters Count ↓ | **0.00** | 0.01 | **0.00** | 0.01 |
| | Force Path Cost Average Efficiency Ratio (%) ↑ | **96.47** | 93.48 | 91.79 | **93.44** |

# H ADDITIONAL FEW-SHOT EXPERIMENT RESULTS

## H.1 PROMPT AND COMPLETION - 1 RANDOM CELL EASY: 1-SHOT

---

**Prompt and Completion - 1 Random Cell Easy: 1-Shot**

**Prompt:**
You are given a structural material distribution represented as a grid. Each cell can
have one of the following states:
- 'L' indicates applied load.
- 'V' indicates void.
- 'S' indicates support.
The goal is to predict the correct material distribution by filling in all 'V' cells
with either '1' (solid) or '0' (empty), based on the surrounding structure and implicit
physical reasoning (such as load paths, supports, and forces).
Example input grid with masked regions:
```
0 0 0 L L L L 0 0 0
0 0 0 0 1 1 0 0 0 0
0 0 0 0 1 1 0 V 0 0
0 0 0 0 1 1 0 0 0 0
0 0 0 0 1 1 0 0 0 0
0 0 0 0 1 1 0 0 0 0
0 0 0 0 1 1 0 0 0 0
0 0 0 0 1 1 0 0 0 0
0 0 0 0 1 1 0 0 0 0
0 0 0 0 S S S S 0
```
Corresponding completed output grid:
```
0 0 0 L L L L 0 0 0
0 0 0 0 1 1 0 0 0 0
0 0 0 0 1 1 0 0 0 0
0 0 0 0 1 1 0 0 0 0
0 0 0 0 1 1 0 0 0 0
0 0 0 0 1 1 0 0 0 0
0 0 0 0 1 1 0 0 0 0
0 0 0 0 1 1 0 0 0 0
0 0 0 0 S S S S 0
```
Important:  The completed structure should use as little material as possible while
remaining stable and plausible for carrying the applied forces.  Minimize material usage
unless necessary for structural support.
Below is the input grid with masked regions:
```
0 0 L L L 0 0 0 0 0
0 0 0 1 0 0 0 0 0 0
0 0 0 V 0 0 0 0 0 0
0 0 0 1 0 0 0 0 0 0
0 0 0 1 0 0 0 0 0 0
0 0 0 1 0 0 0 0 0 0
0 0 1 1 1 0 0 0 0 0
0 0 1 1 1 0 0 0 0 0
0 0 1 1 1 0 0 0 0 0
S S S S S S S S S 0
```
Please output the completed grid by replacing all 'V' cells with either '1' (solid) or '0'
(empty).
Maintain the same format as the input:  one row per line, cells separated by spaces, and
the total number of rows and columns unchanged.
Return only the completed grid without any additional explanation.
**Model Completion (differences in red):**
```
0 0 L L L 0 0 0 0 0
0 0 0 1 0 0 0 0 0 0
0 0 0 1 0 0 0 0 0 0
0 0 0 1 0 0 0 0 0 0
0 0 0 1 0 0 0 0 0 0
0 0 0 1 0 0 0 0 0 0
0 0 1 1 1 0 0 0 0 0
0 0 1 1 1 0 0 0 0 0
0 0 1 1 1 0 0 0 0 0
S S S S S S S S S 0
```
**Ground Truth:**
```
0 0 L L L 0 0 0 0 0
0 0 0 1 0 0 0 0 0 0
0 0 0 1 0 0 0 0 0 0
0 0 0 1 0 0 0 0 0 0
0 0 0 1 0 0 0 0 0 0
0 0 0 1 0 0 0 0 0 0
0 0 1 1 1 0 0 0 0 0
0 0 1 1 1 0 0 0 0 0
0 0 1 1 1 0 0 0 0 0
S S S S S S S S S 0
```

---

## H.2 PROMPT AND COMPLETION - 1 RANDOM CELL EASY: 3-SHOT

### Prompt and Completion - 1 Random Cell Easy: 3-Shot

**Prompt:**
```
You are given a structural material distribution represented as a grid.  Each cell can
have one of the following states:
- 'L' indicates applied load.
- 'V' indicates void.
- 'S' indicates support.
The goal is to predict the correct material distribution by filling in all 'V' cells
with either '1' (solid) or '0' (empty), based on the surrounding structure and implicit
physical reasoning (such as load paths, supports, and forces).
Example input grid with masked regions:
0 0 0 L L L L 0 0 0
0 0 0 0 1 1 0 0 0 0
0 0 0 0 1 1 0 V 0 0
0 0 0 0 1 1 0 0 0 0
0 0 0 0 1 1 0 0 0 0
0 0 0 0 1 1 0 0 0 0
0 0 0 0 1 1 0 0 0 0
0 0 0 0 1 1 0 0 0 0
0 0 0 0 S S S S 0
Corresponding completed output grid:
0 0 0 L L L L 0 0 0
0 0 0 0 1 1 0 0 0 0
0 0 0 0 1 1 0 0 0 0
0 0 0 0 1 1 0 0 0 0
0 0 0 0 1 1 0 0 0 0
0 0 0 0 1 1 0 0 0 0
0 0 0 0 1 1 0 0 0 0
0 0 0 0 1 1 0 0 0 0
0 0 0 0 S S S S 0
Example input grid with masked regions:
0 0 0 L L L 0 0 0 0
0 0 0 0 1 0 0 0 0 0
0 0 0 0 1 0 0 0 0 0
0 0 0 0 1 0 0 0 0 0
0 0 0 0 1 0 0 0 0 0
V 0 0 0 1 0 0 0 0 0
0 0 0 0 1 0 0 0 0 0
0 0 0 0 1 1 1 1 0 0
0 0 0 0 0 1 1 1 0 0
0 0 0 0 0 1 S S S 0
Corresponding completed output grid:
0 0 0 L L L 0 0 0 0
0 0 0 0 1 0 0 0 0 0
0 0 0 0 1 0 0 0 0 0
0 0 0 0 1 0 0 0 0 0
0 0 0 0 1 0 0 0 0 0
0 0 0 0 1 0 0 0 0 0
0 0 0 0 1 0 0 0 0 0
0 0 0 0 1 1 1 1 0 0
0 0 0 0 0 1 1 1 0 0
0 0 0 0 0 1 S S S 0
Example input grid with masked regions:
0 0 L L L L 0 0 0
0 0 0 1 1 1 0 0 0 0
0 0 0 1 1 1 V 0 0 0
0 0 0 0 1 0 0 0 0 0
0 0 0 0 1 0 0 0 0 0
0 0 0 0 1 0 0 0 0 0
0 0 0 1 1 1 0 0 0 0
0 0 0 1 1 1 0 0 0 0
0 0 0 1 1 1 0 0 0 0
0 S S S S S S 0 0
Corresponding completed output grid:
0 0 L L L L 0 0 0
0 0 0 1 1 1 0 0 0 0
0 0 0 1 1 1 0 0 0 0
0 0 0 0 1 0 0 0 0 0
0 0 0 0 1 0 0 0 0 0
0 0 0 0 1 0 0 0 0 0
0 0 0 1 1 1 0 0 0 0
0 0 0 1 1 1 0 0 0 0
0 0 0 1 1 1 0 0 0 0
0 S S S S S S 0 0
```

```
Important:  The completed structure should use as little material as possible while
remaining stable and plausible for carrying the applied forces.  Minimize material usage
unless necessary for structural support.
Below is the input grid with masked regions:
0 0 L L L 0 0 0 0 0
0 0 0 1 0 0 0 0 0 0
0 0 0 V 0 0 0 0 0 0
0 0 0 1 0 0 0 0 0 0
0 0 0 1 0 0 0 0 0 0
0 0 0 1 0 0 0 0 0 0
0 0 1 1 1 0 0 0 0 0
0 0 1 1 1 0 0 0 0 0
0 0 1 1 1 0 0 0 0 0
S S S S S S S S S 0
Please output the completed grid by replacing all 'V' cells with either '1' (solid) or '0'
(empty).
Maintain the same format as the input:  one row per line, cells separated by spaces, and
the total number of rows and columns unchanged.
Return only the completed grid without any additional explanation.
```

**Model Completion (differences in red):**
```
0 0 L L L 0 0 0 0 0
0 0 0 1 0 0 0 0 0 0
0 0 0 1 0 0 0 0 0 0
0 0 0 1 0 0 0 0 0 0
0 0 0 1 0 0 0 0 0 0
0 0 0 1 0 0 0 0 0 0
0 0 1 1 1 0 0 0 0 0
0 0 1 1 1 0 0 0 0 0
0 0 1 1 1 0 0 0 0 0
S S S S S S S S S 0
```

**Ground Truth:**
```
0 0 L L L 0 0 0 0 0
0 0 0 1 0 0 0 0 0 0
0 0 0 1 0 0 0 0 0 0
0 0 0 1 0 0 0 0 0 0
0 0 0 1 0 0 0 0 0 0
0 0 0 1 0 0 0 0 0 0
0 0 1 1 1 0 0 0 0 0
0 0 1 1 1 0 0 0 0 0
0 0 1 1 1 0 0 0 0 0
S S S S S S S S S 0
```

## H.3 RESULTS FOR CLAUDE OPUS 4 ON ALL TASKS

Few-Shot Evaluation Results

**Reconstruction Accuracy Metrics 1/2**

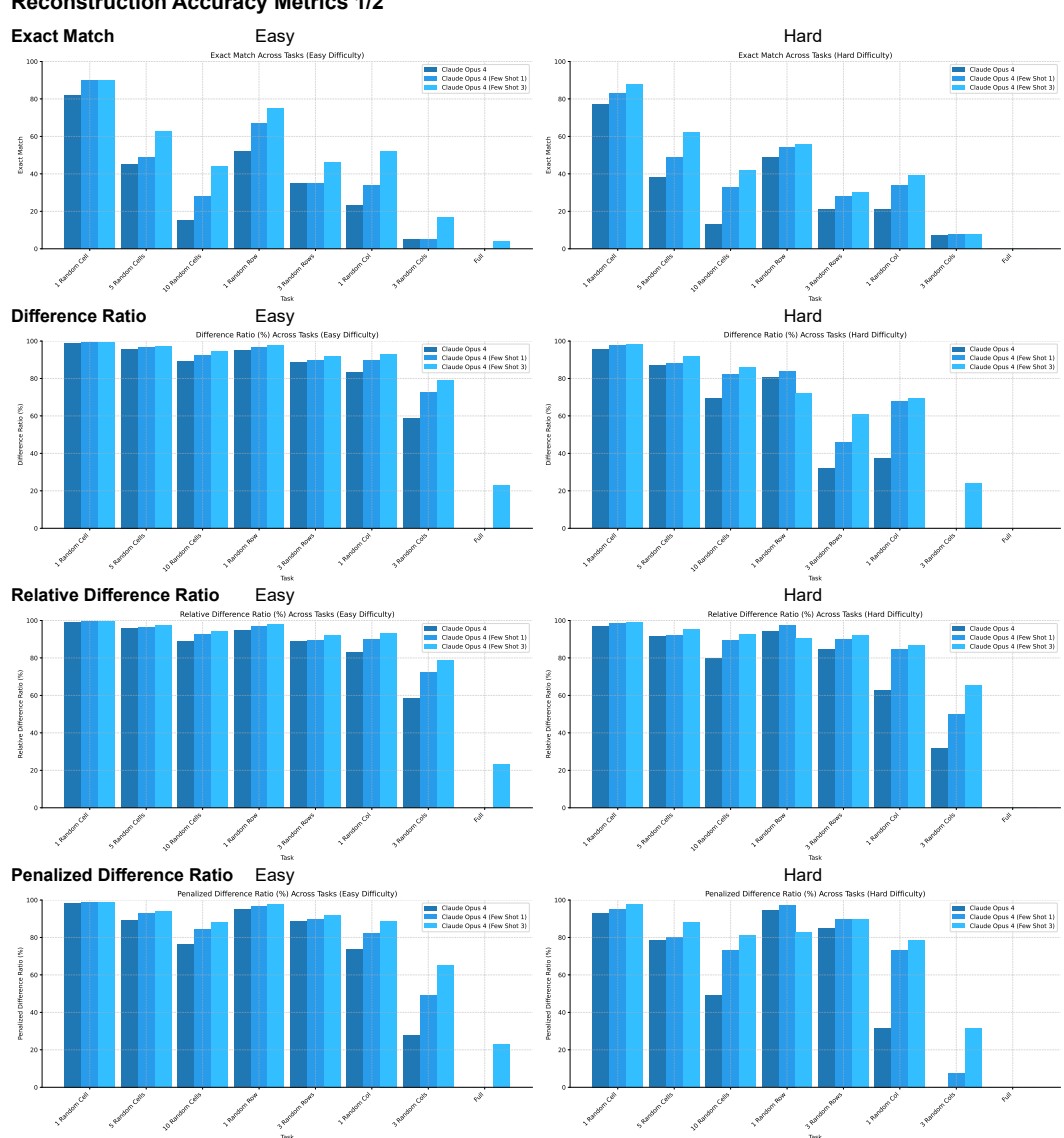

Figure 14: Few-shot (1, 3) evaluation results: Exact Match, Difference Ratio, Relative Difference Ratio and Penalized Difference Ratio for Claude Opus 4, for all tasks, easy and hard difficulty.

Few-Shot Evaluation Results

**Reconstruction Accuracy Metrics 2/2**

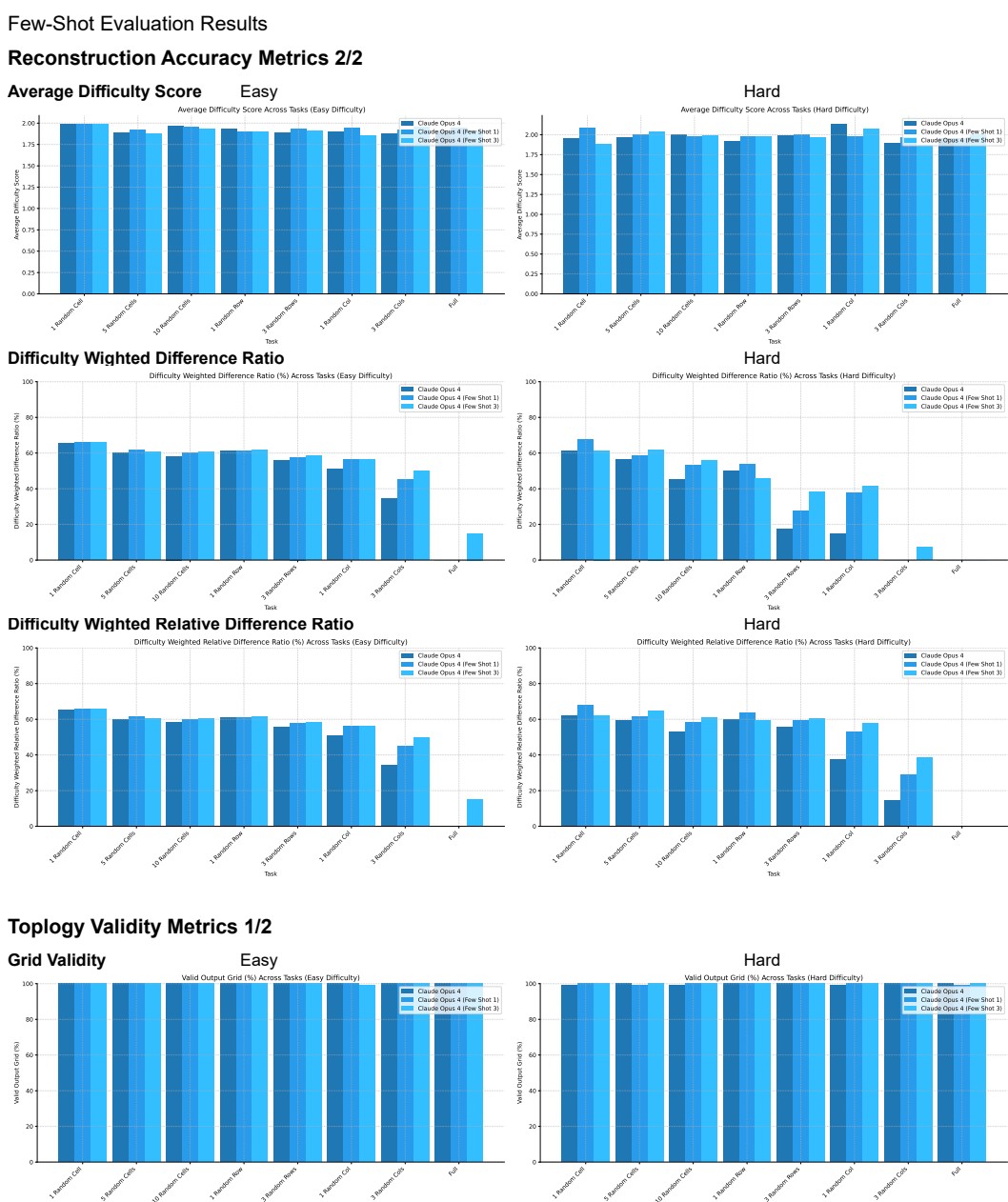

Figure 15: Few-shot (1, 3) evaluation results: Average Difficulty Score, Difficulty Weighted Difference Ratio, Difficulty Weighted Relative Difference Ratio and Grid Validity for Claude Opus 4, for all tasks, easy and hard difficulty.

Few-Shot Evaluation Results

**Topology Validity Metrics 2/2**

**Load-Support Connectivity**    Easy                                          Hard

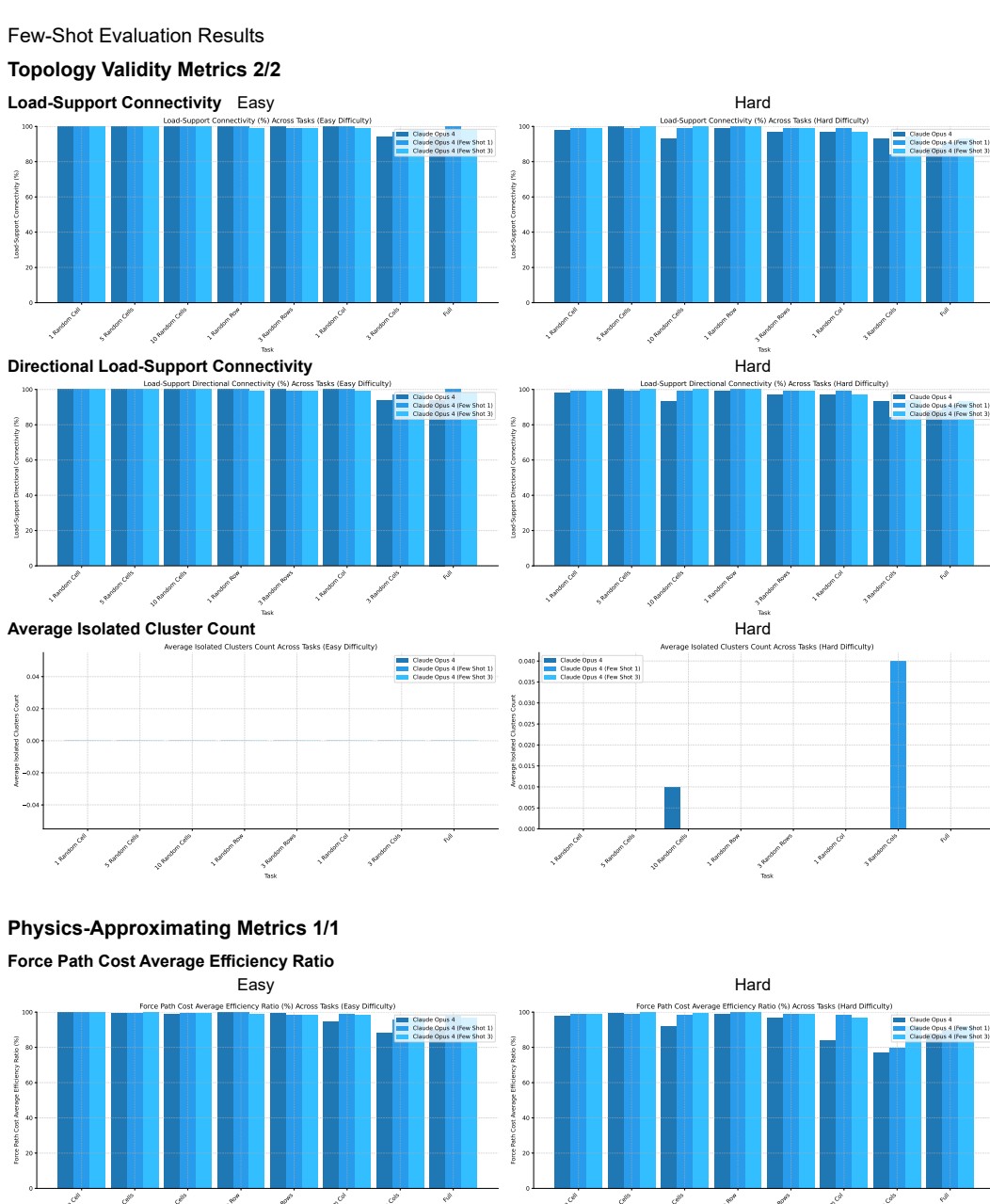

Figure 16: Few-shot (1, 3) evaluation results: Load-Support Connectivity, Directional Load-Support Connectivity, Average Isolated Cluster Count and Force Path Cost Average Efficiency Ratio for Claude Opus 4, for all tasks, easy and hard difficulty.

Table 6: Few-shot (1, 3) evaluation results for all metrics, for Claude Opus 4, for all tasks, easy and hard difficulty.

| Task | Metric | Easy Claude Opus 4 (Zero-Shot) | Claude Opus 4 (1-Shot) | Claude Opus 4 (3-Shot) | Hard Claude Opus 4 (Zero-Shot) | Claude Opus 4 (1-Shot) | Claude Opus 4 (3-Shot) |
|---|---|---|---|---|---|---|---|
| 1 Random Cell | Exact Match ↑ | 82 | **90** | **90** | 77 | 83 | **88** |
| | Difference Ratio (%) ↑ | 99.05 | **99.51** | 99.50 | 95.45 | 97.64 | **98.33** |
| | Relative Difference Ratio (%) ↑ | 99.05 | **99.51** | 99.50 | 96.72 | 98.28 | **99.13** |
| | Penalized Difference Ratio (%) ↑ | 98.40 | 98.93 | **99.03** | 93.11 | 95.20 | **97.72** |
| | Average Difficulty Score | **1.99** | **1.99** | **1.99** | 1.96 | **2.09** | 1.89 |
| | Difficulty Weighted Difference Ratio (%) ↑ | 65.51 | **65.92** | 65.88 | 60.97 | **67.45** | 61.43 |
| | Difficulty Weighted Relative Difference Ratio (%) ↑ | 65.51 | **65.92** | 65.88 | 62.25 | **68.09** | 62.22 |
| | Valid Output Grid ↑ | 100.00 | 100.00 | 100.00 | 99.00 | 100.00 | 100.00 |
| | Load-Support Connectivity (%) ↑ | 100.00 | 100.00 | 100.00 | 98.00 | 99.00 | 99.00 |
| | Load-Support Directional Connectivity (%) ↑ | 100.00 | 100.00 | 100.00 | 98.00 | 99.00 | 99.00 |
| | Average Isolated Clusters Count ↓ | 0.00 | 0.00 | 0.00 | 0.00 | 0.00 | 0.00 |
| | Force Path Cost Average Efficiency Ratio (%) ↑ | 99.94 | 99.94 | **99.95** | 97.91 | 98.96 | **99.00** |
| 5 Random Cells | Exact Match ↑ | 45 | 49 | **63** | 38 | 49 | **62** |
| | Difference Ratio (%) ↑ | 95.76 | 96.53 | **97.25** | 87.27 | 88.28 | **91.85** |
| | Relative Difference Ratio (%) ↑ | 95.76 | 96.53 | **97.25** | 91.58 | 92.10 | **95.51** |
| | Penalized Difference Ratio (%) ↑ | 89.26 | 92.90 | **93.93** | 78.42 | 80.15 | **88.27** |
| | Average Difficulty Score | 1.89 | **1.92** | 1.88 | 1.97 | 2.01 | **2.04** |
| | Difficulty Weighted Difference Ratio (%) ↑ | 59.89 | **61.73** | 60.69 | 56.17 | 58.62 | **61.94** |
| | Difficulty Weighted Relative Difference Ratio (%) ↑ | 59.89 | **61.73** | 60.69 | 59.65 | 61.59 | **64.85** |
| | Valid Output Grid ↑ | 100.00 | 100.00 | 100.00 | 100.00 | 99.00 | 100.00 |
| | Load-Support Connectivity (%) ↑ | 100.00 | 100.00 | 100.00 | 100.00 | 99.00 | 100.00 |
| | Load-Support Directional Connectivity (%) ↑ | 100.00 | 100.00 | 100.00 | 100.00 | 99.00 | 100.00 |
| | Average Isolated Clusters Count ↓ | 0.00 | 0.00 | 0.00 | 0.00 | 0.00 | 0.00 |
| | Force Path Cost Average Efficiency Ratio (%) ↑ | 99.70 | 99.66 | **99.87** | 99.49 | 98.89 | **99.90** |
| 10 Random Cells | Exact Match ↑ | 15 | 28 | **44** | 13 | 33 | **42** |
| | Difference Ratio (%) ↑ | 89.08 | 92.63 | **94.31** | 69.70 | 82.11 | **85.88** |
| | Relative Difference Ratio (%) ↑ | 89.08 | 92.63 | **94.31** | 79.91 | 89.57 | **92.63** |
| | Penalized Difference Ratio (%) ↑ | 76.41 | 84.37 | **88.11** | 49.33 | 73.32 | **81.11** |
| | Average Difficulty Score | **1.97** | 1.96 | 1.94 | 2.01 | 1.98 | 1.99 |
| | Difficulty Weighted Difference Ratio (%) ↑ | 58.23 | 60.24 | **60.77** | 45.17 | 52.92 | **56.05** |
| | Difficulty Weighted Relative Difference Ratio (%) ↑ | 58.23 | 60.24 | **60.77** | 52.88 | 58.43 | **61.07** |
| | Valid Output Grid ↑ | 100.00 | 100.00 | 100.00 | 99.00 | 100.00 | 100.00 |
| | Load-Support Connectivity (%) ↑ | 100.00 | 100.00 | 100.00 | 93.00 | 99.00 | 100.00 |
| | Load-Support Directional Connectivity (%) ↑ | 100.00 | 100.00 | 100.00 | 93.00 | 99.00 | 100.00 |
| | Average Isolated Clusters Count ↓ | 0.00 | 0.00 | 0.00 | 0.01 | 0.00 | 0.00 |
| | Force Path Cost Average Efficiency Ratio (%) ↑ | 99.06 | 99.32 | **99.37** | 91.98 | 98.41 | **99.55** |
| 1 Random Row | Exact Match ↑ | 52 | 67 | **75** | 49 | 54 | **56** |
| | Difference Ratio (%) ↑ | 94.92 | 96.86 | **97.72** | 80.55 | **83.72** | 72.00 |
| | Relative Difference Ratio (%) ↑ | 94.92 | 96.86 | **97.72** | 94.39 | **97.25** | 90.29 |
| | Penalized Difference Ratio (%) ↑ | 94.92 | 96.86 | **97.72** | 94.39 | **97.25** | 82.61 |
| | Average Difficulty Score | **1.94** | 1.91 | 1.90 | 1.92 | 1.98 | **1.98** |
| | Difficulty Weighted Difference Ratio (%) ↑ | 61.04 | 61.29 | **61.66** | 49.95 | **53.53** | 45.64 |
| | Difficulty Weighted Relative Difference Ratio (%) ↑ | 61.04 | 61.29 | **61.66** | 60.02 | **63.78** | 59.35 |
| | Valid Output Grid ↑ | 100.00 | 100.00 | 100.00 | 100.00 | 100.00 | 100.00 |
| | Load-Support Connectivity (%) ↑ | 100.00 | 100.00 | 99.00 | 99.00 | 100.00 | 100.00 |
| | Load-Support Directional Connectivity (%) ↑ | 100.00 | 100.00 | 99.00 | 99.00 | 100.00 | 100.00 |
| | Average Isolated Clusters Count ↓ | 0.00 | 0.00 | 0.00 | 0.00 | 0.00 | 0.00 |
| | Force Path Cost Average Efficiency Ratio (%) ↑ | 99.88 | **99.96** | 98.85 | 98.92 | **99.99** | 99.84 |
| 3 Random Rows | Exact Match ↑ | 35 | 35 | **46** | 21 | 28 | **30** |
| | Difference Ratio (%) ↑ | 88.64 | 89.67 | **91.83** | 32.01 | 45.88 | **60.79** |
| | Relative Difference Ratio (%) ↑ | 88.64 | 89.67 | **91.83** | 84.70 | 89.71 | **92.24** |
| | Penalized Difference Ratio (%) ↑ | 88.64 | 89.67 | **91.83** | 84.70 | 89.71 | **89.97** |
| | Average Difficulty Score | 1.89 | **1.94** | 1.92 | 1.99 | **2.00** | 1.97 |
| | Difficulty Weighted Difference Ratio (%) ↑ | 55.81 | 57.69 | **58.32** | 17.50 | 27.29 | **38.05** |
| | Difficulty Weighted Relative Difference Ratio (%) ↑ | 55.81 | 57.69 | **58.32** | 55.67 | 59.27 | **60.37** |
| | Valid Output Grid ↑ | 100.00 | 100.00 | 100.00 | 100.00 | 100.00 | 100.00 |
| | Load-Support Connectivity (%) ↑ | 100.00 | 99.00 | 99.00 | 97.00 | 99.00 | 99.00 |
| | Load-Support Directional Connectivity (%) ↑ | 100.00 | 99.00 | 99.00 | 97.00 | 99.00 | 99.00 |
| | Average Isolated Clusters Count ↓ | 0.00 | 0.00 | 0.00 | 0.00 | 0.00 | 0.00 |
| | Force Path Cost Average Efficiency Ratio (%) ↑ | 99.68 | 98.49 | 98.62 | 96.97 | **98.75** | 98.73 |
| 1 Random Column | Exact Match ↑ | 23 | 34 | **52** | 21 | 34 | **39** |
| | Difference Ratio (%) ↑ | 83.09 | 89.98 | **93.14** | 37.46 | 67.96 | **69.20** |
| | Relative Difference Ratio (%) ↑ | 83.09 | 89.98 | **93.14** | 62.68 | 84.74 | **86.73** |
| | Penalized Difference Ratio (%) ↑ | 73.51 | 82.16 | **88.94** | 31.65 | 73.11 | **78.56** |
| | Average Difficulty Score | 1.90 | **1.95** | 1.86 | **2.13** | 1.98 | 2.07 |
| | Difficulty Weighted Difference Ratio (%) ↑ | 50.85 | **56.22** | 56.20 | 14.68 | 37.66 | **41.22** |
| | Difficulty Weighted Relative Difference Ratio (%) ↑ | 50.85 | **56.22** | 56.20 | 37.62 | 53.13 | **57.79** |
| | Valid Output Grid ↑ | 100.00 | 100.00 | 99.00 | 99.00 | 100.00 | 100.00 |
| | Load-Support Connectivity (%) ↑ | 100.00 | 100.00 | 99.00 | 97.00 | 99.00 | 97.00 |
| | Load-Support Directional Connectivity (%) ↑ | 100.00 | 100.00 | 99.00 | 97.00 | 99.00 | 97.00 |
| | Average Isolated Clusters Count ↓ | 0.00 | 0.00 | 0.00 | 0.00 | 0.00 | 0.00 |
| | Force Path Cost Average Efficiency Ratio (%) ↑ | 94.90 | **98.74** | 98.65 | 84.17 | **98.62** | 96.66 |
| 3 Random Columns | Exact Match ↑ | 5 | 5 | **17** | 7 | 8 | 8 |
| | Difference Ratio (%) ↑ | 58.69 | 72.50 | **78.86** | -17.63 | -0.45 | **24.33** |
| | Relative Difference Ratio (%) ↑ | 58.69 | 72.50 | **78.86** | 31.78 | 49.90 | **65.66** |
| | Penalized Difference Ratio (%) ↑ | 27.96 | 49.17 | **65.13** | -43.98 | 7.40 | **31.72** |
| | Average Difficulty Score | 1.88 | 1.93 | **1.96** | 1.90 | **1.96** | 1.88 |
| | Difficulty Weighted Difference Ratio (%) ↑ | 34.52 | 45.21 | **50.03** | -22.99 | -10.19 | **7.45** |
| | Difficulty Weighted Relative Difference Ratio (%) ↑ | 34.52 | 45.21 | **50.03** | 14.44 | 28.98 | **38.82** |
| | Valid Output Grid ↑ | 100.00 | 100.00 | 100.00 | 100.00 | 100.00 | 100.00 |
| | Load-Support Connectivity (%) ↑ | 94.00 | 97.00 | 98.00 | 93.00 | 84.00 | 94.00 |
| | Load-Support Directional Connectivity (%) ↑ | 94.00 | 97.00 | 98.00 | 93.00 | 84.00 | 94.00 |
| | Average Isolated Clusters Count ↓ | 0.00 | 0.00 | 0.00 | 0.00 | 0.04 | 0.00 |
| | Force Path Cost Average Efficiency Ratio (%) ↑ | 88.28 | 95.82 | **96.94** | 77.05 | 79.48 | **92.20** |
| Full | Exact Match ↑ | 0 | 0 | **4** | 0 | 0 | 0 |
| | Difference Ratio (%) ↑ | -35.78 | -14.18 | **23.06** | -466.42 | -292.37 | **-240.02** |
| | Relative Difference Ratio (%) ↑ | -35.78 | -14.18 | **23.06** | -177.48 | -27.72 | **-4.41** |
| | Penalized Difference Ratio (%) ↑ | -35.78 | -14.18 | **23.06** | -177.48 | -27.72 | **-4.41** |
| | Average Difficulty Score | 1.92 | **1.94** | 1.91 | 1.96 | 1.96 | **2.00** |
| | Difficulty Weighted Difference Ratio (%) ↑ | -21.79 | -8.72 | **14.96** | -310.26 | -195.92 | **-169.54** |
| | Difficulty Weighted Relative Difference Ratio (%) ↑ | -21.79 | -8.72 | **14.96** | -117.17 | -19.24 | **-5.04** |
| | Valid Output Grid ↑ | 100.00 | 100.00 | 100.00 | 100.00 | 99.00 | 100.00 |
| | Load-Support Connectivity (%) ↑ | 94.00 | 100.00 | 98.00 | 88.00 | 91.00 | 93.00 |
| | Load-Support Directional Connectivity (%) ↑ | 94.00 | 100.00 | 98.00 | 88.00 | 91.00 | 93.00 |
| | Average Isolated Clusters Count ↓ | 0.00 | 0.00 | 0.00 | 0.00 | 0.00 | 0.00 |
| | Force Path Cost Average Efficiency Ratio (%) ↑ | 90.33 | **97.86** | 96.77 | 87.83 | 90.10 | **91.89** |
| Average | Exact Match ↑ | 32.12 | 38.50 | **48.88** | 28.25 | 36.12 | **40.62** |
| | Difference Ratio (%) ↑ | 71.68 | 77.94 | **84.46** | -10.20 | 21.60 | **32.79** |
| | Relative Difference Ratio (%) ↑ | 71.68 | 77.94 | **84.46** | 45.53 | 71.73 | **77.22** |
| | Penalized Difference Ratio (%) ↑ | 64.16 | 72.48 | **80.97** | 26.27 | 61.05 | **68.19** |
| | Average Difficulty Score | 1.92 | **1.94** | 1.92 | 1.98 | **1.99** | 1.98 |
| | Difficulty Weighted Difference Ratio (%) ↑ | 45.51 | 49.95 | **53.56** | -11.10 | 11.42 | **17.78** |
| | Difficulty Weighted Relative Difference Ratio (%) ↑ | 45.51 | 49.95 | **53.56** | 28.17 | 46.75 | **49.93** |
| | Valid Output Grid ↑ | 100.00 | 100.00 | 99.88 | 99.62 | 99.75 | 100.00 |
| | Load-Support Connectivity (%) ↑ | 98.50 | 99.50 | 99.12 | 99.62 | 96.25 | 97.75 |
| | Load-Support Directional Connectivity (%) ↑ | 98.50 | 99.50 | 99.12 | 99.62 | 96.25 | 97.75 |
| | Average Isolated Clusters Count ↓ | 0.00 | 0.00 | 0.00 | 0.00 | 0.01 | 0.00 |
| | Force Path Cost Average Efficiency Ratio (%) ↑ | 96.47 | **98.72** | 98.63 | 91.79 | 95.40 | **97.22** |

# I  ADDITIONAL PHYSICS-ENHANCED AND -NEUTRAL EXPERIMENT RESULTS

## I.1  PROMPT AND COMPLETION - 1 RANDOM CELL EASY: PHYSICS-NETURAL

---

**Prompt and Completion - 1 Random Cell Easy: Physics-Netural**

**Prompt:**
You are given a grid of cells.  Each cell can have one of the following states:
- 'L' indicates a special marker.
- 'V' indicates an empty cell.
- 'S' indicates a fixed marker.

The goal is to predict the correct grid configuration by filling in all 'V' cells with
either '1' (solid) or '0' (empty), based on the surrounding structure and general
consistency rules.

Relevant rules for reasoning:
- Filled cells should form continuous connections between relevant markers when necessary.
- Avoid placing isolated filled cells that are not connected to anything meaningful.
- Minimize the number of filled cells while maintaining a coherent and connected
structure.

Below is the input grid with masked regions:

```
L L L L L 0 0 0 0 0
0 1 1 1 0 0 0 0 0 0
0 1 1 1 0 0 0 0 0 0
0 0 1 0 0 0 0 0 0 0
0 0 1 0 0 0 0 0 0 0
0 0 1 0 0 0 0 0 0 0
0 0 1 0 0 0 0 0 0 0
0 0 1 0 0 0 0 0 0 0
V 0 1 1 1 0 0 0 0 0
0 0 1 S S S 0 0 0 0
```

Please output the completed grid by replacing all 'V' cells with either '1' (solid) or '0'
(empty).
Maintain the same format as the input:  one row per line, cells separated by spaces, and
the total number of rows and columns unchanged.
Return only the completed grid without any additional explanation.

**Model Completion (differences in red):**
```
L L L L L 0 0 0 0 0
0 1 1 1 0 0 0 0 0 0
0 1 1 1 0 0 0 0 0 0
0 0 1 0 0 0 0 0 0 0
0 0 1 0 0 0 0 0 0 0
0 0 1 0 0 0 0 0 0 0
0 0 1 0 0 0 0 0 0 0
0 0 1 0 0 0 0 0 0 0
0 0 1 1 1 0 0 0 0 0
0 0 1 S S S 0 0 0 0
```

**Ground Truth:**
```
L L L L L 0 0 0 0 0
0 1 1 1 0 0 0 0 0 0
0 1 1 1 0 0 0 0 0 0
0 0 1 0 0 0 0 0 0 0
0 0 1 0 0 0 0 0 0 0
0 0 1 0 0 0 0 0 0 0
0 0 1 0 0 0 0 0 0 0
0 0 1 0 0 0 0 0 0 0
0 0 1 1 1 0 0 0 0 0
0 0 1 S S S 0 0 0 0
```

---

## I.2 PROMPT AND COMPLETION - 1 RANDOM CELL EASY: PHYSICS-ENHANCED

---

**Prompt and Completion - 1 Random Cell Easy: Physics-Enhanced**

**Prompt:**
You are given a structural material distribution represented as a grid. Each cell can
have one of the following states:
- 'L' indicates applied load.
- 'V' indicates void.
- 'S' indicates support.

The goal is to predict the correct material distribution by filling in all 'V' cells
with either '1' (solid) or '0' (empty), based on the surrounding structure and implicit
physical reasoning (such as load paths, supports, and forces).

Relevant physical knowledge for reasoning:
- Loads ('L') create forces that must be transferred through continuous material paths to
supports ('S').
- Stress follows the shortest stiff path from loads to supports.
- Any material cell that is disconnected from both loads and supports does not carry force
and should be avoided.
- Structures should satisfy equilibrium: all loads must eventually be resisted by at
least one support.
- Use as little material as possible while still maintaining at least one valid load path.

Below is the input grid with masked regions:

```
0 0 L L L 0 0 0 0 0
0 0 0 1 0 0 0 0 0 0
0 0 0 V 0 0 0 0 0 0
0 0 0 1 0 0 0 0 0 0
0 0 0 1 0 0 0 0 0 0
0 0 0 1 0 0 0 0 0 0
0 0 1 1 1 0 0 0 0 0
0 0 1 1 1 0 0 0 0 0
0 0 1 1 1 0 0 0 0 0
S S S S S S S S S 0
```

Please output the completed grid by replacing all 'V' cells with either '1' (solid) or '0'
(empty).
Maintain the same format as the input: one row per line, cells separated by spaces, and
the total number of rows and columns unchanged.
Return only the completed grid without any additional explanation.

**Model Completion (differences in red):**
```
0 0 L L L 0 0 0 0 0
0 0 0 1 0 0 0 0 0 0
0 0 0 1 0 0 0 0 0 0
0 0 0 1 0 0 0 0 0 0
0 0 0 1 0 0 0 0 0 0
0 0 0 1 0 0 0 0 0 0
0 0 1 1 1 0 0 0 0 0
0 0 1 1 1 0 0 0 0 0
0 0 1 1 1 0 0 0 0 0
S S S S S S S S S 0
```

**Ground Truth:**
```
0 0 L L L 0 0 0 0 0
0 0 0 1 0 0 0 0 0 0
0 0 0 1 0 0 0 0 0 0
0 0 0 1 0 0 0 0 0 0
0 0 0 1 0 0 0 0 0 0
0 0 0 1 0 0 0 0 0 0
0 0 1 1 1 0 0 0 0 0
0 0 1 1 1 0 0 0 0 0
0 0 1 1 1 0 0 0 0 0
S S S S S S S S S 0
```

---

## I.3 RESULTS FOR CLAUDE OPUS 4 ON ALL TASKS

Physics Enhanced and Neutral Prompt Comparison Evaluation Results

**Reconstruction Accuracy Metrics 1/2**

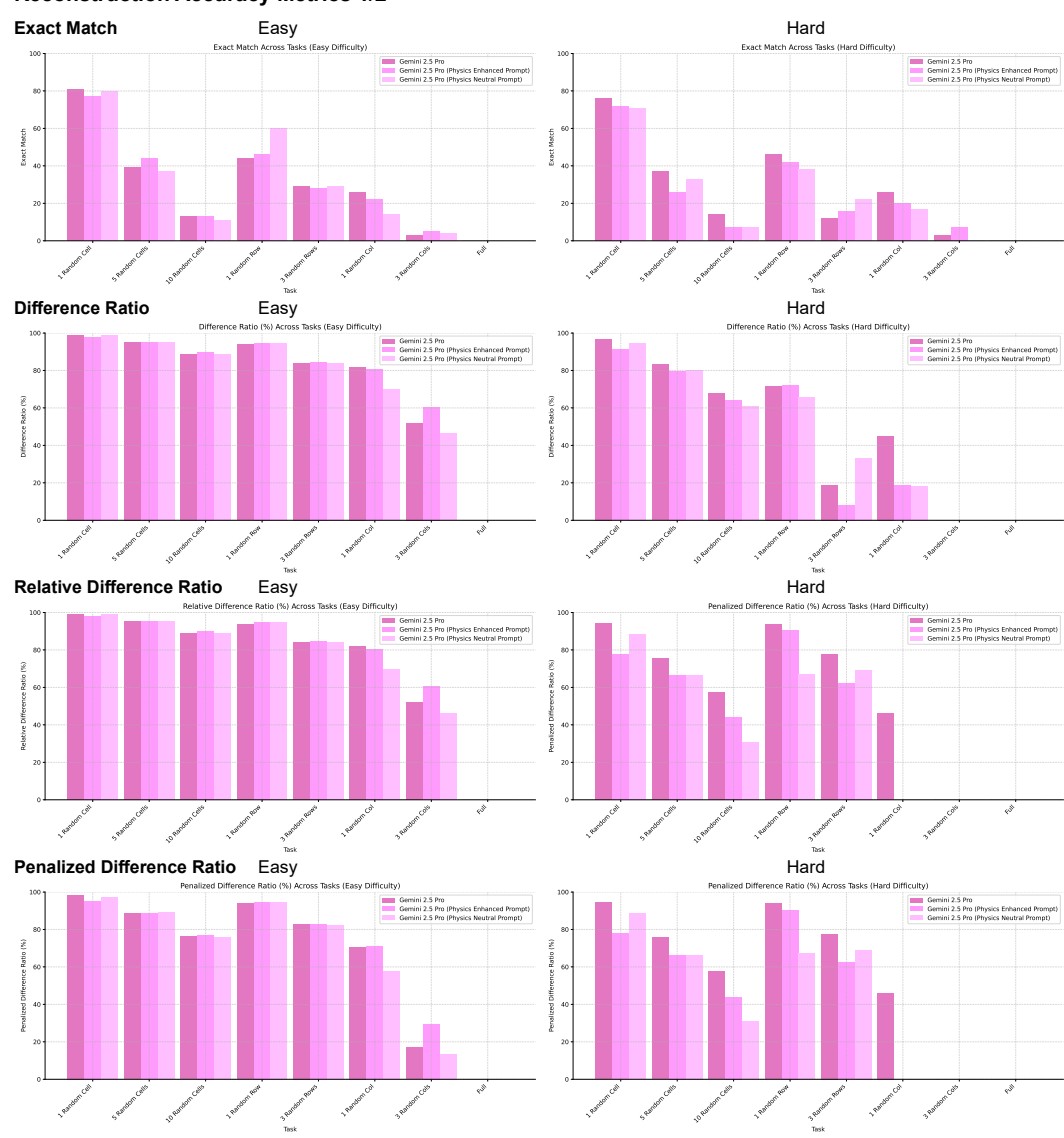

Figure 17: Physics-Enhanced and -Neutral evaluation run metric: Exact Match, Difference Ratio, Relative Difference Ratio and Penalized Difference Ratio for Claude Opus 4, for all tasks, easy and hard difficulty.

Physics Enhanced and Neutral Prompt Comparison Evaluation Results

**Reconstruction Accuracy Metrics 2/2**

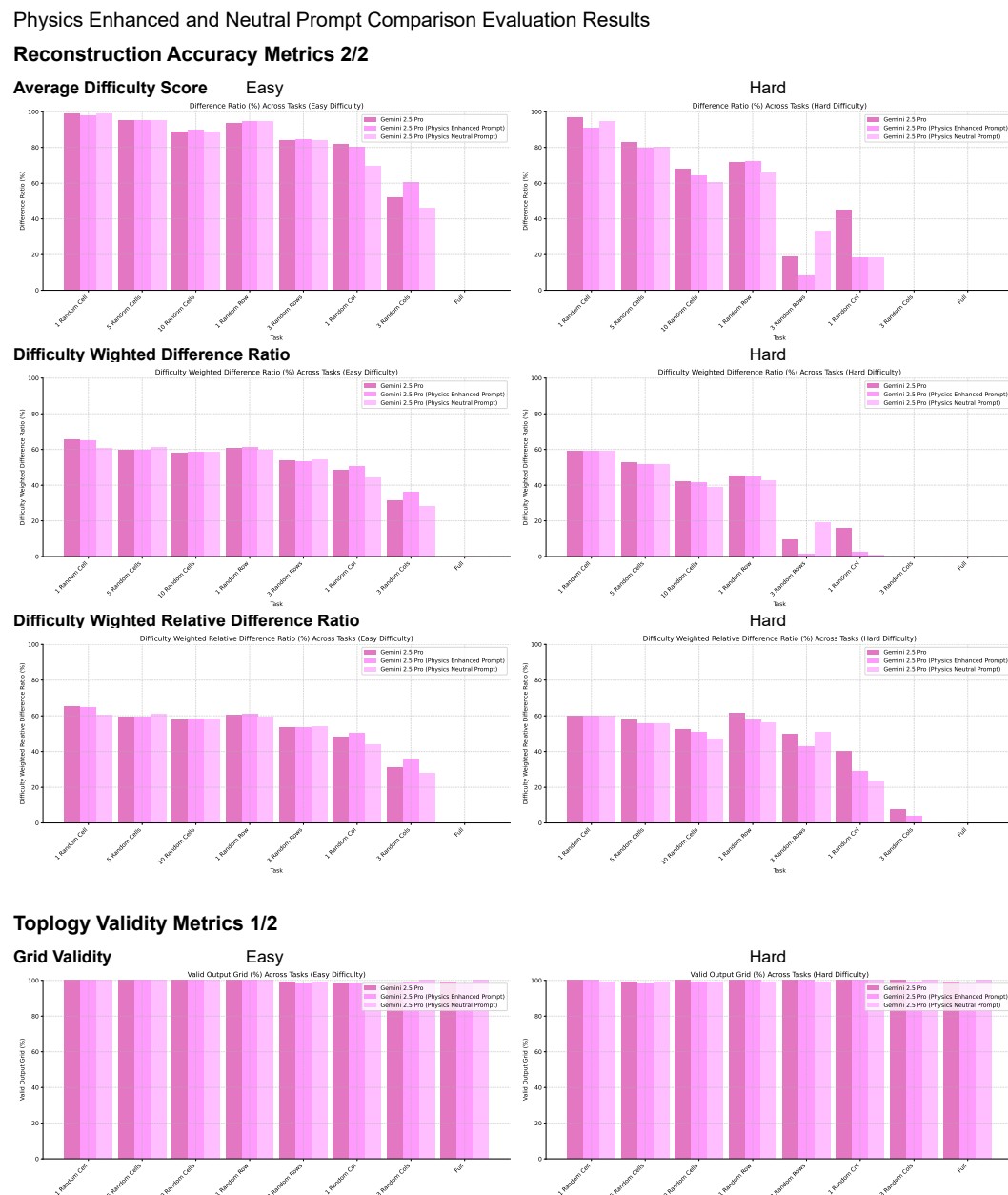

Figure 18: Physics-Enhanced and -Neutral evaluation run metric: Average Difficulty Score, Difficulty Weighted Difference Ratio, Difficulty Weighted Relative Difference Ratio and Grid Validity for Claude Opus 4, for all tasks, easy and hard difficulty.

Physics Enhanced and Neutral Prompt Comparison Evaluation Results

**Topology Validity Metrics 2/2**

**Load-Support Connectivity**   Easy

**Directional Load-Support Connectivity**   Hard

**Average Isolated Cluster Count**   Hard

**Physics-Approximating Metrics 1/1**

**Force Path Cost Average Efficiency Ratio**   Easy   Hard

Figure 19: Physics-Enhanced and -Neutral evaluation result: Load-Support Connectivity, Directional Load-Support Connectivity, Average Isolated Cluster Count and Force Path Cost Average Efficiency Ratio for Claude Opus 4, for all tasks, easy and hard difficulty.

Table 7: Physics-Enhanced and -Neutral evaluation results for all metrics, for Claude Opus 4, for all tasks, easy and hard difficulty.

| Task | Metric | Easy | | | Hard | | |
|---|---|---|---|---|---|---|---|
| | | Gemini 2.5 Pro (Base) | Gemini 2.5 Pro (Physics-Enhanced Prompt) | Gemini 2.5 Pro (Physics-Neutral Prompt) | Gemini 2.5 Pro (Base) | Gemini 2.5 Pro (Physics-Enhanced Prompt) | Gemini 2.5 Pro (Physics-Neutral Prompt) |
| 1 Random Cell | Exact Match ↑ | 81 | 77 | 80 | 76 | 72 | 71 |
| | Difference Ratio (%) ↑ | 99.03 | 97.87 | 98.93 | 96.70 | 91.20 | 94.52 |
| | Relative Difference Ratio (%) ↑ | 99.03 | 97.87 | 98.93 | 97.88 | 92.22 | 95.33 |
| | Penalized Difference Ratio (%) ↑ | 98.37 | 95.26 | 97.20 | 94.31 | 77.77 | 88.60 |
| | Average Difficulty Score | 1.99 | 1.85 | 1.85 | 1.86 | 1.96 | 1.92 |
| | Difficulty Weighted Difference Ratio (%) ↑ | 65.47 | 64.69 | 60.78 | 58.87 | 59.00 | 59.30 |
| | Difficulty Weighted Relative Difference Ratio (%) ↑ | 65.47 | 64.69 | 60.78 | 60.01 | 60.03 | 60.11 |
| | Valid Output Grid ↑ | 100.00 | 100.00 | 100.00 | 100.00 | 100.00 | 99.00 |
| | Load-Support Connectivity (%) ↑ | 100.00 | 98.00 | 100.00 | 100.00 | 95.00 | 97.00 |
| | Load-Support Directional Connectivity (%) ↑ | 100.00 | 98.00 | 100.00 | 100.00 | 95.00 | 97.00 |
| | Average Isolated Clusters Count ↓ | 0.00 | 0.02 | 0.00 | 0.01 | 0.04 | 0.01 |
| | Force Path Cost Average Efficiency Ratio (%) ↑ | 99.94 | 97.94 | 99.83 | 99.92 | 94.93 | 96.80 |
| 5 Random Cells | Exact Match ↑ | 39 | 44 | 37 | 37 | 26 | 33 |
| | Difference Ratio (%) ↑ | 95.08 | 95.12 | 95.08 | 83.19 | 79.81 | 80.21 |
| | Relative Difference Ratio (%) ↑ | 95.08 | 95.12 | 95.08 | 89.98 | 85.92 | 85.99 |
| | Penalized Difference Ratio (%) ↑ | 88.59 | 88.62 | 89.15 | 75.62 | 66.34 | 66.29 |
| | Average Difficulty Score | 1.89 | 1.89 | 1.93 | 1.96 | 1.97 | 1.99 |
| | Difficulty Weighted Difference Ratio (%) ↑ | 59.48 | 59.56 | 60.93 | 52.60 | 51.32 | 51.35 |
| | Difficulty Weighted Relative Difference Ratio (%) ↑ | 59.48 | 59.56 | 60.93 | 57.87 | 55.85 | 55.77 |
| | Valid Output Grid ↑ | 100.00 | 100.00 | 100.00 | 99.00 | 98.00 | 97.00 |
| | Load-Support Connectivity (%) ↑ | 100.00 | 100.00 | 100.00 | 98.00 | 96.00 | 97.00 |
| | Load-Support Directional Connectivity (%) ↑ | 100.00 | 100.00 | 100.00 | 98.00 | 96.00 | 97.00 |
| | Average Isolated Clusters Count ↓ | 0.00 | 0.00 | 0.00 | 0.03 | 0.02 | 0.06 |
| | Force Path Cost Average Efficiency Ratio (%) ↑ | 99.75 | 99.75 | 99.73 | 97.84 | 95.55 | 96.83 |
| 10 Random Cells | Exact Match ↑ | 13 | 13 | 11 | 14 | 7 | 7 |
| | Difference Ratio (%) ↑ | 88.88 | 89.86 | 88.77 | 67.83 | 64.28 | 60.77 |
| | Relative Difference Ratio (%) ↑ | 88.88 | 89.86 | 88.77 | 82.80 | 77.50 | 72.96 |
| | Penalized Difference Ratio (%) ↑ | 76.21 | 77.19 | 75.70 | 57.50 | 44.08 | 30.78 |
| | Average Difficulty Score | 1.97 | 1.97 | 1.99 | 1.94 | 2.01 | 1.97 |
| | Difficulty Weighted Difference Ratio (%) ↑ | 58.06 | 58.69 | 58.53 | 41.99 | 41.53 | 38.61 |
| | Difficulty Weighted Relative Difference Ratio (%) ↑ | 58.06 | 58.69 | 58.53 | 52.66 | 51.13 | 47.43 |
| | Valid Output Grid ↑ | 100.00 | 100.00 | 100.00 | 100.00 | 99.00 | 99.00 |
| | Load-Support Connectivity (%) ↑ | 100.00 | 100.00 | 97.00 | 100.00 | 98.00 | 92.00 |
| | Load-Support Directional Connectivity (%) ↑ | 100.00 | 100.00 | 97.00 | 100.00 | 98.00 | 92.00 |
| | Average Isolated Clusters Count ↓ | 0.00 | 0.00 | 0.02 | 0.02 | 0.01 | 0.07 |
| | Force Path Cost Average Efficiency Ratio (%) ↑ | 99.31 | 99.34 | 96.19 | 99.34 | 97.12 | 91.07 |
| 1 Random Row | Exact Match ↑ | 44 | 46 | 60 | 46 | 42 | 38 |
| | Difference Ratio (%) ↑ | 93.90 | 94.56 | 94.59 | 71.69 | 72.25 | 65.84 |
| | Relative Difference Ratio (%) ↑ | 93.90 | 94.56 | 94.59 | 93.86 | 90.55 | 84.59 |
| | Penalized Difference Ratio (%) ↑ | 93.90 | 94.56 | 94.59 | 93.86 | 90.55 | 67.23 |
| | Average Difficulty Score | 1.94 | 1.94 | 1.89 | 1.99 | 1.92 | 1.99 |
| | Difficulty Weighted Difference Ratio (%) ↑ | 60.55 | 60.98 | 59.59 | 45.14 | 44.52 | 42.47 |
| | Difficulty Weighted Relative Difference Ratio (%) ↑ | 60.55 | 60.98 | 59.59 | 61.91 | 57.70 | 56.45 |
| | Valid Output Grid ↑ | 100.00 | 100.00 | 100.00 | 100.00 | 100.00 | 99.00 |
| | Load-Support Connectivity (%) ↑ | 100.00 | 100.00 | 99.00 | 100.00 | 99.00 | 92.00 |
| | Load-Support Directional Connectivity (%) ↑ | 100.00 | 100.00 | 99.00 | 100.00 | 99.00 | 92.00 |
| | Average Isolated Clusters Count ↓ | 0.00 | 0.00 | 0.00 | 0.00 | 0.00 | 0.07 |
| | Force Path Cost Average Efficiency Ratio (%) ↑ | 99.99 | 99.94 | 98.65 | 99.99 | 99.00 | 91.96 |
| 3 Random Rows | Exact Match ↑ | 29 | 28 | 29 | 12 | 16 | 22 |
| | Difference Ratio (%) ↑ | 84.09 | 84.46 | 83.85 | 18.75 | 8.05 | 33.21 |
| | Relative Difference Ratio (%) ↑ | 84.09 | 84.46 | 83.85 | 77.42 | 66.67 | 78.96 |
| | Penalized Difference Ratio (%) ↑ | 82.84 | 82.61 | 82.18 | 77.42 | 62.27 | 69.10 |
| | Average Difficulty Score | 1.92 | 1.89 | 1.95 | 1.95 | 1.99 | 1.94 |
| | Difficulty Weighted Difference Ratio (%) ↑ | 53.52 | 53.40 | 54.34 | 9.64 | 1.50 | 19.13 |
| | Difficulty Weighted Relative Difference Ratio (%) ↑ | 53.52 | 53.40 | 54.34 | 49.98 | 43.20 | 50.91 |
| | Valid Output Grid ↑ | 99.00 | 98.00 | 99.00 | 100.00 | 100.00 | 99.00 |
| | Load-Support Connectivity (%) ↑ | 98.00 | 96.00 | 98.00 | 100.00 | 98.00 | 95.00 |
| | Load-Support Directional Connectivity (%) ↑ | 98.00 | 96.00 | 98.00 | 100.00 | 98.00 | 95.00 |
| | Average Isolated Clusters Count ↓ | 0.01 | 0.02 | 0.01 | 0.00 | 0.02 | 0.04 |
| | Force Path Cost Average Efficiency Ratio (%) ↑ | 97.54 | 95.88 | 97.43 | 99.87 | 97.97 | 94.45 |
| 1 Random Column | Exact Match ↑ | 26 | 22 | 14 | 26 | 20 | 17 |
| | Difference Ratio (%) ↑ | 81.95 | 80.45 | 69.86 | 44.93 | 18.64 | 18.41 |
| | Relative Difference Ratio (%) ↑ | 81.95 | 80.45 | 69.86 | 72.53 | 48.66 | 42.16 |
| | Penalized Difference Ratio (%) ↑ | 70.55 | 70.85 | 57.94 | 46.17 | -2.29 | -5.47 |
| | Average Difficulty Score | 1.85 | 1.90 | 1.88 | 1.87 | 2.13 | 2.08 |
| | Difficulty Weighted Difference Ratio (%) ↑ | 48.21 | 50.24 | 43.85 | 15.91 | 2.64 | 0.94 |
| | Difficulty Weighted Relative Difference Ratio (%) ↑ | 48.21 | 50.24 | 43.85 | 40.27 | 29.14 | 22.96 |
| | Valid Output Grid ↑ | 98.00 | 98.00 | 97.00 | 100.00 | 100.00 | 100.00 |
| | Load-Support Connectivity (%) ↑ | 98.00 | 98.00 | 96.00 | 100.00 | 91.00 | 94.00 |
| | Load-Support Directional Connectivity (%) ↑ | 98.00 | 98.00 | 96.00 | 100.00 | 91.00 | 94.00 |
| | Average Isolated Clusters Count ↓ | 0.00 | 0.01 | 0.03 | 0.00 | 0.08 | 0.07 |
| | Force Path Cost Average Efficiency Ratio (%) ↑ | 97.30 | 97.34 | 95.36 | 99.48 | 90.32 | 92.43 |
| 3 Random Columns | Exact Match ↑ | 3 | 5 | 4 | 3 | 7 | 0 |
| | Difference Ratio (%) ↑ | 52.03 | 60.60 | 46.29 | -56.28 | -38.97 | -56.82 |
| | Relative Difference Ratio (%) ↑ | 52.03 | 60.60 | 46.29 | 20.55 | 16.32 | 3.17 |
| | Penalized Difference Ratio (%) ↑ | 17.06 | 29.47 | 13.61 | -60.21 | -68.23 | -87.68 |
| | Average Difficulty Score | 1.90 | 1.88 | 1.88 | 1.96 | 1.90 | 1.95 |
| | Difficulty Weighted Difference Ratio (%) ↑ | 31.14 | 36.09 | 28.05 | -50.66 | -36.28 | -43.02 |
| | Difficulty Weighted Relative Difference Ratio (%) ↑ | 31.14 | 36.09 | 28.05 | 7.72 | 4.18 | 0.15 |
| | Valid Output Grid ↑ | 98.00 | 99.00 | 97.00 | 100.00 | 99.00 | 100.00 |
| | Load-Support Connectivity (%) ↑ | 96.00 | 97.00 | 96.00 | 96.00 | 90.00 | 81.00 |
| | Load-Support Directional Connectivity (%) ↑ | 96.00 | 97.00 | 96.00 | 96.00 | 90.00 | 81.00 |
| | Average Isolated Clusters Count ↓ | 0.02 | 0.01 | 0.02 | 0.02 | 0.04 | 0.14 |
| | Force Path Cost Average Efficiency Ratio (%) ↑ | 92.30 | 95.26 | 92.66 | 94.28 | 86.24 | 70.47 |
| Full | Exact Match ↑ | 0 | 0 | 0 | 0 | 0 | 0 |
| | Difference Ratio (%) ↑ | -25.03 | -39.34 | -16.98 | -548.98 | -577.88 | -439.73 |
| | Relative Difference Ratio (%) ↑ | -25.03 | -39.34 | -16.98 | -316.57 | -358.96 | -282.89 |
| | Penalized Difference Ratio (%) ↑ | -25.96 | -39.34 | -17.62 | -318.95 | -390.30 | -326.53 |
| | Average Difficulty Score | 1.91 | 1.93 | 1.95 | 1.95 | 1.95 | 1.98 |
| | Difficulty Weighted Difference Ratio (%) ↑ | -14.61 | -23.66 | -10.88 | -360.89 | -380.19 | -298.08 |
| | Difficulty Weighted Relative Difference Ratio (%) ↑ | -14.61 | -23.66 | -10.88 | -208.21 | -236.96 | -192.17 |
| | Valid Output Grid ↑ | 99.00 | 98.00 | 98.00 | 99.00 | 98.00 | 100.00 |
| | Load-Support Connectivity (%) ↑ | 98.00 | 98.00 | 98.00 | 98.00 | 87.00 | 83.00 |
| | Load-Support Directional Connectivity (%) ↑ | 98.00 | 98.00 | 98.00 | 98.00 | 87.00 | 83.00 |
| | Average Isolated Clusters Count ↓ | 0.01 | 0.00 | 0.02 | 0.01 | 0.11 | 0.17 |
| | Force Path Cost Average Efficiency Ratio (%) ↑ | 96.85 | 95.30 | 96.63 | 97.83 | 86.29 | 81.43 |
| Average | Exact Match ↑ | 29.38 | 29.38 | 29.38 | 26.75 | 23.75 | 23.50 |
| | Difference Ratio (%) ↑ | 71.24 | 70.45 | 70.05 | -27.77 | -35.33 | -17.95 |
| | Relative Difference Ratio (%) ↑ | 71.24 | 70.45 | 70.05 | 27.31 | 14.86 | 22.53 |
| | Penalized Difference Ratio (%) ↑ | 62.70 | 62.40 | 61.59 | 8.21 | -14.97 | -12.21 |
| | Average Difficulty Score | 1.92 | 1.92 | 1.91 | 1.94 | 1.98 | 1.98 |
| | Difficulty Weighted Difference Ratio (%) ↑ | 45.23 | 45.00 | 44.40 | -23.42 | -26.99 | -16.16 |
| | Difficulty Weighted Relative Difference Ratio (%) ↑ | 45.23 | 45.00 | 44.40 | 15.28 | 8.03 | 12.70 |
| | Valid Output Grid ↑ | 99.25 | 99.12 | 99.50 | 99.75 | 99.25 | 99.38 |
| | Load-Support Connectivity (%) ↑ | 98.75 | 98.38 | 98.00 | 99.00 | 94.25 | 91.38 |
| | Load-Support Directional Connectivity (%) ↑ | 98.75 | 98.38 | 98.00 | 99.00 | 94.25 | 91.38 |
| | Average Isolated Clusters Count ↓ | 0.01 | 0.01 | 0.02 | 0.01 | 0.04 | 0.08 |
| | Force Path Cost Average Efficiency Ratio (%) ↑ | 97.87 | 97.59 | 97.06 | 98.57 | 93.43 | 89.43 |

# J SAMPLE PROMPTS & COMPLETIONS

## J.1 CLAUDE 3.7 SONNET - 1 RANDOM CELL EASY: SAMPLE 1

---

**Claude 3.7 Sonnet - 1 Random Cell Easy: Sample 1**

**Prompt:**
You are given a structural material distribution represented as a grid.  Each cell can
have one of the following states:
- 'L' indicates applied load.
- 'V' indicates void.
- 'S' indicates support.

The goal is to predict the correct material distribution by filling in all 'V' cells
with either '1' (solid) or '0' (empty), based on the surrounding structure and implicit
physical reasoning (such as load paths, supports, and forces).

Important:  The completed structure should use as little material as possible while
remaining stable and plausible for carrying the applied forces.  Minimize material usage
unless necessary for structural support.

Below is the input grid with masked regions:

```
0 0 L L L 0 0 0 0 0
0 0 0 1 0 0 0 0 0 0
0 0 0 V 0 0 0 0 0 0
0 0 0 1 0 0 0 0 0 0
0 0 0 1 0 0 0 0 0 0
0 0 0 1 0 0 0 0 0 0
0 0 1 1 1 0 0 0 0 0
0 0 1 1 1 0 0 0 0 0
0 0 1 1 1 0 0 0 0 0
S S S S S S S S S 0
```

Please output the completed grid by replacing all 'V' cells with either '1' (solid) or '0'
(empty).
Maintain the same format as the input:  one row per line, cells separated by spaces, and
the total number of rows and columns unchanged.
Return only the completed grid without any additional explanation.

**Model Completion (differences in red):**
```
0 0 L L L 0 0 0 0 0
0 0 0 1 0 0 0 0 0 0
0 0 0 1 0 0 0 0 0 0
0 0 0 1 0 0 0 0 0 0
0 0 0 1 0 0 0 0 0 0
0 0 0 1 0 0 0 0 0 0
0 0 1 1 1 0 0 0 0 0
0 0 1 1 1 0 0 0 0 0
0 0 1 1 1 0 0 0 0 0
S S S S S S S S S 0
```

**Ground Truth:**
```
0 0 L L L 0 0 0 0 0
0 0 0 1 0 0 0 0 0 0
0 0 0 1 0 0 0 0 0 0
0 0 0 1 0 0 0 0 0 0
0 0 0 1 0 0 0 0 0 0
0 0 0 1 0 0 0 0 0 0
0 0 1 1 1 0 0 0 0 0
0 0 1 1 1 0 0 0 0 0
0 0 1 1 1 0 0 0 0 0
S S S S S S S S S 0
```

---

## J.2 CLAUDE 3.7 SONNET - 1 RANDOM CELL EASY: SAMPLE 2

---

### Claude 3.7 Sonnet - 1 Random Cell Easy: Sample 2

**Prompt:**
You are given a structural material distribution represented as a grid. Each cell can
have one of the following states:
- 'L' indicates applied load.
- 'V' indicates void.
- 'S' indicates support.

The goal is to predict the correct material distribution by filling in all 'V' cells
with either '1' (solid) or '0' (empty), based on the surrounding structure and implicit
physical reasoning (such as load paths, supports, and forces).

Important: The completed structure should use as little material as possible while
remaining stable and plausible for carrying the applied forces. Minimize material usage
unless necessary for structural support.

Below is the input grid with masked regions:

```
0 0 0 0 0 0 L L L 0
0 0 0 0 0 0 0 1 0 0
0 0 0 0 0 0 0 1 0 0
0 0 0 0 0 0 0 1 0 0
0 0 0 0 0 0 0 1 0 0
0 0 0 0 1 1 1 1 0 0
0 0 0 1 1 1 1 1 0 0
0 0 1 1 1 1 0 0 0 0
0 1 1 1 1 1 V 0 0 0
S S S S S 1 0 0 0 0
```

Please output the completed grid by replacing all 'V' cells with either '1' (solid) or '0'
(empty).
Maintain the same format as the input: one row per line, cells separated by spaces, and
the total number of rows and columns unchanged.
Return only the completed grid without any additional explanation.

**Model Completion (differences in red):**
```
0 0 0 0 0 0 L L L 0
0 0 0 0 0 0 0 1 0 0
0 0 0 0 0 0 0 1 0 0
0 0 0 0 0 0 0 1 0 0
0 0 0 0 0 0 0 1 0 0
0 0 0 0 1 1 1 1 0 0
0 0 0 1 1 1 1 1 0 0
0 0 1 1 1 1 0 0 0 0
0 1 1 1 1 1 1 0 0 0
S S S S S 1 0 0 0 0
```

**Ground Truth:**
```
0 0 0 0 0 0 L L L 0
0 0 0 0 0 0 0 1 0 0
0 0 0 0 0 0 0 1 0 0
0 0 0 0 0 0 0 1 0 0
0 0 0 0 0 0 0 1 0 0
0 0 0 0 1 1 1 1 0 0
0 0 0 1 1 1 1 1 0 0
0 0 1 1 1 1 0 0 0 0
0 1 1 1 1 1 0 0 0 0
S S S S S 1 0 0 0 0
```

## J.3 CLAUDE 3.7 SONNET - 1 RANDOM CELL HARD

---

**Claude 3.7 Sonnet - 1 Random Cell Hard**

**Prompt:**
You are given a structural material distribution represented as a grid. Each cell can
have one of the following states:
– 'L' indicates applied load.
– 'V' indicates void.
– 'S' indicates support.

The goal is to predict the correct material distribution by filling in all 'V' cells with
a floating point number between 0 and 1, with one decimal place (e.g., 0.0, 0.1, 0.2, ...,
1.0), based on the surrounding structure and implicit physical reasoning (such as load
paths, supports, and forces).

Important: The completed structure should use as little material as possible while
remaining stable and plausible for carrying the applied forces. Minimize material usage
unless necessary for structural support.

Below is the input grid with masked regions:

```
0.0 0.0 0.0 L   L   L   0.0 0.0 0.0 0.0
0.0 0.0 0.0 0.0 0.5 0.0 0.0 0.0 0.0 0.0
0.0 0.0 0.0 0.0 0.5 0.0 0.0 0.0 0.0 0.0
0.0 0.0 0.0 0.0 0.5 0.0 0.0 0.0 0.0 0.0
0.0 0.0 0.0 0.0 0.5 0.0 0.0 0.0 0.0 0.0
0.0 0.0 V   0.0 0.5 0.0 0.0 0.0 0.0 0.0
0.0 0.0 0.0 0.0 0.6 0.0 0.0 0.0 0.0 0.0
0.0 0.1 0.6 0.6 0.6 0.0 0.0 0.0 0.0 0.0
0.0 0.7 0.2 0.8 0.0 0.0 0.0 0.0 0.0 0.0
S   S   S   0.1 0.0 0.0 0.0 0.0 0.0 0.0
```

Please output the completed grid by replacing all 'V' cells with a floating point number
between 0 and 1, with one decimal place (e.g., 0.0, 0.1, 0.2, ..., 1.0).
Maintain the same format as the input: one row per line, cells separated by spaces, and
the total number of rows and columns unchanged.
Return only the completed grid without any additional explanation.

**Model Completion (differences in red):**
```
0.0 0.0 0.0 L   L   L   0.0 0.0 0.0 0.0
0.0 0.0 0.0 0.0 0.5 0.0 0.0 0.0 0.0 0.0
0.0 0.0 0.0 0.0 0.5 0.0 0.0 0.0 0.0 0.0
0.0 0.0 0.0 0.0 0.5 0.0 0.0 0.0 0.0 0.0
0.0 0.0 0.0 0.0 0.5 0.0 0.0 0.0 0.0 0.0
0.0 0.0 0.3 0.0 0.5 0.0 0.0 0.0 0.0 0.0
0.0 0.0 0.0 0.0 0.6 0.0 0.0 0.0 0.0 0.0
0.0 0.1 0.6 0.6 0.6 0.0 0.0 0.0 0.0 0.0
0.0 0.7 0.2 0.8 0.0 0.0 0.0 0.0 0.0 0.0
S   S   S   0.1 0.0 0.0 0.0 0.0 0.0 0.0
```

**Ground Truth:**
```
0.0 0.0 0.0 L   L   L   0.0 0.0 0.0 0.0
0.0 0.0 0.0 0.0 0.5 0.0 0.0 0.0 0.0 0.0
0.0 0.0 0.0 0.0 0.5 0.0 0.0 0.0 0.0 0.0
0.0 0.0 0.0 0.0 0.5 0.0 0.0 0.0 0.0 0.0
0.0 0.0 0.0 0.0 0.5 0.0 0.0 0.0 0.0 0.0
0.0 0.0 0.0 0.0 0.5 0.0 0.0 0.0 0.0 0.0
0.0 0.0 0.0 0.0 0.6 0.0 0.0 0.0 0.0 0.0
0.0 0.1 0.6 0.6 0.6 0.0 0.0 0.0 0.0 0.0
0.0 0.7 0.2 0.8 0.0 0.0 0.0 0.0 0.0 0.0
S   S   S   0.1 0.0 0.0 0.0 0.0 0.0 0.0
```

## J.4 CLAUDE 3.7 SONNET - FULL EASY

**Claude 3.7 Sonnet - Full Easy**

**Prompt:**
You are given a structural material distribution represented as a grid. Each cell can
have one of the following states:
– 'L' indicates applied load.
– 'V' indicates void.
– 'S' indicates support.

The goal is to predict the correct material distribution by filling in all 'V' cells
with either '1' (solid) or '0' (empty), based on the surrounding structure and implicit
physical reasoning (such as load paths, supports, and forces).

Important: The completed structure should use as little material as possible while
remaining stable and plausible for carrying the applied forces. Minimize material usage
unless necessary for structural support.

Below is the input grid with masked regions:

```
0 L L L L 0 0 0 0
V V V V V V V V V
V V V V V V V V V
V V V V V V V V V
V V V V V V V V V
V V V V V V V V V
V V V V V V V V V
V V V V V V V V V
V V V V V V V V V
0 S S S S S 0 0 0
```

Please output the completed grid by replacing all 'V' cells with either '1' (solid) or '0'
(empty).
Maintain the same format as the input: one row per line, cells separated by spaces, and
the total number of rows and columns unchanged.
Return only the completed grid without any additional explanation.

**Model Completion (differences in red):**
```
0 L L L L 0 0 0 0
0 1 1 1 1 0 0 0 0
0 1 0 0 0 1 0 0 0
0 1 0 0 0 1 0 0 0
0 1 0 0 0 1 0 0 0
0 1 0 0 0 1 0 0 0
0 1 0 0 0 1 0 0 0
0 1 0 0 0 1 0 0 0
0 1 1 1 1 1 0 0 0
0 S S S S S 0 0 0
```

**Ground Truth:**
```
0 L L L L 0 0 0 0
0 0 1 1 1 0 0 0 0
0 0 1 1 1 0 0 0 0
0 0 0 1 0 0 0 0 0
0 0 0 1 0 0 0 0 0
0 0 0 1 0 0 0 0 0
0 0 0 1 0 0 0 0 0
0 0 0 1 0 0 0 0 0
0 0 0 1 0 0 0 0 0
0 S S S S S 0 0 0
```

## J.5 CLAUDE 3.7 SONNET - FULL HARD

### Claude 3.7 Sonnet - Full Hard

**Prompt:**
You are given a structural material distribution represented as a grid. Each cell can
have one of the following states:
- 'L' indicates applied load.
- 'V' indicates void.
- 'S' indicates support.

The goal is to predict the correct material distribution by filling in all 'V' cells with
a floating point number between 0 and 1, with one decimal place (e.g., 0.0, 0.1, 0.2, ...,
1.0), based on the surrounding structure and implicit physical reasoning (such as load
paths, supports, and forces).

Important: The completed structure should use as little material as possible while
remaining stable and plausible for carrying the applied forces. Minimize material usage
unless necessary for structural support.

Below is the input grid with masked regions:

```
0.0 L   L L L L L L   L   L
V   V   V V V V V V   V   V
V   V   V V V V V V   V   V
V   V   V V V V V V   V   V
V   V   V V V V V V   V   V
V   V   V V V V V V   V   V
V   V   V V V V V V   V   V
V   V   V V V V V V   V   V
V   V   V V V V V V   V   V
0.0 0.0 S S S S S 0.0 0.0 0.0
```

Please output the completed grid by replacing all 'V' cells with a floating point number
between 0 and 1, with one decimal place (e.g., 0.0, 0.1, 0.2, ..., 1.0).
Maintain the same format as the input: one row per line, cells separated by spaces, and
the total number of rows and columns unchanged.
Return only the completed grid without any additional explanation.

**Model Completion (differences in red):**
```
0.0 L   L   L   L   L   L   L   L
0.0 0.0 0.1 0.2 0.6 0.6 0.2 0.1 0.0 0.0
0.0 0.0 0.2 0.3 0.8 0.8 0.3 0.2 0.0 0.0
0.0 0.0 0.3 0.5 1.0 1.0 0.5 0.3 0.0 0.0
0.0 0.0 0.4 0.7 1.0 1.0 0.7 0.4 0.0 0.0
0.0 0.0 0.5 0.9 1.0 1.0 0.9 0.5 0.0 0.0
0.0 0.0 0.6 1.0 1.0 1.0 1.0 0.6 0.0 0.0
0.0 0.0 0.7 1.0 1.0 1.0 1.0 0.7 0.0 0.0
0.0 0.0 0.9 1.0 1.0 1.0 1.0 0.9 0.0 0.0
0.0 0.0 S   S   S   S   S   0.0 0.0 0.0
```

**Ground Truth:**
```
0.0 L   L   L   L   L   L   L   L
0.0 0.0 0.2 0.3 0.0 0.3 0.0 0.3 0.2 0.0
0.0 0.0 0.0 0.4 0.2 0.3 0.2 0.4 0.0 0.0
0.0 0.0 0.0 0.2 0.4 0.0 0.4 0.2 0.0 0.0
0.0 0.0 0.0 0.0 0.4 0.0 0.4 0.0 0.0 0.0
0.0 0.0 0.0 0.0 0.2 0.5 0.2 0.0 0.0 0.0
0.0 0.0 0.0 0.0 0.6 0.0 0.0 0.0 0.0 0.0
0.0 0.0 0.0 0.0 0.6 0.0 0.0 0.0 0.0 0.0
0.0 0.0 0.0 0.0 0.6 0.0 0.0 0.0 0.0 0.0
0.0 0.0 S   S   S   S   S   0.0 0.0 0.0
```

## J.6 DEEPSEEK-R1 - 1 RANDOM CELL EASY: SAMPLE 1

---

**DeepSeek-R1 - 1 Random Cell Easy: Sample 1**

**Prompt:**
You are given a structural material distribution represented as a grid. Each cell can
have one of the following states:
- 'L' indicates applied load.
- 'V' indicates void.
- 'S' indicates support.

The goal is to predict the correct material distribution by filling in all 'V' cells
with either '1' (solid) or '0' (empty), based on the surrounding structure and implicit
physical reasoning (such as load paths, supports, and forces).

Important: The completed structure should use as little material as possible while
remaining stable and plausible for carrying the applied forces. Minimize material usage
unless necessary for structural support.

Below is the input grid with masked regions:

```
0 0 L L L 0 0 0 0 0
0 0 0 1 0 0 0 0 0 0
0 0 0 V 0 0 0 0 0 0
0 0 0 1 0 0 0 0 0 0
0 0 0 1 0 0 0 0 0 0
0 0 0 1 0 0 0 0 0 0
0 0 1 1 1 0 0 0 0 0
0 0 1 1 1 0 0 0 0 0
0 0 1 1 1 0 0 0 0 0
S S S S S S S S S 0
```

Please output the completed grid by replacing all 'V' cells with either '1' (solid) or '0'
(empty).
Maintain the same format as the input: one row per line, cells separated by spaces, and
the total number of rows and columns unchanged.
Return only the completed grid without any additional explanation.

**Model Completion (differences in red):**
```
0 0 L L L 0 0 0 0 0
0 0 0 1 0 0 0 0 0 0
0 0 0 1 0 0 0 0 0 0
0 0 0 1 0 0 0 0 0 0
0 0 0 1 0 0 0 0 0 0
0 0 0 1 0 0 0 0 0 0
0 0 1 1 1 0 0 0 0 0
0 0 1 1 1 0 0 0 0 0
0 0 1 1 1 0 0 0 0 0
S S S S S S S S S 0
```

**Ground Truth:**
```
0 0 L L L 0 0 0 0 0
0 0 0 1 0 0 0 0 0 0
0 0 0 1 0 0 0 0 0 0
0 0 0 1 0 0 0 0 0 0
0 0 0 1 0 0 0 0 0 0
0 0 0 1 0 0 0 0 0 0
0 0 1 1 1 0 0 0 0 0
0 0 1 1 1 0 0 0 0 0
0 0 1 1 1 0 0 0 0 0
S S S S S S S S S 0
```

---

## J.7 DEEPSEEK-R1 - 1 RANDOM CELL EASY: SAMPLE 2

### DeepSeek-R1 - 1 Random Cell Easy: Sample 2

**Prompt:**
You are given a structural material distribution represented as a grid. Each cell can have one of the following states:
- 'L' indicates applied load.
- 'V' indicates void.
- 'S' indicates support.

The goal is to predict the correct material distribution by filling in all 'V' cells with either '1' (solid) or '0' (empty), based on the surrounding structure and implicit physical reasoning (such as load paths, supports, and forces).

Important: The completed structure should use as little material as possible while remaining stable and plausible for carrying the applied forces. Minimize material usage unless necessary for structural support.

Below is the input grid with masked regions:

```
L L L L L L 0 0 0 0
0 1 1 1 1 0 0 0 0 0
0 0 1 1 0 0 0 0 0 0
0 0 1 1 0 0 0 0 0 0
0 0 1 1 1 0 0 0 0 0
0 0 1 1 1 1 0 0 0 0
V 0 0 1 1 1 1 0 0 0
0 0 0 1 1 1 1 1 0 0
0 0 0 0 1 1 1 1 0 0
0 0 0 0 0 1 S S S 0
```

Please output the completed grid by replacing all 'V' cells with either '1' (solid) or '0' (empty).
Maintain the same format as the input: one row per line, cells separated by spaces, and the total number of rows and columns unchanged.
Return only the completed grid without any additional explanation.

**Model Completion (differences in red):**

```
L L L L L L 0 0 0 0
0 1 1 1 1 0 0 0 0 0
0 0 1 1 0 0 0 0 0 0
0 0 1 1 0 0 0 0 0 0
0 0 1 1 1 0 0 0 0 0
0 0 1 1 1 1 0 0 0 0
1 0 0 1 1 1 1 0 0 0
0 0 0 1 1 1 1 1 0 0
0 0 0 0 1 1 1 1 0 0
0 0 0 0 0 1 S S S 0
```

**Ground Truth:**

```
L L L L L L 0 0 0 0
0 1 1 1 1 0 0 0 0 0
0 0 1 1 0 0 0 0 0 0
0 0 1 1 0 0 0 0 0 0
0 0 1 1 1 0 0 0 0 0
0 0 1 1 1 1 0 0 0 0
0 0 0 1 1 1 1 0 0 0
0 0 0 1 1 1 1 1 0 0
0 0 0 0 1 1 1 1 0 0
0 0 0 0 0 1 S S S 0
```

## J.8 DEEPSEEK-R1 - 1 RANDOM CELL HARD

---

### DeepSeek-R1 - 1 Random Cell Hard

**Prompt:**
You are given a structural material distribution represented as a grid. Each cell can
have one of the following states:
- 'L' indicates applied load.
- 'V' indicates void.
- 'S' indicates support.

The goal is to predict the correct material distribution by filling in all 'V' cells with
a floating point number between 0 and 1, with one decimal place (e.g., 0.0, 0.1, 0.2, ...,
1.0), based on the surrounding structure and implicit physical reasoning (such as load
paths, supports, and forces).

Important:  The completed structure should use as little material as possible while
remaining stable and plausible for carrying the applied forces.  Minimize material usage
unless necessary for structural support.

Below is the input grid with masked regions:

```
0.0 0.0 0.0  L   L   L   0.0 0.0 0.0 0.0
0.0 0.0 0.0 0.0 0.5 0.0 0.0 0.0 0.0 0.0
0.0 0.0 0.0 0.0 0.5 0.0 0.0 0.0 0.0 0.0
0.0 0.0 0.0 0.0 0.5 0.0 0.0 0.0 0.0 0.0
0.0 0.0 0.0 0.0 0.5 0.0 0.0 0.0 0.0 0.0
0.0 0.0  V   0.0 0.5 0.0 0.0 0.0 0.0 0.0
0.0 0.0 0.0 0.0 0.6 0.0 0.0 0.0 0.0 0.0
0.0 0.1 0.6 0.6 0.6 0.0 0.0 0.0 0.0 0.0
0.0 0.7 0.2 0.8 0.0 0.0 0.0 0.0 0.0 0.0
S   S   S   0.1 0.0 0.0 0.0 0.0 0.0 0.0
```

Please output the completed grid by replacing all 'V' cells with a floating point number
between 0 and 1, with one decimal place (e.g., 0.0, 0.1, 0.2, ..., 1.0).
Maintain the same format as the input:  one row per line, cells separated by spaces, and
the total number of rows and columns unchanged.
Return only the completed grid without any additional explanation.

**Model Completion (differences in red):**
```
0.0 0.0 0.0  L   L   L   0.0 0.0 0.0 0.0
0.0 0.0 0.0 0.0 0.5 0.0 0.0 0.0 0.0 0.0
0.0 0.0 0.0 0.0 0.5 0.0 0.0 0.0 0.0 0.0
0.0 0.0 0.0 0.0 0.5 0.0 0.0 0.0 0.0 0.0
0.0 0.0 0.0 0.0 0.5 0.0 0.0 0.0 0.0 0.0
0.0 0.0 0.3 0.0 0.5 0.0 0.0 0.0 0.0 0.0
0.0 0.0 0.0 0.0 0.6 0.0 0.0 0.0 0.0 0.0
0.0 0.1 0.6 0.6 0.6 0.0 0.0 0.0 0.0 0.0
0.0 0.7 0.2 0.8 0.0 0.0 0.0 0.0 0.0 0.0
S   S   S   0.1 0.0 0.0 0.0 0.0 0.0 0.0
```

**Ground Truth:**
```
0.0 0.0 0.0  L   L   L   0.0 0.0 0.0 0.0
0.0 0.0 0.0 0.0 0.5 0.0 0.0 0.0 0.0 0.0
0.0 0.0 0.0 0.0 0.5 0.0 0.0 0.0 0.0 0.0
0.0 0.0 0.0 0.0 0.5 0.0 0.0 0.0 0.0 0.0
0.0 0.0 0.0 0.0 0.5 0.0 0.0 0.0 0.0 0.0
0.0 0.0 0.0 0.0 0.5 0.0 0.0 0.0 0.0 0.0
0.0 0.0 0.0 0.0 0.6 0.0 0.0 0.0 0.0 0.0
0.0 0.1 0.6 0.6 0.6 0.0 0.0 0.0 0.0 0.0
0.0 0.7 0.2 0.8 0.0 0.0 0.0 0.0 0.0 0.0
S   S   S   0.1 0.0 0.0 0.0 0.0 0.0 0.0
```

---

## J.9 DEEPSEEK-R1 - FULL EASY

### DeepSeek-R1 - Full Easy

**Prompt:**
You are given a structural material distribution represented as a grid. Each cell can have one of the following states:
– 'L' indicates applied load.
– 'V' indicates void.
– 'S' indicates support.

The goal is to predict the correct material distribution by filling in all 'V' cells with either '1' (solid) or '0' (empty), based on the surrounding structure and implicit physical reasoning (such as load paths, supports, and forces).

Important: The completed structure should use as little material as possible while remaining stable and plausible for carrying the applied forces. Minimize material usage unless necessary for structural support.

Below is the input grid with masked regions:

```
0 L L L L 0 0 0 0
V V V V V V V V V
V V V V V V V V V
V V V V V V V V V
V V V V V V V V V
V V V V V V V V V
V V V V V V V V V
V V V V V V V V V
V V V V V V V V V
0 S S S S S 0 0 0
```

Please output the completed grid by replacing all 'V' cells with either '1' (solid) or '0' (empty).
Maintain the same format as the input: one row per line, cells separated by spaces, and the total number of rows and columns unchanged.
Return only the completed grid without any additional explanation.

**Model Completion (differences in red):**

```
0 L L L L 0 0 0 0
1 1 1 1 1 1 1 1 1
1 1 1 1 1 1 1 1 1
1 1 1 1 1 1 1 1 1
1 1 1 1 1 1 1 1 1
1 1 1 1 1 1 1 1 1
1 1 1 1 1 1 1 1 1
1 1 1 1 1 1 1 1 1
1 1 1 1 1 1 1 1 1
0 S S S S S 0 0 0
```

**Ground Truth:**

```
0 L L L L 0 0 0 0
0 0 1 1 1 0 0 0 0
0 0 1 1 1 0 0 0 0
0 0 0 1 0 0 0 0 0
0 0 0 1 0 0 0 0 0
0 0 0 1 0 0 0 0 0
0 0 0 1 0 0 0 0 0
0 0 0 1 0 0 0 0 0
0 0 0 1 0 0 0 0 0
0 S S S S S 0 0 0
```

## J.10 DEEPSEEK-R1 - FULL HARD

---

**DeepSeek-R1 - Full Hard**

**Prompt:**
```
You are given a structural material distribution represented as a grid.  Each cell can
have one of the following states:
- 'L' indicates applied load.
- 'V' indicates void.
- 'S' indicates support.

The goal is to predict the correct material distribution by filling in all 'V' cells with
a floating point number between 0 and 1, with one decimal place (e.g., 0.0, 0.1, 0.2, ...,
1.0), based on the surrounding structure and implicit physical reasoning (such as load
paths, supports, and forces).

Important:  The completed structure should use as little material as possible while
remaining stable and plausible for carrying the applied forces.  Minimize material usage
unless necessary for structural support.

Below is the input grid with masked regions:

0.0 L   L L L L L L   L   L
V   V   V V V V V V   V   V
V   V   V V V V V V   V   V
V   V   V V V V V V   V   V
V   V   V V V V V V   V   V
V   V   V V V V V V   V   V
V   V   V V V V V V   V   V
V   V   V V V V V V   V   V
V   V   V V V V V V   V   V
0.0 0.0 S S S S S 0.0 0.0 0.0

Please output the completed grid by replacing all 'V' cells with a floating point number
between 0 and 1, with one decimal place (e.g., 0.0, 0.1, 0.2, ..., 1.0).
Maintain the same format as the input:  one row per line, cells separated by spaces, and
the total number of rows and columns unchanged.
Return only the completed grid without any additional explanation.
```

**Model Completion (differences in red):**
```
0.0 L   L   L   L   L   L   L   L   L
0.0 0.0 0.0 0.0 0.0 0.0 0.0 0.0 0.0 0.0
0.0 0.0 0.0 0.0 0.0 0.0 0.0 0.0 0.0 0.0
0.0 0.0 0.0 0.0 0.0 0.0 0.0 0.0 0.0 0.0
0.0 0.0 0.0 0.0 0.0 0.0 0.0 0.0 0.0 0.0
0.0 0.0 0.0 0.0 0.0 0.0 0.0 0.0 0.0 0.0
0.0 0.0 0.0 0.0 0.0 0.0 0.0 0.0 0.0 0.0
0.0 0.0 0.0 0.0 0.0 0.0 0.0 0.0 0.0 0.0
0.0 0.0 0.0 0.0 0.0 0.0 0.0 0.0 0.0 0.0
0.0 0.0 S   S   S   S   S   0.0 0.0 0.0
```

**Ground Truth:**
```
0.0 L   L   L   L   L   L   L   L   L
0.0 0.0 0.2 0.3 0.0 0.3 0.0 0.3 0.2 0.0
0.0 0.0 0.0 0.4 0.2 0.3 0.2 0.4 0.0 0.0
0.0 0.0 0.0 0.2 0.4 0.0 0.4 0.2 0.0 0.0
0.0 0.0 0.0 0.0 0.4 0.0 0.4 0.0 0.0 0.0
0.0 0.0 0.0 0.0 0.2 0.5 0.2 0.0 0.0 0.0
0.0 0.0 0.0 0.0 0.0 0.6 0.0 0.0 0.0 0.0
0.0 0.0 0.0 0.0 0.0 0.6 0.0 0.0 0.0 0.0
0.0 0.0 0.0 0.0 0.0 0.6 0.0 0.0 0.0 0.0
0.0 0.0 S   S   S   S   S   0.0 0.0 0.0
```

---

## J.11 GPT-4.1 - 1 RANDOM CELL EASY: SAMPLE 1

---

**GPT-4.1 – 1 Random Cell Easy: Sample 1**

**Prompt:**
You are given a structural material distribution represented as a grid. Each cell can have one of the following states:
– 'L' indicates applied load.
– 'V' indicates void.
– 'S' indicates support.

The goal is to predict the correct material distribution by filling in all 'V' cells with either '1' (solid) or '0' (empty), based on the surrounding structure and implicit physical reasoning (such as load paths, supports, and forces).

Important: The completed structure should use as little material as possible while remaining stable and plausible for carrying the applied forces. Minimize material usage unless necessary for structural support.

Below is the input grid with masked regions:

```
0 0 L L L 0 0 0 0 0
0 0 0 1 0 0 0 0 0 0
0 0 0 V 0 0 0 0 0 0
0 0 0 1 0 0 0 0 0 0
0 0 0 1 0 0 0 0 0 0
0 0 0 1 0 0 0 0 0 0
0 0 1 1 1 0 0 0 0 0
0 0 1 1 1 0 0 0 0 0
0 0 1 1 1 0 0 0 0 0
S S S S S S S S S 0
```

Please output the completed grid by replacing all 'V' cells with either '1' (solid) or '0' (empty).
Maintain the same format as the input: one row per line, cells separated by spaces, and the total number of rows and columns unchanged.
Return only the completed grid without any additional explanation.

**Model Completion (differences in red):**
```
0 0 L L L 0 0 0 0 0
0 0 0 1 0 0 0 0 0 0
0 0 0 1 0 0 0 0 0 0
0 0 0 1 0 0 0 0 0 0
0 0 0 1 0 0 0 0 0 0
0 0 0 1 0 0 0 0 0 0
0 0 1 1 1 0 0 0 0 0
0 0 1 1 1 0 0 0 0 0
0 0 1 1 1 0 0 0 0 0
S S S S S S S S S 0
```

**Ground Truth:**
```
0 0 L L L 0 0 0 0 0
0 0 0 1 0 0 0 0 0 0
0 0 0 1 0 0 0 0 0 0
0 0 0 1 0 0 0 0 0 0
0 0 0 1 0 0 0 0 0 0
0 0 0 1 0 0 0 0 0 0
0 0 1 1 1 0 0 0 0 0
0 0 1 1 1 0 0 0 0 0
0 0 1 1 1 0 0 0 0 0
S S S S S S S S S 0
```

---

## J.12 GPT-4.1 - 1 RANDOM CELL EASY: SAMPLE 2

### GPT-4.1 – 1 Random Cell Easy: Sample 2

**Prompt:**
You are given a structural material distribution represented as a grid. Each cell can
have one of the following states:
– 'L' indicates applied load.
– 'V' indicates void.
– 'S' indicates support.

The goal is to predict the correct material distribution by filling in all 'V' cells
with either '1' (solid) or '0' (empty), based on the surrounding structure and implicit
physical reasoning (such as load paths, supports, and forces).

Important:  The completed structure should use as little material as possible while
remaining stable and plausible for carrying the applied forces.  Minimize material usage
unless necessary for structural support.

Below is the input grid with masked regions:

```
L L L L L L 0 0 0 0
0 1 1 1 1 0 0 0 0 0
0 0 1 1 0 0 0 0 0 0
0 0 1 1 0 0 0 0 0 0
0 0 1 1 1 0 0 0 0 0
0 0 1 1 1 1 0 0 0 0
V 0 0 1 1 1 1 0 0 0
0 0 0 1 1 1 1 1 0 0
0 0 0 0 1 1 1 1 0 0
0 0 0 0 0 1 S S S 0
```

**Model Completion (differences in red):**

```
L L L L L L 0 0 0 0
0 1 1 1 1 0 0 0 0 0
0 0 1 1 0 0 0 0 0 0
0 0 1 1 0 0 0 0 0 0
0 0 1 1 1 0 0 0 0 0
0 0 1 1 1 1 0 0 0 0
1 0 0 1 1 1 1 0 0 0
0 0 0 1 1 1 1 1 0 0
0 0 0 0 1 1 1 1 0 0
0 0 0 0 0 1 S S S 0
```

**Ground Truth:**

```
L L L L L L 0 0 0 0
0 1 1 1 1 0 0 0 0 0
0 0 1 1 0 0 0 0 0 0
0 0 1 1 0 0 0 0 0 0
0 0 1 1 1 0 0 0 0 0
0 0 1 1 1 1 0 0 0 0
0 0 0 1 1 1 1 0 0 0
0 0 0 1 1 1 1 1 0 0
0 0 0 0 1 1 1 1 0 0
0 0 0 0 0 1 S S S 0
```

## J.13  GPT-4.1 - 1 RANDOM CELL HARD

---

### GPT-4.1 – 1 Random Cell Hard

**Prompt:**
You are given a structural material distribution represented as a grid.  Each cell can
have one of the following states:
– 'L' indicates applied load.
– 'V' indicates void.
– 'S' indicates support.

The goal is to predict the correct material distribution by filling in all 'V' cells with
a floating-point number between 0 and 1, with one decimal place (e.g., 0.0, 0.1, ...,
1.0), based on the surrounding structure and implicit physical reasoning (such as load
paths, supports, and forces).

Important:  The completed structure should use as little material as possible while
remaining stable and plausible for carrying the applied forces.

Below is the input grid with masked regions:

```
0.0 0.0 0.0  L    L    L    0.0 0.0 0.0 0.0
0.0 0.0 0.0 0.0  0.5 0.0 0.0 0.0 0.0 0.0
0.0 0.0 0.0 0.0  0.5 0.0 0.0 0.0 0.0 0.0
0.0 0.0 0.0 0.0  0.5 0.0 0.0 0.0 0.0 0.0
0.0 0.0 0.0 0.0  0.5 0.0 0.0 0.0 0.0 0.0
0.0 0.0 V    0.0  0.5 0.0 0.0 0.0 0.0 0.0
0.0 0.0 0.0 0.0  0.6 0.0 0.0 0.0 0.0 0.0
0.0 0.1 0.6 0.6  0.6 0.0 0.0 0.0 0.0 0.0
0.0 0.7 0.2 0.8  0.0 0.0 0.0 0.0 0.0 0.0
S   S   S    0.1 0.0 0.0 0.0 0.0 0.0 0.0
```

Please output the completed grid by replacing all 'V' cells with a floating-point number
between 0 and 1.
Maintain the same format as the input:  one row per line, cells separated by spaces, and
the total number of rows and columns unchanged.
Return only the completed grid without any additional explanation.

**Model Completion (differences in red):**

```
0.0 0.0 0.0  L    L    L    0.0 0.0 0.0 0.0
0.0 0.0 0.0 0.0  0.5 0.0 0.0 0.0 0.0 0.0
0.0 0.0 0.0 0.0  0.5 0.0 0.0 0.0 0.0 0.0
0.0 0.0 0.0 0.0  0.5 0.0 0.0 0.0 0.0 0.0
0.0 0.0 0.0 0.0  0.5 0.0 0.0 0.0 0.0 0.0
0.0 0.0 0.3 0.0  0.5 0.0 0.0 0.0 0.0 0.0
0.0 0.0 0.0 0.0  0.6 0.0 0.0 0.0 0.0 0.0
0.0 0.1 0.6 0.6  0.6 0.0 0.0 0.0 0.0 0.0
0.0 0.7 0.2 0.8  0.0 0.0 0.0 0.0 0.0 0.0
S   S   S    0.1 0.0 0.0 0.0 0.0 0.0 0.0
```

**Ground Truth:**

```
0.0 0.0 0.0  L    L    L    0.0 0.0 0.0 0.0
0.0 0.0 0.0 0.0  0.5 0.0 0.0 0.0 0.0 0.0
0.0 0.0 0.0 0.0  0.5 0.0 0.0 0.0 0.0 0.0
0.0 0.0 0.0 0.0  0.5 0.0 0.0 0.0 0.0 0.0
0.0 0.0 0.0 0.0  0.5 0.0 0.0 0.0 0.0 0.0
0.0 0.0 0.0 0.0  0.5 0.0 0.0 0.0 0.0 0.0
0.0 0.0 0.0 0.0  0.6 0.0 0.0 0.0 0.0 0.0
0.0 0.1 0.6 0.6  0.6 0.0 0.0 0.0 0.0 0.0
0.0 0.7 0.2 0.8  0.0 0.0 0.0 0.0 0.0 0.0
S   S   S    0.1 0.0 0.0 0.0 0.0 0.0 0.0
```

---

## J.14   GPT-4.1 - FULL EASY

---

**GPT-4.1 – Full Easy**

**Prompt:**
You are given a structural material distribution represented as a grid.  Each cell can
have one of the following states:
- 'L' indicates applied load.
- 'V' indicates void.
- 'S' indicates support.

The goal is to predict the correct material distribution by filling in all 'V' cells
with either '1' (solid) or '0' (empty), based on the surrounding structure and implicit
physical reasoning (such as load paths, supports, and forces).

Important:  The completed structure should use as little material as possible while
remaining stable and plausible for carrying the applied forces.

Below is the input grid with masked regions:

```
0 L L L L L 0 0 0 0
V V V V V V V V V V
V V V V V V V V V V
V V V V V V V V V V
V V V V V V V V V V
V V V V V V V V V V
V V V V V V V V V V
V V V V V V V V V V
V V V V V V V V V V
0 S S S S S S 0 0 0
```

Please output the completed grid by replacing all 'V' cells with either '1' (solid) or '0'
(empty).
Maintain the same format as the input:  one row per line, cells separated by spaces, and
the total number of rows and columns unchanged.
Return only the completed grid without any additional explanation.

**Model Completion (differences in red):**
```
0 L L L L L 0 0 0 0
1 1 1 1 1 1 1 0 0 0
1 0 0 0 0 1 0 0 0
1 0 0 0 0 0 1 0 0 0
1 0 0 0 0 0 1 0 0 0
1 0 0 0 0 0 1 0 0 0
1 0 0 0 0 0 1 0 0 0
1 0 0 0 0 0 1 0 0 0
1 1 1 1 1 1 1 0 0 0
0 S S S S S S 0 0 0
```

**Ground Truth:**
```
0 L L L L L 0 0 0 0
0 0 1 1 1 0 0 0 0 0
0 0 1 1 1 0 0 0 0 0
0 0 0 1 0 0 0 0 0 0
0 0 0 1 0 0 0 0 0 0
0 0 0 1 0 0 0 0 0 0
0 0 0 1 0 0 0 0 0 0
0 0 0 1 0 0 0 0 0 0
0 0 0 1 0 0 0 0 0 0
0 S S S S S S 0 0 0
```

---

## J.15 GPT-4.1 - FULL HARD

---

**GPT-4.1 – Full Hard**

**Prompt:**
You are given a structural material distribution represented as a grid. Each cell can have one of the following states:
– 'L' indicates applied load.
– 'V' indicates void.
– 'S' indicates support.

The goal is to predict the correct material distribution by filling in all 'V' cells with a floating-point number between 0 and 1, with one decimal place (e.g., 0.0, 0.1, ..., 1.0), based on the surrounding structure and implicit physical reasoning (such as load paths, supports, and forces).

Important: The completed structure should use as little material as possible while remaining stable and plausible for carrying the applied forces.

Below is the input grid with masked regions:

```
0.0 L   L L L L L   L   L
V   V   V V V V V   V   V
V   V   V V V V V   V   V
V   V   V V V V V   V   V
V   V   V V V V V   V   V
V   V   V V V V V   V   V
V   V   V V V V V   V   V
V   V   V V V V V   V   V
V   V   V V V V V   V   V
0.0 0.0 S S S S 0.0 0.0 0.0
```

Please output the completed grid by replacing all 'V' cells with a floating-point number between 0 and 1, with one decimal place.
Maintain the same format as the input: one row per line, cells separated by spaces, and the total number of rows and columns unchanged.
Return only the completed grid without any additional explanation.

**Model Completion (differences in red):**
```
0.0 L   L   L   L   L   L   L   L
0.0 0.4 0.6 0.8 1.0 1.0 0.8 0.6 0.4 0.0
0.0 0.3 0.5 0.7 0.9 0.9 0.7 0.5 0.3 0.0
0.0 0.2 0.4 0.6 0.8 0.8 0.6 0.4 0.2 0.0
0.0 0.1 0.3 0.5 0.7 0.7 0.5 0.3 0.1 0.0
0.0 0.1 0.2 0.4 0.6 0.6 0.4 0.2 0.1 0.0
0.0 0.1 0.2 0.3 0.5 0.5 0.3 0.2 0.1 0.0
0.0 0.1 0.2 0.2 0.4 0.4 0.2 0.2 0.1 0.0
0.0 0.1 0.1 0.1 0.3 0.3 0.1 0.1 0.1 0.0
0.0 0.0 S   S   S   S   S   0.0 0.0 0.0
```

**Ground Truth:**
```
0.0 L   L   L   L   L   L   L   L
0.0 0.0 0.2 0.3 0.0 0.3 0.0 0.3 0.2 0.0
0.0 0.0 0.0 0.4 0.2 0.3 0.2 0.4 0.0 0.0
0.0 0.0 0.0 0.2 0.4 0.0 0.4 0.2 0.0 0.0
0.0 0.0 0.0 0.0 0.4 0.0 0.4 0.0 0.0 0.0
0.0 0.0 0.0 0.0 0.2 0.5 0.2 0.0 0.0 0.0
0.0 0.0 0.0 0.0 0.0 0.6 0.0 0.0 0.0 0.0
0.0 0.0 0.0 0.0 0.0 0.6 0.0 0.0 0.0 0.0
0.0 0.0 0.0 0.0 0.0 0.6 0.0 0.0 0.0 0.0
0.0 0.0 S   S   S   S   S   0.0 0.0 0.0
```

---

## J.16 GEMINI 2.5 PRO - 1 RANDOM CELL EASY: SAMPLE 1

### Gemini 2.5 Pro – 1 Random Cell Easy: Sample 1

**Prompt:**
```
You are given a structural material distribution represented as a grid.  Each cell can
have one of the following states:
- 'L' indicates applied load.
- 'V' indicates void.
- 'S' indicates support.

The goal is to predict the correct material distribution by filling in all 'V' cells
with either '1' (solid) or '0' (empty), based on the surrounding structure and implicit
physical reasoning (such as load paths, supports, and forces).

Important:  The completed structure should use as little material as possible while
remaining stable and plausible for carrying the applied forces.

Below is the input grid with masked regions:

0 0 L L L 0 0 0 0 0
0 0 0 1 0 0 0 0 0 0
0 0 0 V 0 0 0 0 0 0
0 0 0 1 0 0 0 0 0 0
0 0 0 1 0 0 0 0 0 0
0 0 0 1 0 0 0 0 0 0
0 0 1 1 1 0 0 0 0 0
0 0 1 1 1 0 0 0 0 0
0 0 1 1 1 0 0 0 0 0
S S S S S S S S S 0

Please output the completed grid by replacing all 'V' cells with either '1' (solid) or '0'
(empty).
Maintain the same format as the input:  one row per line, cells separated by spaces, and
the total number of rows and columns unchanged.
Return only the completed grid without any additional explanation.
```

**Model Completion (differences in red):**
```
0 0 L L L 0 0 0 0 0
0 0 0 1 0 0 0 0 0 0
0 0 0 1 0 0 0 0 0 0
0 0 0 1 0 0 0 0 0 0
0 0 0 1 0 0 0 0 0 0
0 0 0 1 0 0 0 0 0 0
0 0 1 1 1 0 0 0 0 0
0 0 1 1 1 0 0 0 0 0
0 0 1 1 1 0 0 0 0 0
```

**Ground Truth:**
```
0 0 L L L 0 0 0 0 0
0 0 0 1 0 0 0 0 0 0
0 0 0 1 0 0 0 0 0 0
0 0 0 1 0 0 0 0 0 0
0 0 0 1 0 0 0 0 0 0
0 0 0 1 0 0 0 0 0 0
0 0 1 1 1 0 0 0 0 0
0 0 1 1 1 0 0 0 0 0
0 0 1 1 1 0 0 0 0 0
S S S S S S S S S 0
```

## J.17 GEMINI 2.5 PRO - 1 RANDOM CELL EASY: SAMPLE 2

### Gemini 2.5 Pro – 1 Random Cell Easy: Sample 2

**Prompt:**
You are given a structural material distribution represented as a grid. Each cell can
have one of the following states:
- 'L' indicates applied load.
- 'V' indicates void.
- 'S' indicates support.

The goal is to predict the correct material distribution by filling in all 'V' cells
with either '1' (solid) or '0' (empty), based on the surrounding structure and implicit
physical reasoning (such as load paths, supports, and forces).

Important: The completed structure should use as little material as possible while
remaining stable and plausible for carrying the applied forces.

Below is the input grid with masked regions:

```
L L L L L L 0 0 0 0
0 1 1 1 1 0 0 0 0 0
0 0 1 1 0 0 0 0 0 0
0 0 1 1 0 0 0 0 0 0
0 0 1 1 1 0 0 0 0 0
0 0 1 1 1 1 0 0 0 0
V 0 0 1 1 1 1 0 0 0
0 0 0 1 1 1 1 1 0 0
0 0 0 0 1 1 1 1 0 0
0 0 0 0 0 1 S S S 0
```

Please output the completed grid by replacing all 'V' cells with either '1' (solid) or '0'
(empty).
Maintain the same format as the input: one row per line, cells separated by spaces, and
the total number of rows and columns unchanged.
Return only the completed grid without any additional explanation.

**Model Completion (differences in red):**
```
L L L L L L 0 0 0 0
0 1 1 1 1 0 0 0 0 0
0 0 1 1 0 0 0 0 0 0
0 0 1 1 0 0 0 0 0 0
0 0 1 1 1 0 0 0 0 0
0 0 1 1 1 1 0 0 0 0
1 0 0 1 1 1 1 0 0 0
0 0 0 1 1 1 1 1 0 0
0 0 0 0 1 1 1 1 0 0
0 0 0 0 0 1 S S S 0
```

**Ground Truth:**
```
L L L L L L 0 0 0 0
0 1 1 1 1 0 0 0 0 0
0 0 1 1 0 0 0 0 0 0
0 0 1 1 0 0 0 0 0 0
0 0 1 1 1 0 0 0 0 0
0 0 1 1 1 1 0 0 0 0
0 0 0 1 1 1 1 0 0 0
0 0 0 1 1 1 1 1 0 0
0 0 0 0 1 1 1 1 0 0
0 0 0 0 0 1 S S S 0
```

## J.18 GEMINI 2.5 PRO - 1 RANDOM CELL HARD

### Gemini 2.5 Pro – 1 Random Cell Hard

**Prompt:**
You are given a structural material distribution represented as a grid. Each cell can
have one of the following states:
- 'L' indicates applied load.
- 'V' indicates void.
- 'S' indicates support.

The goal is to predict the correct material distribution by filling in all 'V' cells with
a floating-point number between 0 and 1 (with one decimal place), based on the surrounding
structure and implicit physical reasoning (such as load paths, supports, and forces).

Important: The completed structure should use as little material as possible while
remaining stable and plausible for carrying the applied forces.

Below is the input grid with masked regions:

```
0.0 L   V   L   L   L   L   L   L   0.0
0.0 0.0 0.2 0.3 0.1 0.1 0.3 0.2 0.0 0.0
0.0 0.0 0.1 0.4 0.0 0.0 0.4 0.1 0.0 0.0
0.0 0.0 0.0 0.4 0.0 0.0 0.4 0.0 0.0 0.0
0.0 0.0 0.0 0.4 0.0 0.0 0.4 0.0 0.0 0.0
0.0 0.0 0.0 0.4 0.0 0.0 0.4 0.0 0.0 0.0
0.0 0.0 0.0 0.4 0.0 0.0 0.4 0.0 0.0 0.0
0.0 0.0 0.0 0.4 0.1 0.1 0.4 0.0 0.0 0.0
0.0 0.0 0.0 0.4 0.1 0.1 0.4 0.0 0.0 0.0
0.0 S   S   S   S   S   S   S   S   0.0
```

**Model Completion (differences in red):**
```
0.0 L   0.2 L   L   L   L   L   L   0.0
0.0 0.0 0.2 0.3 0.1 0.1 0.3 0.2 0.0 0.0
0.0 0.0 0.1 0.4 0.0 0.0 0.4 0.1 0.0 0.0
0.0 0.0 0.0 0.4 0.0 0.0 0.4 0.0 0.0 0.0
0.0 0.0 0.0 0.4 0.0 0.0 0.4 0.0 0.0 0.0
0.0 0.0 0.0 0.4 0.0 0.0 0.4 0.0 0.0 0.0
0.0 0.0 0.0 0.4 0.0 0.0 0.4 0.0 0.0 0.0
0.0 0.0 0.0 0.4 0.1 0.1 0.4 0.0 0.0 0.0
0.0 0.0 0.0 0.4 0.1 0.1 0.4 0.0 0.0 0.0
0.0 S   S   S   S   S   S   S   S   0.0
```

**Ground Truth:**
```
0.0 L L L L L L L 0.0
0.0 0.0 0.2 0.3 0.1 0.1 0.3 0.2 0.0 0.0
0.0 0.0 0.1 0.4 0.0 0.0 0.4 0.1 0.0 0.0
0.0 0.0 0.0 0.4 0.0 0.0 0.4 0.0 0.0 0.0
0.0 0.0 0.0 0.4 0.0 0.0 0.4 0.0 0.0 0.0
0.0 0.0 0.0 0.4 0.0 0.0 0.4 0.0 0.0 0.0
0.0 0.0 0.0 0.4 0.0 0.0 0.4 0.0 0.0 0.0
0.0 0.0 0.0 0.4 0.1 0.1 0.4 0.0 0.0 0.0
0.0 0.0 0.0 0.4 0.1 0.1 0.4 0.0 0.0 0.0
0.0 S   S   S   S   S   S   S   S   0.0
```

## J.19    GEMINI 2.5 PRO - FULL EASY

---

**Gemini 2.5 Pro – Full Easy**

**Prompt:**
You are given a structural material distribution represented as a grid.  Each cell can
have one of the following states:
– 'L' indicates applied load.
– 'V' indicates void.
– 'S' indicates support.

The goal is to predict the correct material distribution by filling in all 'V' cells
with either '1' (solid) or '0' (empty), based on the surrounding structure and implicit
physical reasoning (such as load paths, supports, and forces).

Important:  The completed structure should use as little material as possible while
remaining stable and plausible for carrying the applied forces.

Below is the input grid with masked regions:

```
0 0 0 L L L L 0 0 0
V V V V V V V V V V
V V V V V V V V V V
V V V V V V V V V V
V V V V V V V V V V
V V V V V V V V V V
V V V V V V V V V V
V V V V V V V V V V
V V V V V V V V V V
0 0 0 0 S S S 0 0 0
```

**Model Completion (differences in red):**

```
0 0 0 L L L L 0 0 0
0 0 0 1 1 1 1 0 0 0
0 0 0 0 1 1 1 0 0 0
0 0 0 0 1 1 1 0 0 0
0 0 0 0 1 1 1 0 0 0
0 0 0 0 1 1 1 0 0 0
0 0 0 0 1 1 1 0 0 0
0 0 0 0 1 1 1 0 0 0
0 0 0 0 1 1 1 0 0 0
0 0 0 0 S S S 0 0 0
```

**Ground Truth:**

```
0 0 0 L L L L 0 0 0
0 0 0 0 1 1 0 0 0 0
0 0 0 0 1 1 0 0 0 0
0 0 0 0 1 1 0 0 0 0
0 0 0 0 1 1 0 0 0 0
0 0 0 0 1 1 0 0 0 0
0 0 0 0 1 1 0 0 0 0
0 0 0 0 1 1 0 0 0 0
0 0 0 0 1 1 0 0 0 0
0 0 0 0 S S S 0 0 0
```

---

## J.20 GEMINI 2.5 PRO - FULL HARD

### Gemini 2.5 Pro – Full Hard

**Prompt:**
You are given a structural material distribution represented as a grid. Each cell can have one of the following states:
- 'L' indicates applied load.
- 'V' indicates void.
- 'S' indicates support.

The goal is to predict the correct material distribution by filling in all 'V' cells with a floating-point number between 0 and 1 (with one decimal place), based on the surrounding structure and implicit physical reasoning.

Below is the input grid with masked regions:

```
0.0 0.0 L L L 0.0 0.0 0.0 0.0 0.0
V   V   V V V  V   V   V   V
V   V   V V V  V   V   V   V
V   V   V V V  V   V   V   V
V   V   V V V  V   V   V   V
V   V   V V V  V   V   V   V
V   V   V V V  V   V   V   V
V   V   V V V  V   V   V   V
V   V   V V V  V   V   V   V
S   S   S S S  S   S   S   0.0
```

**Model Completion (differences in red):**
```
0.0 0.0 L   L   L   0.0 0.0 0.0 0.0 0.0
0.0 0.1 1.0 1.0 1.0 0.1 0.0 0.0 0.0 0.0
0.0 0.2 1.0 1.0 1.0 0.2 0.0 0.0 0.0 0.0
0.1 0.3 1.0 1.0 1.0 0.3 0.1 0.0 0.0 0.0
0.2 0.4 1.0 1.0 1.0 0.4 0.2 0.0 0.0 0.0
0.3 0.5 1.0 1.0 1.0 0.5 0.3 0.1 0.0 0.0
0.4 0.6 1.0 1.0 1.0 0.6 0.4 0.2 0.0 0.0
0.5 0.7 1.0 1.0 1.0 0.7 0.5 0.3 0.1 0.0
0.6 0.8 1.0 1.0 1.0 0.8 0.6 0.4 0.2 0.0
S S S   S   S   S S S S 0.0
```

**Ground Truth:**
```
0.0 0.0 L   L   L   0.0 0.0 0.0 0.0 0.0
0.0 0.0 0.0 0.9 0.0 0.0 0.0 0.0 0.0 0.0
0.0 0.0 0.0 0.8 0.0 0.0 0.0 0.0 0.0 0.0
0.0 0.0 0.0 0.8 0.0 0.0 0.0 0.0 0.0 0.0
0.0 0.0 0.0 0.8 0.0 0.0 0.0 0.0 0.0 0.0
0.0 0.0 0.0 0.9 0.0 0.0 0.0 0.0 0.0 0.0
0.0 0.0 0.1 0.8 0.1 0.0 0.0 0.0 0.0 0.0
0.0 0.0 0.1 0.8 0.1 0.0 0.0 0.0 0.0 0.0
0.0 0.0 0.2 0.8 0.2 0.0 0.0 0.0 0.0 0.0
S   S   S   S   S   S   S   S   S   0.0
```

