# OpenReview forum: "SPhyR: Spatial-Physical Reasoning Benchmark on Material Distribution"
_ICLR.cc/2026/Conference — Submitted to ICLR 2026_

### Official Review · Reviewer_pXqj · 2025-10-29

**Soundness:** 2
**Presentation:** 2
**Contribution:** 1
**Rating:** 2
**Confidence:** 3

**Summary:**

In this work, the authors explored LLM's reasoning ability on topology optimization problems as a new benchmark. The problem is defined as material distribution in grid cells, under the constraints of fixed points and applied load. Through some 2D test examples, they concluded that LLMs so far have limited understanding of spatial physical reasoning, and advocated more future research to be done in this area.

**Strengths:**

- The problem definition and setup is very clear.
- The material is easy to follow.

**Weaknesses:**

Overall, I think the contribution of this work is very limited. It does not invent new techniques, or improve any existing frameworks. So I consider it well below the par of acceptance. Here to list a few weaknesses:

- The quantitive analysis is not very convincing. Since the solution is in form of binary values, 0s and 1s, there is 50% chance LLM can have a right guess. To me it is hard to tell if a model actually has a good physical-spatial understanding or it relies on random guess, especially for those with 50% matching score.
- Some topology optimization problems can have multiple plausible solutions. The evaluation presented does not consider the cases when a model outputs an alternative solution.

**Questions:**

- What is your definition of difficulty level based on? Size of the domain or number of random cells, or boundary conditions?
- Is it possible to test on 3D problems, and how about domains of other shapes?
- Can you explain why some models perform better than the others on this task?

---

> ### Author Response · Authors · 2025-11-17
> **Response to: Official Review of Submission14430 by Reviewer pXqj**
>
> Dear Reviewer pXqj,
>
> Thank you very much for taking the time and reading through our work, your feedback and consideration is incredibly valuable to us.
>
> We would like to address some of your concerns:
>
> ---
>
> **Weaknesses:**
>
> > "The quantitive analysis is not very convincing. Since the solution is in form of binary values, 0s and 1s, there is 50% chance LLM can have a right guess. To me it is hard to tell if a model actually has a good physical-spatial understanding or it relies on random guess, especially for those with 50% matching score."
>
> The case you describe is only valid for the "1_random_cell_easy" subject, but it's a great point we will mention in our manuscript for users to be cautious. However, all other subjects have more than 1 cell to be predicted i.e.: 5-10_random_cell, 1-3_random_row, 1-3_random_column or full setups. Additionally, the "_hard" subjects do NOT use binary values, to precisely tackle that issue.
>
> > Some topology optimization problems can have multiple plausible solutions. The evaluation presented does not consider the cases when a model outputs an alternative solution.
>
> Your statement is correct, they can have multiple plausible solutions within a threshold. However, this is only relevant for the two "full" tasks, where a model is prompted to predict the full distribution space. The vast majority of tasks 14/16 are masking tasks.
>
> ---
>
> **Questions:**
>
> > What is your definition of difficulty level based on? Size of the domain or number of random cells, or boundary conditions?
>
> The levels of difficulty we introduce, as also mentioned above, are easy and hard. Easy is binary values, and hard are floating point numbers with one numbers with one decimal place.
>
> > Is it possible to test on 3D problems, and how about domains of other shapes?
>
> It is possible to test 3D problems, we have included a subset of data in our github repository. However we planned to introduce that part of the dataset in a later work, especially since current SOTA models perform not very well on the 2D data. Introducing other shapes is a fantastic idea, but currently out of scope.
>
> > Can you explain why some models perform better than the others on this task?
>
> We were hoping our results and discussion sections would explain that, but we take your feedback by heart, as it seems this has not been answered propperly.
>
> ---
>
> Once again, thank you very much for your time and consideration, we apprechaite the effort.

---

> > ### Comment · Reviewer_pXqj · 2025-11-24
> >
> > Thank you for your further explanation. However, I decided to maintain the score.

---

> ### Author Response · Authors · 2025-11-22
> **Rebuttal Revisions Implemented**
>
> Dear Reviewer pXqj,
>
> Once again, thank you for your time and effort. We would like to inform you that we have just uploaded a revision to our manuscript and have addressed your review including the following action-items:
>
> ## Improve Quantitative and Qualitative Analysis
>
> A fellow reviewer of yours had suggested to improve the metrics used in this paper to gain mor insight into model behaviour. We have done so and made the following changes to our metrics in the following three families:
>
> - Reconstruction Accuracy Metrics
>     - exact_match (Old)
>     - score (Old, removed)
>     - normalized_score (Old, removed)
>     - difference ratio (New)
>     - penalised difference ratio (New)
>     - relative difference ratio (New)
>     - difficulty weighted difference ratio (New)
>     - difficulty weighted relative difference ratio (New)
> - Topology Metrics (New)
>     - grid validity (bool)
>     - load_support_connected (bool)
>     - load_support_connected_force_directional (bool)
>     - relative_unsupported_loads (float)
>     - isolated_clusters_count (int)
>     - difficulty score (float)
>     - difficulty weighted (float)
> - Physics-Approximated Metrics (New)
>     - force_path_cost_average_efficiency_ratio (float)
>
> Doing so helped us address your comment, and we are happy to report that we have significantly updated our analysis: Results, Discussion and Conclusion have been re-worked and we hope these changes address your comments adequately.
>
> ## Improve Difficulty Term Clarity
>
> We have made sure in text, tables and figures that it becomes clear what we mean with easy and hard difficulty (binary / continous).
>
> ## Explain Individual Model Performance
>
> We believe we have done justice to the point you have raised with the revised Results, Discussion and Conclusion section. We have detailed our qualitative and quantitative observations and reveal interesting insight, also on model-level.
>
> ---
>
> We would like to extend continued appreciation for additiona efforts and time spent, this is much appreciated.

---

### Official Review · Reviewer_TmWr · 2025-10-31

**Soundness:** 3
**Presentation:** 3
**Contribution:** 3
**Rating:** 8
**Confidence:** 2

**Summary:**

This paper introduces SPhyR, a benchmark dataset that evaluates Large Language Models' spatial and physical reasoning capabilities through topology optimization tasks. Models must predict optimal material distributions in 2D grids given boundary conditions (loads and supports) without access to simulation tools, with tasks ranging from filling single masked cells to predicting complete structures.

**Strengths:**

- Addresses an underexplored gap in LLM evaluation by testing physically-grounded spatial reasoning rather than just linguistic or visual tasks.
- Provides graduated difficulty levels (easy/hard) and multiple task variants (cell, row, column, full structure), enabling granular analysis of model capabilities
- Evaluates 10 state-of-the-art models with multiple experimental setups (rotations, prompt variations) and well-defined metrics

**Weaknesses:**

- Restricted to small 10×10 grids and relatively simple 2D scenarios, which may not fully reveal model limitations or generalize to realistic structural problems.
- Topology optimization can have multiple valid solutions; the paper doesn't address whether alternative plausible structures should be considered correct

**Questions:**

See Weaknesses.

---

> ### Author Response · Authors · 2025-11-17
> **Response to: Official Review of Submission14430 by Reviewer TmWr**
>
> Dear Reviewer TmWr,
>
> Thank you very much for your review. We are happy to see the strengths that you have raised, and we also believe that our benchmark attempting to close a gap in LLM evaluation on physically-grounded spatial reasoning as a non-linguistic setting.
>
> We would like to comment on the weaknesses raise:
>
> ---
>
> **Weaknesses:**
>
> > Restricted to small 10×10 grids and relatively simple 2D scenarios, which may not fully reveal model limitations or generalize to realistic structural problems.
>
> We intentionally use small 2D grids to isloate core spatial and physical reasoning, minimze confounding factors such as resolution, but especially numerical noise. We believe that this aligns with other reasoning benchmarks that start with simple settings (CLEVR, bAbl, ARC). We provide 3D data in our github repository, however due to relatively low performance on SOTA models on our 2D benchmark, we believe it makes sense to introduce and explore this in a follow-up work.
>
> > Topology optimization can have multiple valid solutions; the paper doesn't address whether alternative plausible structures should be considered correct
>
> That is a good point to be raised, however, we believe this is only relevant for the two "full" tasks, where a model is prompted to predict the full distribution space. The vast majority of tasks 14/16 are masking tasks, where we believe this has no high impact.
>
> ---
>
> We thank for the review and are happy that the strengths raised align with our motivation for the benchmark. Thank you for your time and consideration, we appreciate this!

---

### Official Review · Reviewer_5ydt · 2025-10-31

**Soundness:** 2
**Presentation:** 2
**Contribution:** 2
**Rating:** 2
**Confidence:** 3

**Summary:**

The authors introduce SPhyR a new benchmark for evaluating spatial and physical reasoning ability of  LLM about material distribution. LLMs are provided with 2D boundary conditions, applied loads and supports, and must predict the stable material layout that satisfies those constraints.

**Strengths:**

The main strength of the paper is that:

- It is the first benchmark for this particular task of reasoning about optimal material distribution.

**Weaknesses:**

The main weaknesses in the paper are as follows:
- It is unclear why would we need to benchmark LLMs for this particular task ? Are LLMs expected to provide more optimal answers than numerical solvers ? Is it going to be faster?
- If we need to empower LLMs with physical reasoning ability, doesn't it make more sense to just give LLMs access to a numerical solver tool or a simulation tool ? Why is there a need for providing LLMs with the intrinsic ability to solve a physics problem ?
- There is also no clear motivation as to why LLMs could be better than regular ML methods that rely on PINNs or CNNs.
- Figure 4 and Figure 5 are poorly presented with very small fonts.

**Questions:**

Why is this task important for LLMs specifically ? What will LLMs offer better than traditional solvers or PINNs or CNNs ? I understand that this a benchmark paper, but the authors should clarify what scientific insight or practical capability we gain if future LLMs achieve high performance on SPhyR.

---

> ### Author Response · Authors · 2025-11-17
> **Response to: Official Review of Submission14430 by Reviewer 5ydt**
>
> Dear Reviewer 5ydt,
>
> We thank you for your time and consideration, going through our work and remarks made. Especially for acknowledging that this is the first benchmark for "reasoning about optimal material distribution".
>
> We would like to respond to the weaknesses and questions raised by you:
>
> ---
>
> **Weaknesses**
>
> > It is unclear why would we need to benchmark LLMs for this particular task ? Are LLMs expected to provide more optimal answers than numerical solvers ? Is it going to be faster?
>
> We do agree that numerical solvers will remain (most likely for a long time) gold standard for computing topology optimization structures, however we do not intend to replace solvers, nor do we claim that LLMs will outperform them, neither in performance nor in speed. Instead, we are motivated comes from the fact that LLMs are increasingly used as general purpose reasoning agents, where they must interpret physical setups (world models) and potentially make decisions that depend on structural or mechanical intuition (for exampl ein robotics, design co-pilots or even CAD assistants).
> The purpose of the benchmark is NOT to replace solvers, but evaluate the models capability to reason about cause-and-effect in spatial-physical scenarios.
>
> > If we need to empower LLMs with physical reasoning ability, doesn't it make more sense to just give LLMs access to a numerical solver tool or a simulation tool ? Why is there a need for providing LLMs with the intrinsic ability to solve a physics problem ?
>
> Fantastic point raised: Tool use and intrinsic reasoning are complementing each other. Even we we include tool access, still the LLM must correctly formulate solver inputs, interpret outputs and detect implausible states before calling a tool. We think approximate physical intuition, instead of exact simulation, is important, even when (simulation-)tools are available.
>
> > There is also no clear motivation as to why LLMs could be better than regular ML methods that rely on PINNs or CNNs.
>
> We do not claim for LLMs to outperform PINNs/CNNs. However, these specialized ML models solve the numeris, but the LLM offer advantages such as reasoning over natural language, partial geometry and symbolic constraints, zero-shot or few-shot adaptability, and finally combining physical reasoning with spatial constraints and conditions.
>
> > Figure 4 and Figure 5 are poorly presented with very small fonts.
>
> Thank you, we will improve legibility.
>
> ---
>
> **Questions:**
>
> > Why is this task important for LLMs specifically ? What will LLMs offer better than traditional solvers or PINNs or CNNs ?
>
> To continue that line of question: "And what do we gain if they perform well?"
>
> As mentioned before, our goal is not to replace specialized solvers or compete on performance of such. LLMs/AI-Systems are used for reasoning, interpretation and interactive decision making. SPhyR evaluates abilities that traditional models do not address such as: understanding loads, supports and geometry from natural language or partial inputs, but also anticipating structural behavoir, and finally supporting interactive design, planning, and tool use where physical intuition is required.
>
> This means, if a model/system has a high performance on SPhyR, this would show that LLMs can reliably reason about force flow and structural stability. These abilities are essential for safe, general-purpose agents in design, robotics and engineering.
>
> ---
>
> Once again, thank you for raising great points, taking the time and we thank you for your efforts. This is much appreciated.

---

> ### Author Response · Authors · 2025-11-22
> **Rebuttal Revisions Implemented**
>
> Reviewer 5ydt,
>
> We would like to thank you again for your time and consideration. We would like to inform you that we have just uploaded a revision. We took your review by heart and identified the following action-items:
>
> ## Improve Motivation for thie Benchmarking Dataset
>
> We have significantly shapred our argument and motivation why it is important for LLMs to perform well on our benchmark. Specifically our newly introduced metrics have revealed deep insights into model behaviour and failure modes. Please refer to our revised Metrics section as well as Results, Discussion and Conclusion sections. Below, please find an overview of updated metrics:
>
> - Reconstruction Accuracy Metrics
>     - exact_match (Old)
>     - score (Old, removed)
>     - normalized_score (Old, removed)
>     - difference ratio (New)
>     - penalised difference ratio (New)
>     - relative difference ratio (New)
>     - difficulty weighted difference ratio (New)
>     - difficulty weighted relative difference ratio (New)
> - Topology Metrics (New)
>     - grid validity (bool)
>     - load_support_connected (bool)
>     - load_support_connected_force_directional (bool)
>     - relative_unsupported_loads (float)
>     - isolated_clusters_count (int)
>     - difficulty score (float)
>     - difficulty weighted (float)
> - Physics-Approximated Metrics (New)
>     - force_path_cost_average_efficiency_ratio (float)
>
> ## Improve Figure Quality
>
> Almost all figures have undergone work to improve legibility and visual quality including prompt examples in the Appendix.
>
> ---
>
> Thank you very much for your continued and potentially future effort and time, this is much appreciated, thank you.

---

### Official Review · Reviewer_scL9 · 2025-11-01

**Soundness:** 2
**Presentation:** 3
**Contribution:** 2
**Rating:** 2
**Confidence:** 3

**Summary:**

This paper provides a spatio-physical benchmark for LLMs, derived from 2D topology optimization problems with a variety of load and fixed support configurations. The dataset includes a ground truth material distribution (as computed by a commercial software suite) and corresponding queries. Each query provides a partially- or fully-masked version of the ground truth, together with a natural language prompt requesting that the model complete the missing elements in a way that uses minimal material while supporting the specified load. The authors benchmark several models using this dataset to form a preliminary baseline, and provide some discussion of the models' performance and common pitfalls. The paper also includes some additional experiments testing alternative formulations, such as rotated examples or query formulations featuring more or less physical intuition.

**Strengths:**

The premise of the paper is intriguing and useful as a general benchmark, as it would certainly be beneficial to query whether models can reliably reason over physical forces, constraints, material interactions, and structural connectivity. Topology optimization seems like a great task through which to measure such reasoning capabilities for VLM/LLMs, especially because the pixel/voxelized format of the problem naturally translates to several modalities, including text serialization as posited in this paper.

The gradation of tasks is clever and poised to provide nuanced insight when used as a benchmark. There are also ample opportunities for extensions based off of this dataset, such as using the samples for curriculum learning to impart progressively more structural knowledge in a model.

The paper is positioned well within existing literature, including a very informative overview of related approaches and their distinctions.
The paper is well structured and illustrated, including enough detail for a non-domain expert to follow it.

**Weaknesses:**

1. My main concern is that the evaluation metrics seem far too simplistic.
	- This is especially true in the "Score" metric for the harder continuous scenario, where, for a ground-truth cell with value 0.6, model predictions of 0 or 0.7 would be marked equally wrong. This sort of all-or-nothing scoring approach based on string matching seems to encourage pattern matching or memorization over physically-based reasoning -- which is precisely the issue this paper claims to address. Relative differences would be a good start. But more critically, it seems like at least one evaluation metric should incorporate a physical simulation -- or at the very least, a force path analysis -- to assign a score based on functionality.
	- It also seems odd to equally penalize errors in cells that were originally masked vs. those that were fixed but the model changed anyway. At worst, fixed-but-changed deviations seem like flagrant disregard for (or forgetting of) the task, so they should be penalized harshly. At best, I'm curious: are there any cases where it might change them to arrive at a different but equally valid or better solution? In that case, they indicate great physical understanding and global reasoning, so perhaps they should be rewarded according to the physical performance (though perhaps still with some penalty for rule-breaking).
	- The discussion in Section 6.2 seems like it comes closer to the core of the task evaluation than any of the official metrics. For example, building the isolated material islands clearly suggests a lack of physical reasoning; a metric that identifies and penalizes this sort of behavior seems far better poised to tease out and rank/reward the model's underlying abilities. Consider also the following cases: (1) a masked cell that mimics all of its surrounding neighbors, where an easy pattern recognition like flood fill would yield the right answer; (2) a masked cell that opposes all of its neighbors, where something like flood fill would be wrong; and (3) a masked cell on an established boundary, where the neighbors are split and it's unclear which way to go. A model's correct prediction on cases (2) or (3) should be more highly rewarded than case (1). If the paper were to develop metrics based around such domain-specific observations, this dataset could offer considerably more than it does in its current form.

2. The formulation of topology optimization used in this dataset is somewhat unclear. Could you clarify the the precise objective that you used in Rhino, and comment on whether/how well that aligns with the one sought by your prompts? Often, topology optimization is "minimum compliance subject to a maximal volume", which usually leads to a well defined solution. However, the phrasing on l.135 suggests that compliance and volume should both be minimized -- leading to multi-objective optimization with a Pareto front of possible solutions. On l.149 (and in the primary prompt template), the phrasing seems to imply something different still: the minimal volume that provides a structure capable of withstanding the forces (with an unspecified relationship between the acceptable level of compliance/stiffness that constitutes "withstanding"). Only the Physics-Enhanced prompt in Appendix F.2 seems specific: as little material as possible, with at least one complete load path. I imagine that each one of these formulations might lead to slightly different TO solutions (and thus, confusion within the benchmark if there are multiple, conflicting, or unclear answers).

**Questions:**

1. I wonder how a more explicitly visual medium might impact performance on this task -- e.g., I imagine the tokenization/attention is different for a text-based grid vs. an explicit 2D image domain. Have you considered a parallel benchmark that supplies images in a VLM? If you wanted to stay in a text based domain, I'd be curious whether/how the LLM response differs if the grid is framed as e.g. a numpy array. I also suspect that the row/column bias discussed in Section 6.1 might be due to the serialized text grid, as rows remain together but columns require models to attend to information that is likely spread over more distant tokens.

2. What motivated your choice to have the model return a full material array, rather than simply returning the masked elements. Did you try the latter, and if so, were there any notable differences?

3. Does the Grasshopper/Millipede solution always converge to a suitably correct answer within the allotted iterations? Do you check for convergence before adding to the dataset? Is there always 1 unique solution for a given setup, or can there be multiple?

4. How many distinct TO configurations are contained in the dataset? It would also be interesting to have a breakdown regarding e.g. how many have holes in the structure (to give a sense of solution complexity, and whether simple pattern recognition like floodfill might often work to solve topology optimziation without necessarily needing physical reasoning)

5. Could you provide additional information about the 100 examples used for evaluation? e.g., how many of each task type? how many distinct TO configurations are represented?

6. The interpretation of experiments is often a bit sparse. For example, there is no analysis of the results shown in Figure 5, aside from a brief mention in the discussion.


## Minor comments
- l. 55, 86 -- This phrasing strikes me as a specialized argument that undersells your intention, and is somewhere in the ballpark of a tautology... "these benchmarks don't test performance on topology optimization, because they don't ask about topology optimization, but we do". It might be worth revisiting, to see if you can formulate a stronger motivating point.
- l. 210,215 -- the explicit mention of binary values was confusing, since you also have continuous-valued versions.
- l. 453 - this is the first and only mention of 3D samples in the paper to this point, which was jarring.
- Figure 2 could be moved to the supplement
- l. 287 - should be Figure 3?
- l. 303,304 - correction --> completion
- Fig 7 (Appendix) -- could you explain the meaning of the differently colored dots?
- I wonder if the terms "void" and "support" are misleading; I consistently find myself interpreting them as "no material" and "material" even though I know that's incorrect. Altering them in the prompts might have an effect on the model responses

---

> ### Author Response · Authors · 2025-11-17
> **Response to: Official Review of Submission14430 by Reviewer scL9 - [Part 1/2]**
>
> [Part 1/2]
>
> Dear Reviewer scL9,
>
> Thank you very much for your concise, information-rich, smart and action-item-driven raised weaknesses, questions and comments on phrasing, figures, styling and structure of the paper. We would also like to thank you for the summarisation and valuable strengths highlighted, which our motivation for the work aligns perfectly with. This is a fantastic review and incredibly valuable for the future of our work. Thank you!
>
> We are happy to comment on the points raised:
>
> ---
>
> **Weaknesses:**
>
> *1. My main concern is that the evaluation metrics seem far too simplistic.*
>
> > This is especially true in the "Score" metric for the harder continuous scenario, where, for a ground-truth cell with value 0.6, model predictions of 0 or 0.7 would be marked equally wrong. This sort of all-or-nothing scoring approach based on string matching seems to encourage pattern matching or memorization over physically-based reasoning -- which is precisely the issue this paper claims to address. Relative differences would be a good start. But more critically, it seems like at least one evaluation metric should incorporate a physical simulation -- or at the very least, a force path analysis -- to assign a score based on functionality.
>
> Thank you for your insightful observation and analysis. We agree that an all-or-nothing match for continuous densities limits the evaluation signal. In the revision we will provide, as suggested, a relative difference metric. Also, including a force-path analysis is a fantastic idea, which we plan to incorporate. We were hoping for this benchmark to stay relatively light-weight, hence argue for physical simulation to be out of scope in this work. However, this is a great suggestion for future efforts. We still hope that the implementation of your suggested metric and validation improvements help to address your concern.
>
> > It also seems odd to equally penalize errors in cells that were originally masked vs. those that were fixed but the model changed anyway. At worst, fixed-but-changed deviations seem like flagrant disregard for (or forgetting of) the task, so they should be penalized harshly. At best, I'm curious: are there any cases where it might change them to arrive at a different but equally valid or better solution? In that case, they indicate great physical understanding and global reasoning, so perhaps they should be rewarded according to the physical performance (though perhaps still with some penalty for rule-breaking).
>
> We agree that these two error types reflect different behavior and will score fixed cells more strongly while masked-cell accuracy is evluated as before.
>
> > The discussion in Section 6.2 seems like it comes closer to the core of the task evaluation than any of the official metrics. For example, building the isolated material islands clearly suggests a lack of physical reasoning; a metric that identifies and penalizes this sort of behavior seems far better poised to tease out and rank/reward the model's underlying abilities. Consider also the following cases: (1) a masked cell that mimics all of its surrounding neighbors, where an easy pattern recognition like flood fill would yield the right answer; (2) a masked cell that opposes all of its neighbors, where something like flood fill would be wrong; and (3) a masked cell on an established boundary, where the neighbors are split and it's unclear which way to go. A model's correct prediction on cases (2) or (3) should be more highly rewarded than case (1). If the paper were to develop metrics based around such domain-specific observations, this dataset could offer considerably more than it does in its current form.
>
> This is fantastic feedback and we take this by heart. To address this we introduce a connectivity / load-path validity metric detecting isolated material islands and verify force to support connectivity. Additionally, we believe that the local-context difficulty categories is a super interesting idea to be reflected in the manuscript and included in the metrics. We will categorize samples and report results per category.
>
> *2.*
>
> > The formulation of topology optimization used in this dataset is somewhat unclear. Could you clarify ...
>
> Thank you for your thorough analysis of our work, and you are correct that the prior phrasing is a bit unfortunate and we understand that it created ambiguity. We will clarify this more explicitly. But here we can say already: All samples in the dataset have been generated using Millipedes standard density-based topology optimization (SIMP) method, minimizing compliance subject to a fixed colume faction. Following are parameters used in the solver:
>
> - Self Weight Coefficient is 0
> - Optimization Iterations is 10
> - Smoothing factor is 0.1
> - Penalization is 3.0
> - Target Density is 0.1
> - Minimum Density is 0.001
> - Delete Threshold is 0.5
> - Compliant Mechanism is False
>
> ---
>
> TBC

---

> ### Author Response · Authors · 2025-11-17
> **Response to: Official Review of Submission14430 by Reviewer scL9 - [Part 2/2]**
>
> [Part 2/2]
>
> CONTINUED
>
> ---
>
> **Questions:**
>
> > 1. I wonder how a more explicitly visual medium might impact performance ...
>
> This is a fantastic question, which really sparked ideas for extending this work. We agree that a visual formulation is a natural fit for this task, and we even already provide plots for all samples on our github repository. However in this first version we chose a serialized text grid to keep the benchmark accessible to text based LLM/Systems and isolate spatial-physical reasoning from the potentially additional noise of visual perception and image encoders. We could easily extend this using heatmaps or images (figures) as a natural extension to this work in the future. This would further help to disentangle perception from reasoning. In regards to the row/column bias: our rotation/directionality experiments, at least partially, support your hypothesis: when rotating the same structures, the performance gap between row and column tasks is reduced, suggesting that serialization and token locality indeed play a role. We will make this connection clearer in the manuscript. We could also add a short ablation using a more array-like format, to check whether this mitigates some of the asymmetry.
>
> > 2. What motivated your choice to have the model return a full ...
>
> Main motivation was to enforce a consistent output format across all tasks and detect "illegal" edits to unmasked cells as an additional signal about global consistency. But you are right, we could have only provided the masked areas as ground-truth and only prompt the model to return such. However we believe there is a higher risk for formatting related errors. Additionally, this helps with evaluation and metrics. It is a great suggestion to run masked-cells-only abblations as an alternative protocol, thank you.
>
> > 3. Does the Grasshopper/Millipede solution always converge ...
>
> All samples in the dataset are generated using Millipeds standard, density-based SIMP schema with fixed parameters (as mentioned above). These parameters are used consistenly across all configurations. Our goal is to obtain plausible and structurally meaningful density fields rather than a fully converged optimum. We do not require strict numerical convergence before including a sample. These parameters reiably produced coherent structures withjout artifacts.
>
> > 4. How many distinct TO configurations are contained in the dataset? It would also be interesting to have a breakdown regarding e.g. how many have holes in the structure (to give a sense of solution complexity, and whether simple pattern recognition like floodfill might often work to solve topology optimziation without necessarily needing physical reasoning)
>
> There are 1296 samples in the 2D dataset, each with a unique load and support configuration and a corresponding optimized material distribution. So we generate a fixed set of TO solutions and apply all task variants (8) and 2 difficulty levels: cells, rows, columns or full masking x easy/hard, resulting in 16 tasks. We agree that characterizing solution complexity is important and will add statistics such as the distribution of material volume fractions and simple connectivity descriptors (number of connected components and presence of internal voids/holes) - this is a great suggestion, thank you!
>
> > 5. Could you provide additional information about the 100 examples used for evaluation? e.g., how many of each task type? how many distinct TO configurations are represented?
>
> We present 100 distinct TO configurations, randomly sampled from the SPhyR benchmark dataset, for each task type (8) and difficulty (2), resulting in 1600 samples evaluated per model.
>
> > 6. The interpretation of experiments is often a bit sparse. For example, there is no analysis of the results shown in Figure 5, aside from a brief mention in the discussion.
>
> We agree that our discussion of the experimental results, including Figure 5, is currently too brief. In our revision we will expand Section 6 to highlight main trends more explicitly. This includes discussion on the sharp drop from easy to hard difficulties, the gap between local completions and full-structure predictions, the row/column asymmetry and its reduction under rotation and finally the limited but non-trivial gains from physics-enhanced prompting. We will also connect those our qualitative results where we identify samples with, for example, isolated islands, broken load paths etc. and hope that this will help us frame our argument that LLMs capture some local spatial regularities but struggle with global, physically grounded reasoning.
>
> ---
>
> **Minor comments**
>
> Thank you for providing this minor comments, the devil is in the detail, so we much appreciate this.
>
> ---
>
> Once again, a heartfelt thank you from our side, your feedback, observations, and thorough analysis of our paper is rather inspiring. Thank you for taking the time and making such an effort.

---

> ### Author Response · Authors · 2025-11-22
> **Rebuttal Revisions Implemented**
>
> Dear Reviewer scL9,
>
> Once again, we would like to extend our heartfelt appreciation. Your detailed and insightful analysis has helped us push our work further.
>
> The following action-items have been identified from your comments:
>
> ## Move Figure 2 to Appendix
> (Overview of task variations: predicting material distributions for N random cells, rows, columns, or a full structure 3.)
>
> ## Revisit metrics
>
> We have introduced the following changes: We now have three metrics families: Reconstruction Accuracy, Topology Metrics and Physics-Approximating Metrics:
>
> - Reconstruction Accuracy Metrics
>     - exact_match (Old)
>     - score (Old, removed)
>     - normalized_score (Old, removed)
>     - difference ratio (New)
>     - penalised difference ratio (New)
>     - relative difference ratio (New)
>     - difficulty weighted difference ratio (New)
>     - difficulty weighted relative difference ratio (New)
> - Topology Metrics (New)
>     - grid validity (bool)
>     - load_support_connected (bool)
>     - load_support_connected_force_directional (bool)
>     - relative_unsupported_loads (float)
>     - isolated_clusters_count (int)
>     - difficulty score (float)
>     - difficulty weighted (float)
> - Physics-Approximated Metrics (New)
>     - force_path_cost_average_efficiency_ratio (float)
>
> Consequently we have done the following changes in the manuscript:
> - re-run our metrics calculations
>     - updated all data, plots, figures and tables
>     - updated the Metrics Section to show precisely how we calculate the new introduced metrics
>     - furthermore we have added detailed prompt and completion examples for each newly introduced metrics in the Appendix
>     - finally additional detailed computation has been added for the Physics-Approximating Metric (Appendix)
>     - most importantly, with the new metrics introduced we were able to do much more detailed and deeper analysis, quantitatively and qualitatively of the results. we report our findings in the revised conclusion and discussion section.
>
> ## New "Topology Optimization Task" Section
>
> We have rephrased this section as per your suggestion to this:
> _Topology optimization determines an optimal material distribution within a domain under prescribed forces and supports. All dataset samples are generated using Millipede’s density-based SIMP formulation, solving a minimum-compliance problem with a fixed volume fraction (Appendix \ref{apx:solver-params} for solver parameters). This yields well-defined, single-objective solutions that capture characteristic load paths and material connectivity.\\
> In this work, we repurpose these topology optimization instances as reasoning tasks for LLMs. Instead of performing numerical optimization, models must predict plausible material distributions from forces, supports, and boundaries alone, requiring them to infer principles of load transfer, stability, and efficient material use, approximating the behavior of minimum-compliance topology optimization without access to simulation tools._
>
> ## Sparsity of Experiment Interpretation
>
> As described above, due to extended metrics design and evaluation we were able to prepare much more detailed and deep analysis of our results. This we report in our Results, Discussion and Conclusion sections.
>
> ## Minor Comments
>
> We have addressed all proposals for change from the "Minor Comments" section. The only question from this section and in general from all suggestions from the review is "I wonder if the terms "void" and "support" are misleading..." as time and resource constraints prevent us from running further ablations for now. We hope to include soon.
>
> ## Additional changes we have made:
>
> - Added metrics overview table
> - Added all test examples for Reconstruction, Topology and Physics Approximating Metrics in the Appendix
> - We significantly improved visual quality of prompt examples in the Appendix
>
> ---
>
> We hope we have addressed your questions and concerns adequately and are looking forward to any further suggestions and feedback, which have been invaluable so far.
>
> We have just uploaded a revision and thank in advance for any additional effort and time spent.

---

### Meta-Review · Area_Chair_grCn · 2026-01-02

**Summary:**

The paper introduces SPhyR, a benchmark designed to evaluate spatial and physical reasoning in LLMs through topology-optimization–derived material distribution tasks. The benchmark targets an underexplored evaluation regime beyond linguistic or visual perception, focusing instead on reasoning about loads, supports, connectivity, and force flow without access to simulation tools. One reviewer finds this direction timely and potentially impactful, while others remain unconvinced about the benchmark's broader scientific value or its relevance to LLM evaluation in particular.

A key strength of the submission lies in its conceptual framing: recasting topology optimization as a reasoning problem for LLMs. The benchmark is systematically constructed, with multiple task variants and difficulty levels that allow fine-grained analysis of model behavior. The authors evaluate a wide range of models and provide qualitative analyses of common failure modes, which are informative.

However, the dominant concerns raised by multiple reviewers relate to the depth of the contribution and the evaluation methodology. In the original submission, the evaluation metrics were viewed as overly simplistic, raising concerns about whether performance reflects genuine physical reasoning or superficial pattern completion. Several reviewers also questioned whether binary predictions permit meaningful conclusions. More fundamentally, reviewers expressed skepticism about the motivation for benchmarking LLMs on this task, arguing that numerical solvers, PINNs, or CNN-based approaches may be more appropriate, or that LLMs should instead be evaluated in conjunction with explicit tool use. As a result, the contribution was perceived primarily as a dataset release, with limited conceptual or methodological novelty.

Based on these considerations, I recommend rejection.

**Reviewer Concerns:**

The main concerns raised by the reviewers can be summarized as follows: 1) the evaluation metrics are overly simplistic; 2) the scientific meaningfulness and motivation of the task for LLMs are unclear; and 3) the work lacks a substantive technical contribution beyond dataset construction.

**Reviewer Scores:**

The reviewers did not actively engage in the rebuttal phase, and therefore no further discussion occurred. While the authors made a strong effort to address the raised concerns in their responses, these efforts were not sufficient to resolve the reviewers' core reservations.

---

### Decision · Program_Chairs · 2026-01-26

Reject